# Record-breaking emergence of upstream-downstream zonal-consistent variation in the Eurasian jet axis

Lifei Lin [1], Chundi Hu [1,2,3] ✉, Dake Chen [2,3], Renguang Wu [4], Tao Lian [2,3], Quanliang Chen[5], Zeming Wu [1] & Song Yang [2,6]

Eurasian weather/climate extremes are often considerably linked to the meandering and/or latitudinal shifts of upper-level westerly jet, but the variation and impact of the upstream-downstream zonal consistency (UDZC) in the Eurasian jet axis are less known. Here we report a record-breaking emergence of an enhanced UDZC during the past two-to-three decades. In particular, the upstream-downstream covariance has increased from <10% to >60%, with the Eurasian upper-tropospheric westerly wind anomalies dominated by the Eurasian jet intensity change mode, which prompts a large-scale interplay between Asian and Pacific subtropical highs and thereby induces mega-shifts in the distribution patterns of circulation and climate anomalies across Eurasia. Such large-range circulation adjustments are also accompanied by a circumglobal Silk Road teleconnection, a Rossby wavetrain with zonal wavenumber 6, which originates from the Northeast Atlantic and connects Asian climate with western Europe and North America heatwave and drought conditions. Climate model simulations suggest that the UDZC would further intensify as the northern mid-latitude warming becomes more pronounced. Our results highlight that the emergence of enhancing UDZC in the Eurasian jet axis is capable of triggering planetary-scale climate extremes even across the whole Northern Hemisphere, imposing more serious climate threats.

Weather/climate extremes (WCEs) have never stopped increasing since the beginning of the 21st century, especially heatwaves and intense precipitation[1–3], posing threats to ecosystems[1], human health[4], and food security[5]. Besides global warming[6], those phenomena have been linked to changes in the westerly jet stream[7–9], a rapid and narrow air current flowing from west to east in the upper troposphere. The jet stream acts as a bridge connecting WCEs over disparate regions of the Earth[5,9–11], even triggering synchronized low yield over major breadbasket regions[12]. For example, the meandering Eurasian jet connected

the record-breaking heatwave in Europe, severe floods in Pakistan, and prolonged heatwave over central and eastern China in summer 2022[10,13–15].

Theoretical frameworks have been proposed to explore the potential impact of the westerly jet stream, acting as a classical Rossby waveguide, on WCEs. A weakened jet stream hinders propagation of synoptic disturbances with a decreased meridional potential vorticity gradient[16,17], promotes breaking of synoptic waves on its poleward side, and amplifies planetary waves[17], potentially leading to

[1]State Key Laboratory of Ocean Sensing & Ocean College, Zhejiang University, Zhoushan, China. [2]Southern Marine Science and Engineering Guangdong Laboratory (Zhuhai), Zhuhai, China. [3]State Key Laboratory of Satellite Ocean Environment Dynamics, Second Institute of Oceanography, Ministry of Natural Resources, Hangzhou, China. [4]School of Earth Sciences, Zhejiang University, Hangzhou, China. [5]School of Atmospheric Sciences, Chengdu University of Information Technology, Chengdu, China. [6]School of Atmospheric Sciences & Guangdong Province Key Laboratory for Climate Change and Natural Disaster Studies, Sun Yat-sen University, Zhuhai, China. ✉e-mail: hucd@zju.edu.cn

atmospheric blocking[18]. Additionally, the jet stream strongly diminishes the efficiency of cross-jet mass and heat transport[19], which is conceptualized as the "mixing barrier" effect. Therefore, a strengthened jet stream, as an enhanced mixing barrier, leads to larger north-south temperature and moisture contrasts, consequently inducing large-scale changes in heatwave occurrences and drought conditions. Different structures of the upper-level jet stream also have great impacts on surface weather and climate anomalies, mainly via changes in the moisture transport[20], jet-related secondary circulation[21,22], activities of synoptic disturbance[23], blocking events[18], and Rossby wave behavior[24].

While the impacts of jet stream strength alterations on WCEs' distribution are acknowledged[18,19], existing research has predominantly focused on its long-term trends, meridional shifts, and meandering[25,26]. A significant knowledge gap exists in understanding the zonal coherence of jet stream intensity variations across vast longitudinal expanses, such as the Eurasian continent. The Eurasian subtropical westerly jet (ESWJ) during high summer, characterized by its extensive longitudinal range (-120°) and notable regional variations[27], presents a complex system where the degree of coordinated intensity changes between its upstream and downstream sectors—what we conceptualize as "upstream-downstream zonal consistency" (UDZC)—has been largely overlooked. Specifically, how this UDZC has evolved, particularly in recent decades, and the mechanisms through which it might govern the spatial synchronicity and large-scale patterns of WCEs during boreal summer, remain critical unanswered questions. Strong UDZC, implying a more uniform, continent-wide modulation of jet intensity, could potentially lead to more spatially extensive and synchronized WCEs[5,18,19], yet this linkage is poorly constrained.

This study addresses these crucial knowledge gaps by investigating the long-term evolution of UDZC and the climate impact of strong UDZC. We demonstrate an exceptional surge in Eurasian jet UDZC since the satellite era (i.e., the post-1979 period), a phenomenon unparalleled in the past two centuries. This surge is accompanied by the emergence of a distinct dominant paradigm in Eurasian upper-level circulation, characterized by jet intensity changes rather than solely latitudinal shifts. We further reveal its profound implications for widespread heatwave and drought occurrences across Eurasia and its teleconnections extending to North America.

## Results

### Record-breaking UDZC in the Eurasian jet

The Eurasian jet, peaking at 32 m s$^{-1}$ in climatology, is quite straight in the west-east direction with its axis situated around 42°N during high summer (July–August; Fig. 1a). There are two prominent activity centers over its western and eastern parts at 200-hPa zonal wind (U200; contours in Fig. 1a). The UDZC of the Eurasian jet during a specific period can be simply quantified by the co-variation between these two activity centers. Therefore, according to its two prominent high-variability ranges at U200, we divided it into the western part and eastern part (Fig. 1a), each with its own dynamical jet axis index: the western jet axis (WJA) index and eastern jet axis (EJA) index (see Methods). The EJA index shows a strong variability since 1999, with a record-breaking status in 2022 (Fig. 1b), during which East Asia experienced an extremely hot and dry summer[7,13,14,28].

Here, we use the correlation coefficient (R) between these two indices to quantify the state of UDZC in a specific period (see Methods). A positive and robust R indicates a strong UDZC state during this period, while a feeble or even negative R represents a weak UDZC state. Therefore, the interdecadal evolution of UDZC state can be simplified as the temporal evolution of R between WJA and EJA indices in a sliding window (Fig. 1c). A 21-year window is used here and the result is not sensitive to the choice of sliding window (Supplementary Fig. 1). Since our results before the satellite era can be greatly influenced due to the

scarcity of observational data, especially in-situ measurements, we choose to use multi-source reanalysis data for cross-validation (see Methods). Although there are some discrepancies between different datasets before the satellite era, a core finding is highly consistent across all datasets: a strong UDZC state in the Eurasian jet has been rare throughout the past century in all reanalysis data (Fig. 1c).

The WJA and EJA indices typically exhibit insignificant linkage, remaining low or even negative since the early 20th century. The correlation has increased markedly since the 1970s especially during the satellite era, and becomes significant after the late-1990s, with covariance between the two indices reaching 30% (Fig. 1c). Specifically, these two indices are highly correlated since the 1999 ($R = 0.79$, $p = 4.7 \times 10^{-6}$) while almost independent of each other before 1998 ($R = 0.03$, $p = 0.85$; Fig. 1b). Note that the results are not sensitive to the sliding windows (Supplementary Fig. 1) and the domain selection of WJA and EJA indices (Supplementary Table 1).

The strong consistency of the results after the late-1990s between datasets supports the robustness of the sudden emergence of a strong UDZC state in the Eurasian jet (Fig. 1b). This feature is unique to the high summer since the late-1990s, with the absence of similar signal in the early summer or other seasons (i.e., June, Supplementary Figs. 2 and 3; see Supplementary Text 1 for details). Using 80 members of the 20thCRV3 reanalysis, which extends the record back to 1806 (Fig. 1c), we further confirmed it as an exceptional phenomenon over the past nearly two centuries. The 20thCRV3 reanalysis exhibits relatively high uncertainty before the 1880s, as green shading indicates. Although the uncertainty is much smaller after the 1990s, their multi-member means still underestimate the strong UDZC (Fig. 1c, light green with shadings).

The shift in UDZC is more intuitive in the changing U200 distribution across periods. Unlike the relatively regional tripolar patterns prior to 1998 (Supplementary Fig. 4a, c), the post-1999 era shows largely synchronized variations between the two parts of the jet, resulting in a broad meridional tripolar pattern spanning the Eurasian continent (Supplementary Fig. 4b, d). Since the different parts of the Eurasian jet can influence surface climate in-situ, the more zonally extended pattern strongly implies more synchronized WCEs across Eurasia.

### Distinct paradigm of Eurasian circulation

Apart from the thermal-driven Eurasian jet[23,29], there is another jet stream over the Eurasian high-latitudes, which is called the polar jet or the eddy-driven jet[23,30]. It grows at the high-latitudes due to the active baroclinic eddies in-situ and is much weaker than the subtropical Eurasian jet[31]. A previous study has revealed the impact of the subtropical jet on polar jet, using an idealized multilevel primitive equation model[23]. It was found that the position where the baroclinic wave grows is modulated by the strength of the subtropical jet stream. Therefore, a weak Eurasian jet favors the development of a baroclinic wave at its 20°–30° poleward side, leading to a strengthened eddy-driven polar jet. This out-of-phase relationship is related to the atmospheric blocking with the phenomenon called double jets[3], favoring the occurrence of WCEs.

As shown in Supplementary Fig. 4, the strong UDZC state during P2 implies a strengthened mode with a zonally uniform "meridional triple pattern" spanning the entire Eurasian continent. Therefore, the emergence of a strong UDZC state in Eurasian jet strength since the late-1990s may coincide or reflect a notable change in the dominant mode of jet stream variations, with consequent change in the circulation pattern across multiple latitudes.

Since the reanalysis data after 1958 is more reliable due to the improved observation network after the International Geophysical Year[32], and to support the robustness of the main findings in this study, we further use the JRA-55 data (this official data has been provided since 1958) to reproduce the results obtained from ERA-5 data; therefore, we mainly focus on the post-1958 period unless otherwise

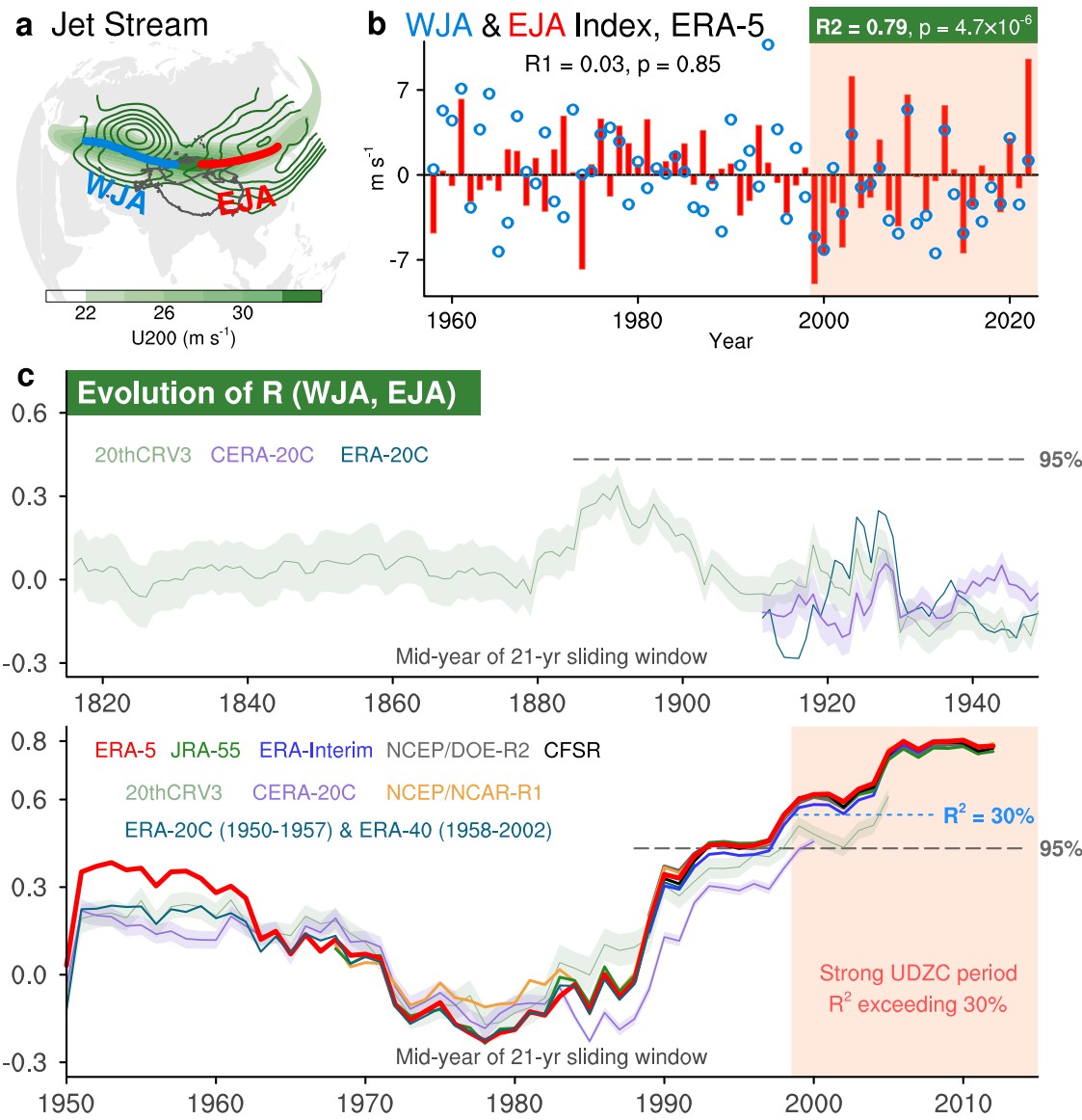

**Fig. 1 | Record-breaking surge of upstream-downstream zonal consistency (UDZC) in the Eurasian jet. a** Climatology of the Eurasian jet by the 200-hPa zonal wind (U200; green shading; units: m s⁻¹) and the standard deviation of U200 (contours; units: m s⁻¹). Blue and red line represent the climatological western jet axis (WJA) and eastern jet axis (EJA). **b** Year-to-year variability of WJA and EJA indices, with R1 and R2 representing the correlation coefficient between the two indices during the first period (1958–1998) and the second period (1999–2022). **c** The evolution of the UDZC measured by the 21-year sliding correlation between WJA and EJA indices in the Eurasian jet axis in multi-source data.

stated. To show changes in the dominant feature of Eurasian upper-level circulation, we divided the post-1958 period into P1 (1958–1998) and P2 (1999–2022) and performed the empirical orthogonal function (EOF) analysis on U200 during the two sub-periods, respectively (see Methods for details).

Eurasian jet variations during the pre-1998 periods are dominated by the meridional shift in both two leading modes (Fig. 2a, b). Specifically, the dominant characteristic during P1 is the well-known meridional displacement of the whole Eurasian jet (Fig. 2a)[27,33], followed by the much local signal of meridional displacement in the western part of the Eurasian jet (Fig. 2b).

However, a distinct meridional triple pattern has emerged as the dominant mode during P2 (Fig. 2c), which indicates a strengthened subtropical jet stream with a weakened polar jet and an intensified tropical easterly jet. The EOF1 mode in P2 spatially manifests as a zonally uniform "meridional triple pattern" spanning the entire Eurasian continent (Fig. 2c). The physical meaning of this pattern is a

simultaneous, continent-wide strengthening of the subtropical jet stream accompanied by a weakening of the polar jet and intensified tropical easterly jet. This continent-wide, synchronized intensity change is the direct physical manifestation of a high UDZC in the spatial field. Therefore, this "zonal consistent intensity change" mode strongly reflects a high UDZC state in Eurasian jet intensity variation and aligns with the dynamical out-of-phase relationship between the subtropical and polar jet streams[23,34]. For the EOF2 during P2, it exhibits similarity to the EOF1 in P1, but with much weaker signals over the western part (Fig. 2d).

Since different structures of the upper-level jet stream have great impacts on surface weather and climate anomalies, this shift from a "meridional shift" mode to an "intensity change" mode reflects a distinct paradigm of Eurasian upper-level circulation variation, with profound impacts on Eurasian WCEs.

Nevertheless, it remains unclear whether such a change in U200 EOF mode actually represents a change of the whole Eurasian

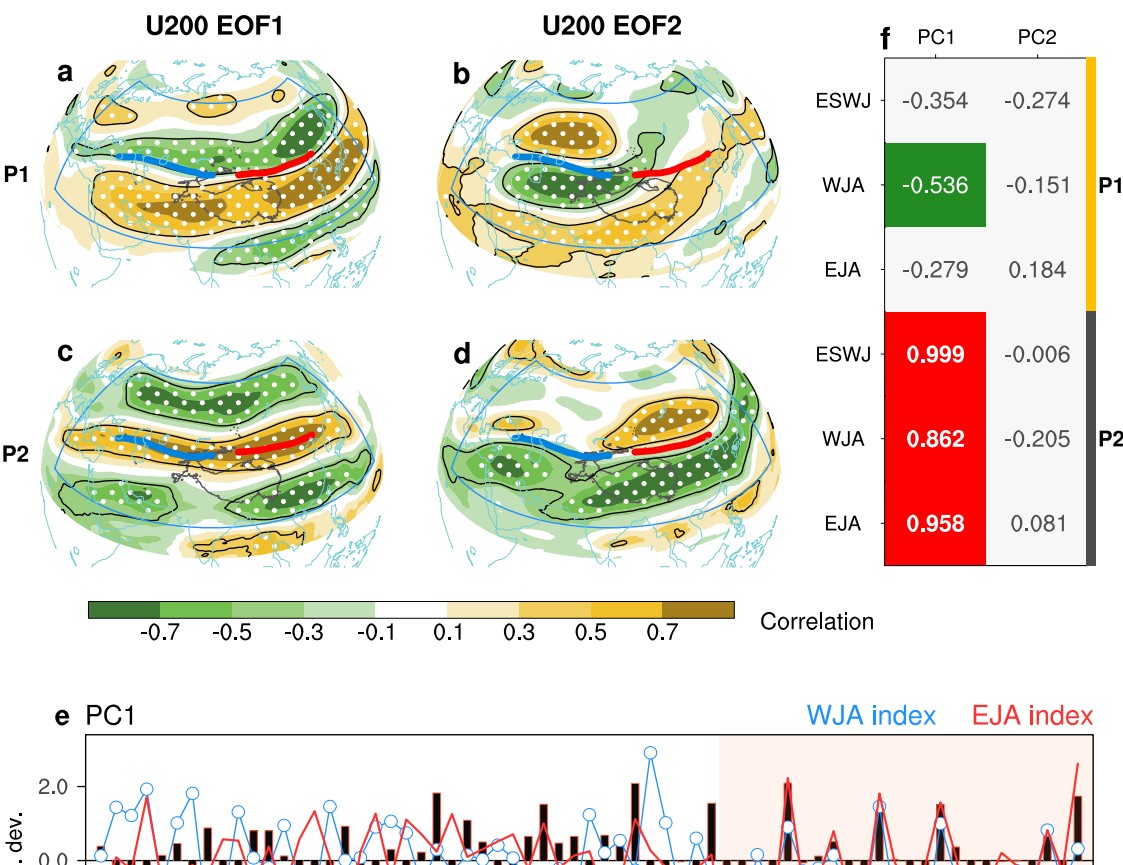

**Fig. 2 | Shifting dominant mode of Eurasian westerly.** The first two leading modes of empirical orthogonal function (EOF) of the 200-hPa zonal wind (U200; shading: correlation map) during the first period (P1; 1958–1998): the EOF1 (**a**) and EOF2 (**b**). The blue box denotes the regions for performing EOF. Black lines outline regions exceeding the 95% confidence level, with that after controlling for the false discovery rate ($\alpha_{FDR} = 0.10$) stippled. The Blue and red line represents the climatological western jet axis (WJA) and eastern jet axis (EJA) during P1. **c**, **d** Same as **a** and **b**, but for the second period (P2; 1999–2022). Also, the blue (red) line is obtained during P2. Shown in **e** are the normalized first principle component (PC1; black bar) corresponding to EOF1 in (**a**, **c**), as well as the WJA (blue curve) index and the EJA (red curve) index. **f** Correlations between U200 PCs and projected Eurasian subtropical westerly jet (ESWJ) index, WJA, and EJA indices for P1 and P2.

circulation and influences the consequent near-surface climate. To further investigate the role of such an "intensity change" mode in Eurasian circulation, we further performed a multi-variable EOF (MVEOF) combining U200 and the zonally averaged vertical cross-section of zonal wind from 500 to 100 hPa. The EOF1 exhibits a clear tripolar structure throughout the troposphere, even reaching stratosphere (Supplementary Fig. 5a), which accounts for 23.28% of the total variance, significantly higher than that of EOF2 (16.55%; Supplementary Fig. 5b). This mode is highly identical to that obtained using only U200, suggesting that the "zonal consistent intensity change" mode is a phenomenon throughout the troposphere rather than a confined upper-level mode.

In addition, we also performed an MVEOF of meridional circulation (represented by the meridional wind and vertical velocity); its EOF1 exhibits a two-cell meridional circulation over the south and north flanks of the Eurasian jet, with low-level southerly wind and northerly wind, respectively. The associated zonal wind anomalies also show a tripolar structure of zonal wind similar to EOF1 (Supplementary Fig. 6a and Supplementary Fig. 5a, right). Therefore, our results strongly suggest that the "zonal consistent intensity change" mode is the leading feature of Eurasian circulation throughout the troposphere.

To characterize the year-to-year variability of the UDZC phenomenon, we performed a projection of the U200 anomaly field during 1958–2022 onto the UDZC-like mode shown in Fig. 2c. Then, we termed the resultant time series the "projected ESWJ index" (Fig. 3a; see Methods for details). A strong positive (negative) value of this index is related to the uniformly strengthening (weakening) of the whole Eurasian jet strength. Therefore, this index exhibits very weak variability during the period of a weak UDZC state (P1), especially after 1979 due to the negative relation between WJA and EJA, but shows a sudden increase in variability after 1999 (Fig. 3a), consistent with the strong UDZC state during this period (Fig. 1c). In other words, the strengthening co-variability of WJA and EJA since the late-1990s has led to a strong UDZC state at decadal scale (Fig. 1c) and enhanced variability of the projected ESWJ index (Fig. 3a). Not only the year 2022, but also years of 1999, 2000, 2003, 2008, 2009, 2013 and 2015 exhibit stronger anomalies than any year during P1. Note that we have tested and confirmed that the UDZC-like mode is robust and not sensitive to the projection reference-period (see Supplementary Text 3, Supplementary Table 2 and Supplementary Figs. 7 and 8).

To further explore how the PCs of U200 are related to Eurasian jet indices (i.e., the ESWJ, WJA and EJA indices), we calculated their Rs

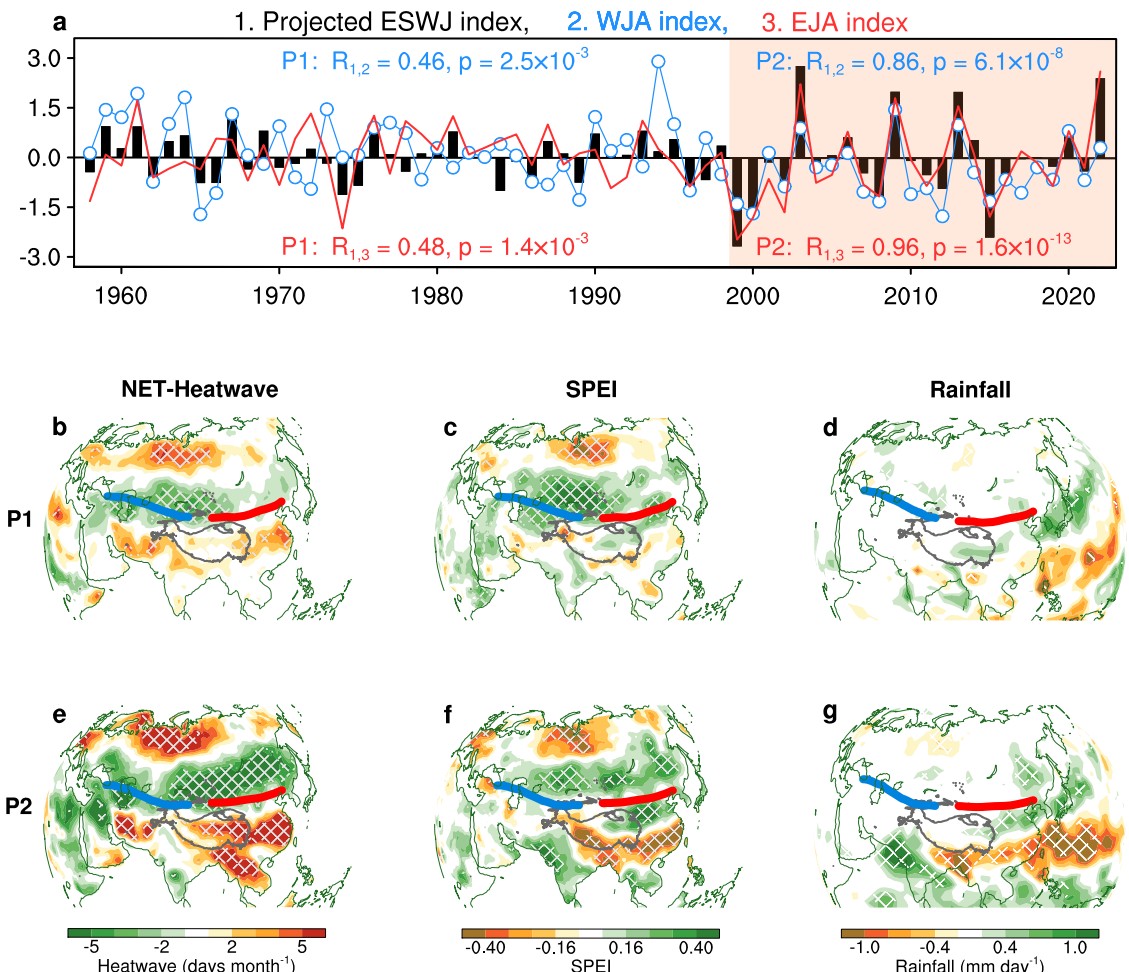

**Fig. 3 | Impacts of enhancing upstream-downstream zonal consistency on Eurasian weather/climate extremes. a** The normalized projected Eurasian subtropical westerly jet index (ESWJ; black bar) with the western jet axis (WJA; blue curve) and eastern jet axis (EJA; red curve) indices. **b** Regression map of heatwave frequency based on the net effective temperature (NET-heatwave) against the normalized projected ESWJ index during the first period (P1), with 95% confidence hatched. **c**, **d** Similar to **b**, but for the Standardized Precipitation Evapotranspiration Index (SPEI) and rainfall anomalies, respectively. The blue (red) line indicates the climatology of WJA (EJA), similar to that in Fig. 1a but obtained during P1. **e–g** Similar to **b–d**, but for the second period (P2). Also, the blue (red) line is obtained during P2.

during two sub-periods (Fig. 2f). The projected ESWJ index is nearly independent of the other PCs (Fig. 2f), except for the PC1 during P2, and is highly correlated with the WJA and EJA indices during P2 (R reaches 0.86 and 0.96 respectively; Fig. 3a). Therefore, the climate impacts of the Eurasian jet with strong UDZC at interannual scale can be illustrated by a comparison of the projected ESWJ index-related pattern between the two sub-periods. Above results can be mutually mirrored with each other between results from the ERA-5 (Fig. 2) and those from the JRA-55 (Supplementary Fig. 9), suggesting the reliability of our findings.

We also examined the spatial patterns of U200 related to the projected ESWJ index during P1 and P2 (Supplementary Fig. 10). During P1, the ESWJ-related U200 pattern is similar to that of the WJA index, indicating a strengthening and poleward shift of WJA with relatively weak anomalies around the EJA (Supplementary Fig. 10a). During P2, the ESWJ-related U200 pattern is identical to the EOF1 during P2 (Fig. 2c); the strong and zonally consistent westerly wind anomalies along the jet axis represent strong UDZC (Supplementary Fig. 10b).

### Eurasian climate anomalies linked to strong UDZC

Increasing heatwaves have exacerbated the mortality burden on humans[4]. The heatwave frequency based on the net effective temperature (NET), which takes near-surface temperature, relative humidity, and wind speed into consideration (see "Net effective temperature and Heatwave frequency" in Methods), is introduced to better quantify the thermal comfort conditions for the human body[35]. During P1, the NET-heatwave distribution related to the projected ESWJ index exhibits a weak tripolar pattern over the Eurasian continent (Fig. 3b), especially feeble signals over high-latitudes of eastern Asia. Similar patterns are also seen over the distribution of SPEI (Standardized Precipitation Evapotranspiration Index; a drought index) and precipitation (Fig. 3c, d). As expected, the impact of ESWJ is greatly enhanced during the post-1999 period (Fig. 3e–g); the NET-heatwave distribution exhibits a widespread meridional tripolar structure: more frequent NET-heatwaves over high- and low-latitudes of Eurasia but less frequent in the mid-latitude Eurasia (Fig. 3e). Such enhancement is most apparent over East Asia, including a strong tripolar heatwave pattern over low- and high- latitudes (Fig. 3e) and tripolar SPEI pattern due to corresponding precipitation anomalies (Fig. 3f, g), consistent with the enhanced variability of the EJA index (Fig. 1b).

In addition, the similar distribution of heatwave and drought anomalies indicates a strong coupling between drought and heatwave (Fig. 3e, f), leading to compound hot-dry WCEs[6,36], which exert greater impact on the ecosystem and human health[6]. For example, drought conditions can quickly trigger a heatwave within one day, while a heatwave takes 2–7 days to induce a drought event[36]. Such drought-heatwave coupling often occurs when persistent atmospheric circulation patterns, such as the subtropical high/anticyclonic patterns,

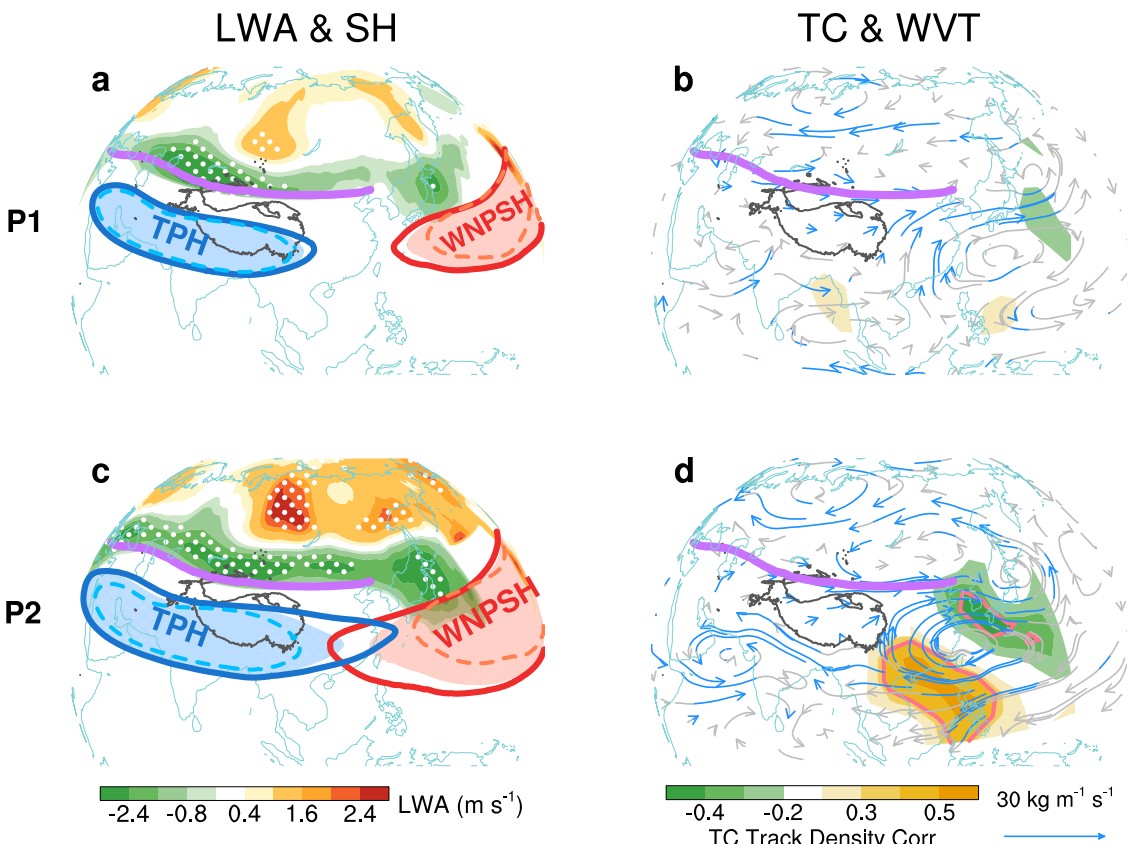

**Fig. 4 | Regime shifts linked to enhancing upstream-downstream zonal consistency in the Eurasian jet. a** Regression map of 200-hPa Local Wave Activity (LWA) against the projected Eurasian subtropical westerly jet index (ESWJ) during the first period (P1), with 95% confidence hatched. Also shown are the composite patterns of the Tibetan Plateau High (TPH; blue; measured by 200-hPa geopotential height in 12530 gpm) and the western North Pacific subtropical high (WNPSH; red; measured by 500-hPa geopotential height in 5870 gpm). Shadings represent their climatology with thick solid (dashed) lines representing their mean states in strong (weak) jet years (selected based on the threshold of 0.5 standard deviation; Supplementary Table 3; see Methods section for details), respectively. **b** Correlation map of tropical cyclone (TC) track density (shading; 95% confidence enclosed by red line) against the projected ESWJ index during P1. Vectors represent the regressed vertical-integrated water vapor flux (WVT), with 95% confidence in blue. **c–d** Same as **a–b**, but for the second period (P2). The purple line indicates the climatology of the jet axis.

create a heat dome phenomenon[7,37]. Furthermore, the atmosphere-land feedbacks are also responsible for the compound hot-dry WCEs[38,39], which have been reported to strengthen since the late-1990s[6]. Previous studies have attributed the zonal asymmetrical distribution of WCEs to atmospheric teleconnections[3,40]. Our results reveal an important role of Eurasian jet stream UDZC in the zonal symmetrical distribution of heatwaves and drought.

Figure 4 shows the corresponding mechanisms involving changes in local wave activity[19] (LWA; see "The Local Finite-Amplitude Wave Activity" in Methods), Asian-Pacific subtropical high-pressure systems, water transport over the Eurasian continent, and alterations in tropical cyclone tracks over the western North Pacific. For LWA, while a slower polar jet promotes more frequent or intense synoptic waves, a stronger subtropical jet stream acts as an enhanced mixing barrier with diminished efficiency of meridional heat transport (Fig. 4a, c). The former results in northward warm air movement and southward cold air intrusion at high latitudes[19]. The latter leads to the convergence of cold and dry (warm and moist) air over the north (south) flank of the jet stream. Therefore, such meridional heatwave distribution (Fig. 4a, c) emerges from the modified heat exchange patterns. This effect is much stronger and more zonal-symmetrical over the post-1999 period (Fig. 4a, c) with greater interannual variability in the projected ESWJ index, leading to stronger dipolar heatwave distribution north of the Eurasian jet axis (Figs. 4c and 3d).

The strengthened impact of ESWJ on low-latitude East Asia is exerted via the enhanced influence on the subtropical high-pressure

systems. In the theory of secondary circulation[21,22], the strong Eurasian jet, with stronger anti-cyclonic shear in the southern flank of its exit region, strengthens the Tibetan Plateau high-pressure (TPH) and the western North Pacific subtropical high-pressure (WNPSH). To show the changes in TPH and WNPSH related to the projected ESWJ index, we performed a composite analysis on the selected strong and weak Eurasian jet years for two sub-periods, respectively (Supplementary Table 3).

Detailed composite analyses confirm that this upper-level divergence, coupled with negative vorticity advection and ageostrophic flow at the southern flank of the jet exit, drives a closed meridional circulation cell that extends the high-pressure anomaly downward (see Supplementary Text S4 and Supplementary Figs. 11 and 12 for more details). This in turn enhances WNPSH via strong downdrafts to create a heat dome over East Asia[7,14,41]. During the strong Eurasian jet years, the eastward (westward) extension of TPH (WNPSH) is much stronger in P2 than P1 (Fig. 4a, c; Supplementary Fig. 13), leading to a stronger downdrafts and anti-cyclonic water vapor transport over East Asia (Fig. 4b, d and Supplementary Figs. 11 and 12) to the north of 30°N and thus leads to the dipole pattern of rainfall anomalies south of the jet axis.

With suppressed convection and stronger northward water vapor transport (Fig. 4b, d), the heat dome eventually leads to compound hot-dry WCEs in situ, especially the high summer of 2022[7,13,14,42]. However, exceptions occur over much of the Indian sub-continent and in the vicinity of Hainan Island (Fig. 3e). The former stems from

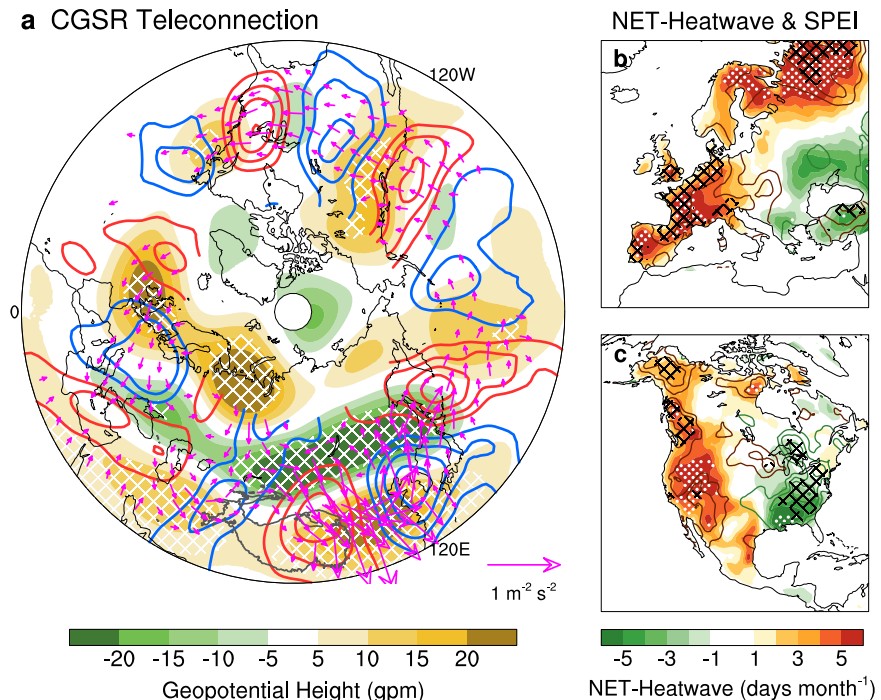

**Fig. 5 | Circumglobal impacts of the enhancing upstream-downstream zonal consistency in the Eurasian jet. a** Regression map of 200-hPa geopotential height (shading; units: gpm) and meridional wind (contour; units: m s$^{-1}$) against the projected Eurasian subtropical westerly jet index (ESWJ) during the second period (P2), with 95% confidence hatched. Vectors represent the regressed wave activity flux (WAF; units: m$^{-2}$ s$^{-2}$) at 200 hPa. **b** Regression map of heatwave frequency based on the net effective temperature (NET-heatwave; shading; unit: days month$^{-1}$) over western Europe against the projected ESWJ index during P2, with 95% confidence stippled. Contours represent the regressions of the Standardized Precipitation Evapotranspiration Index (SPEI) anomalies, with 95% confidence hatched. **c** Similar to (**b**), but for North America.

enhanced precipitation in situ (Fig. 3g) that mitigates the hot weather, and the latter is related to a dipole of tropical cyclone (TC) track density, which represents more westward TCs but less northward TCs to East Asian mid-latitude coastal regions (Fig. 4d), resulting from strengthening WNPSH and associated westward steering flow (Fig. 4c).

## Circumglobal Silk Road teleconnection and impacts on North America

As a Rossby waveguide, the subtropical jet stream is capable of triggering circumglobal teleconnection in its abnormal state[9,12,43], prompting WCEs over distant regions[5,44]. The 200-hPa geopotential height (H200) over the Eurasian sector is consistent with the NET-heatwave and drought distribution (shading in Fig. 5a), reinforcing the role of the heat dome in creating hot and dry weather/climate[7,41]. Previous studies mainly focus on the wave trains with a predominant zonal wavenumber 5 and 7 structures, which are related to the well-known circumglobal wave train[45] and responsible for the increased jet stream waviness[46,47], the Rossby wave resonance phenomenon[47], and synchronized low yields globally[5].

However, the 200-hPa meridional wind pattern clearly depicts a well-structured circumglobal wave train along the subtropical jet with wavenumber 6 (contour in Fig. 5a), resembling the Silk Road pattern[48] over the Eurasian sector but with a phase difference of one-quarter wavelength. Here, we term this wave train the circumglobal Silk Road (CGSR) teleconnection. As the Wave Activity Flux reveals, this wave train originates from the eastern North Atlantic and western Europe, propagates along the westerly jet, and terminates in the western North Atlantic (vector in Fig. 5a).

As expected, the CGSR teleconnection also takes WCEs over western Europe and North America (Fig. 5b, c). Over western Europe, where the CGSR originates, the overlapped anomalies of upper-level high-pressure, heatwave frequency, and SPEI strongly suggest the role of heat dome, which favors more heatwaves and severe drought events in situ (Fig. 5b) via strengthened incoming solar radiation, enhanced evaporation but less precipitation[14,41]. A similar heat dome is also observed in 2022, as a result of the northward shift of the North Atlantic jet, which leads to a record-breaking hot July in western Europe[37]. As the CGSR wavetrain can propagate downstream, it further exerts impacts on North America (Fig. 5c and Supplementary Fig. 14). The wave train triggers an apparent heat dome over western North America, with a feeble low-pressure anomaly over the eastern part (shading in Fig. 5a). The zonal dipole pattern is strongly related to more NET-heatwave occurrences in situ, with a relatively weak center of less NET-heatwave occurrences over the low-latitude of eastern North America (shadings in Fig. 5c and Supplementary Fig. 14a).

A similar dipolar pattern of drought conditions is revealed by the SPEI but with a much weaker signal over western North America (contours in Fig. 5c and Supplementary Fig. 14b). There is significantly more rainfall over the low-latitudes of eastern North America, corresponding to fewer NET-heatwave occurrences and mitigated drought conditions (Supplementary Fig. 14c). Compared to the pre-1998 period, during which there were only weak anomalies over the mid-latitudes of North America (Supplementary Fig. 14d–f), the Eurasian jet with stronger ESWJ variability clearly exhibits an enhanced impact on downstream regions.

The high pressure around the eastern North Atlantic and western Europe, which also resides in the exit region of the North Atlantic jet, is the key center from which the CGSR wave train emanates[30] (shading in Fig. 5a). The excitation of the Rossby wavetrain may involve barotropic instability[49]. In high summer, the air flow over this region is barotropically unstable because of the strong zonal shear of the basic zonal wind[50]. Therefore, if a disturbance is excited by certain forcings like the sea surface temperature (SST) anomalies, it could develop readily

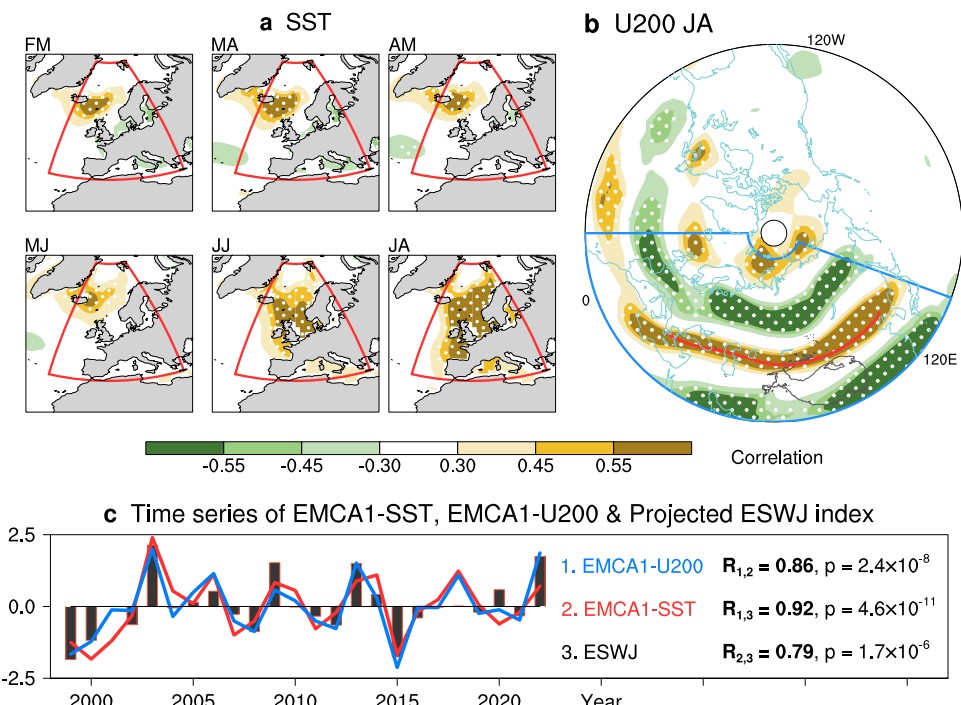

**Fig. 6 | The first pair of modes of the extended maximum covariance analysis (EMCA1) between Northeast Atlantic sea surface temperature (SST) evolution and Eurasian 200-hPa zonal wind (U200). a, b** The homogeneous correlation maps against the temporal coefficients corresponding to the EMCA1, with 95% confidence stippled. **a** The EMCA1-SST modes from early spring (February–March; FM) to high summer (July–August; JA), with red boxes denoting the region for performing EMCA. **b** The EMCA1-U200 mode during JA, with the blue box denoting the region for performing EMCA and the red line indicating the Eurasian jet axis. **c** Time series of EMCA1-SST (red) and EMCA1-U200 (blue), with the projected Eurasian subtropical westerly jet index (ESWJ; black bar).

through gaining barotropic energy from the basic flow and propagate eastward[50].

Given the memory effect of ocean[51,52], if the CGSR is triggered by the eastern North Atlantic SST, the preceding SST signals from several months in advance shall be observed. Therefore, to capture this entire evolutionary process, we employ the Extended Maximum Covariance Analysis (EMCA) method[53,54] to examine the coupled evolution patterns of Northeast Atlantic SST from spring onward to the subsequent summer U200 over Eurasia during P2 (Fig. 6). For the two fields analyzed here, the SST mode with persistent signals from the preceding February–March (Fig. 6a) can be physically interpreted as the oceanic forcing, and the U200 mode as the atmospheric response.

The first pair of EMCA modes (EMCA1) clearly reveals the coupled mode between the Northeast Atlantic SST and UDZC-like U200. The Northeast Atlantic SST warming exhibits an apparent persistent evolution since the previous February (Fig. 6a). Correspondingly, significant westerly wind anomalies are observed along the Eurasian jet axis, with easterly wind anomalies at high- and low- latitudes (Fig. 6b). The correlation between the EMCA1-U200 and the projected ESWJ index is robust ($R = 0.94$, $p = 5 \times 10^{-13}$; Fig. 6c). Moreover, the EMCA-U200 pattern is highly similar to the EOF1 mode during P2 (Fig. 2c; pattern correlation reaches 0.97). Therefore, results from EMCA strongly suggest the potential causality between the Northeast Atlantic SST anomalies and the strong UDZC of the Eurasian jet.

### Model performance and future projection of UDZC

Whether such a phenomenon can be accurately reproduced by the state-of-the-art climate models is crucial for reliable future projections. We primarily utilize outputs from two projects (see "Model simulation data" in Methods): the Coupled Model Intercomparison Project phase 6 (CMIP6) and the CESM2 Large Ensemble Community Project

(LENS2). From the aspect of multi-model mean, results from both CMIP6 and CESM2 simulations show very weak UDZC in the Eurasian jet during both historical and future periods ($R < 0.30$; Supplementary Fig. 15). The weak UDZC in models also supports our finding that a strong UDZC state is not a common phenomenon, emphasizing the uniqueness of the emergence of strong UDZC since the late-1990s.

The CMIP6 models' failure to reproduce the UDZC surge is not surprising given the diverse physical processes and projected warming amplitudes among the models. In addition to model bias in the jet stream climatology[55–57], current models' capabilities in capturing the jet stream variation mainly focus on its meridional displacement[56,58], but do not adequately capture the variation in jet stream intensity[58], which significantly affects Rossby wave behaviors[24] and contributes to the poor performance of UDZC in Eurasian jet intensity.

Therefore, models need to be evaluated before being selected for a better projection of future UDZC. To explore whether models can simulate the evolution of UDZC in observations, models are selected based on the criteria that the correlation coefficient between WJA and EJA is significant ($R > 0.24$; $p < 0.05$) during 2000–2070 and is higher than that during 1900–1970 (bars with gray background in Supplementary Fig. 15). For CMIP6 models, 18 models were selected as they show a clear strengthening trend of UDZC (green line and shading in Fig. 7a, left), but failed to simulate the rapidly increasing UDZC since the late-1970s. However, 11 members from LENS2 show an evolution similar to the observations, especially the surge after the late-1970s (purple and red lines with shadings in Fig. 7a, left), which is more pronounced than in CMIP6 (Fig. 7a, right). Therefore, based on these LENS2 members, a strong UDZC in the Eurasian jet is anticipated in the next near half century under anthropogenic warming.

Accordingly, the contribution of global warming to the Eurasian jet UDZC surge is further assessed based on its trend during the

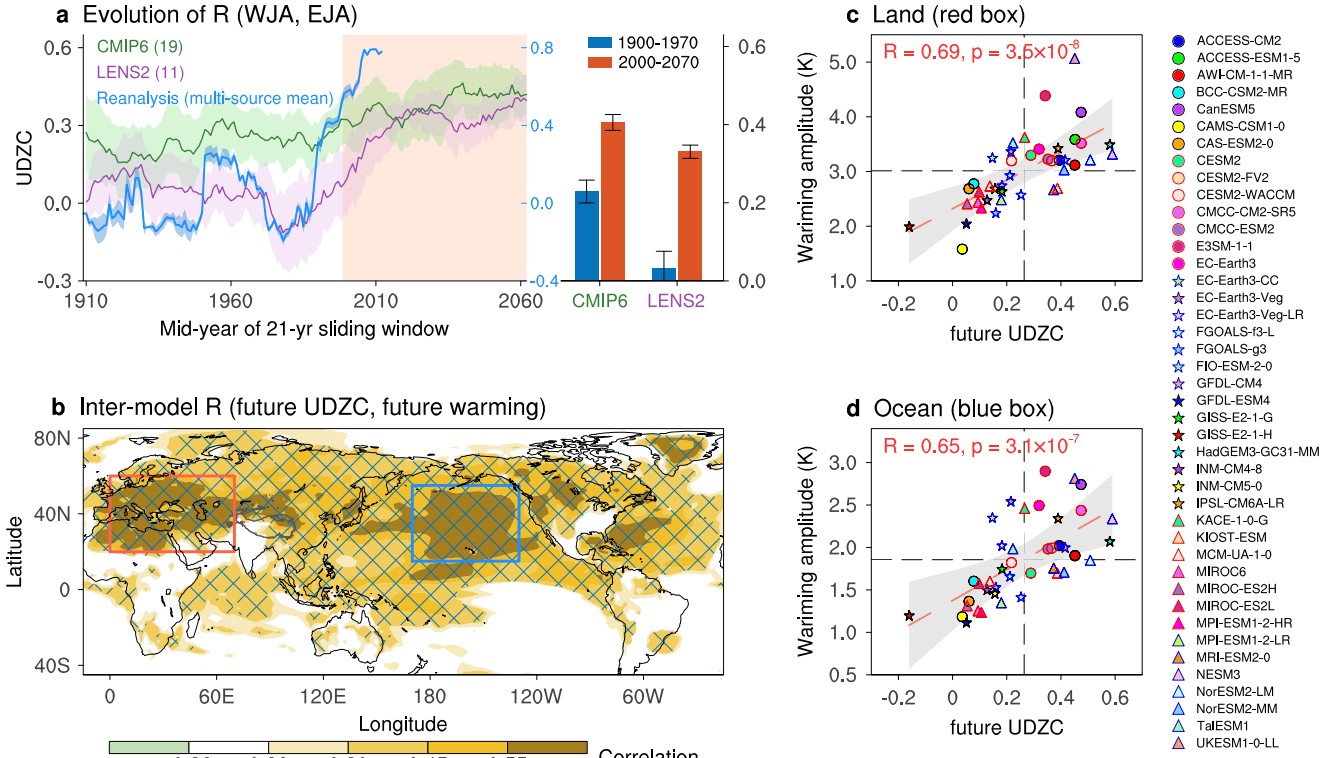

**Fig. 7 | Future changes in upstream-downstream zonal consistency (UDZC) of Eurasian jet. a** On the left is the evolution of correlation coefficient (R) between the western jet axis (WJA) and eastern jet axis index (EJA) in a 21-year sliding window, including results from the multi-source reanalysis data-based observation mean (blue) and from the selected models from Coupled Model Intercomparison Project phase 6 (CMIP6; green) and members from the CESM2 Large Ensemble Community Project (LENS2; purple) according to Supplementary Fig. 15. This figure focuses on the evolution of interannual relationship between WJA and EJA in the context of historical and future warming scenarios. The solid green/purple lines and shading indicate multi-model mean and inter-model spread, respectively. On the right is the multi-model mean of R (WJA, EJA) during the historical period (1900–1970) and the future period (2000–2070), with error bars showing the 95% confidence interval. **b** Map of inter-model correlation between future surface warming amplitude and future UDZC of Eurasian jet (see "Model simulation data" in Methods for details), with statistically significant values after controlling for the false discovery rate ($\alpha_{FDR} = 0.10$) hatched. Red and blue boxes outline the two hotspots of future warming regions over the represented land (20°N–60°N, 0–70°E) and ocean (15°N–55°N, 170°E –130°W), respectively. **c**, **d** Inter-model relationship of future UDZC and future surface warming over the hotspot of land and ocean. The thick red dashed line represents the ordinary least-squares fit, with shading indicating the 95% confidence interval.

satellite era (1979–2014) and past century (1915–2014) (Supplementary Fig. 16). Observations reveal a rapid increase in UDZC (0.31 decade⁻¹) during the satellite era, five times faster than the past-century's rate (0.06 decade⁻¹). As expected, the selected LENS2 members show a relatively larger trend (0.119 ± 0.032 decade⁻¹), accounting for 39% of the observed trend. The selected CMIP6 models show relatively similar weak trends during both periods (0.024 ± 0.019 and 0.018 ± 0.007 decade⁻¹), confirming that they only simulated the long-term trend.

## Discussion

This study reveals a critical yet previously overlooked aspect of Eurasian jet stream variability: namely, the variation in its UDZC. Our primary innovations are listed as follows: (1) conceptualizing the UDZC of the Eurasian jet and quantifying its state through the correlation between WJA and EJA, (2) revealing its rapid intensification since the 1970s (bottom panel of Fig. 1c), and (3) demonstrating this enhancement to be an exceptional phenomenon over the past two centuries using the 20thCRV3 (green curve with shading in top panel of Fig. 1c), which shows consistently weak and statistically insignificant upstream-downstream correlations before the satellite era, although accounting for higher uncertainty in the pre-1950 period.

This finding directly addresses a significant knowledge gap concerning the recent evolution of jet stream coherence over continental scales, moving beyond traditional analyses of meridional shifts or

overall weakening/strengthening trends[25,26]. We demonstrate that this surge in UDZC is related to a distinct Eurasian upper-level circulation, where the dominant mode of variability has transitioned towards synchronized jet intensity changes across the continent.

Furthermore, our research innovatively links this enhanced UDZC to a more zonally uniform and synchronized pattern of heatwave and drought occurrences across Eurasia. A key finding is the identification of the CGSR teleconnection, a wavenumber-6 Rossby wave train originating from the eastern North Atlantic, as the mechanism through which the influence of Eurasian jet UDZC extends to remote regions like North America. This provides a fresh insight into hemispheric-scale climate linkages mediated by the jet stream. Collectively, these results underscore the profound and previously underestimated impact of a zonally coherent jet stream in triggering synchronized WCEs across the Northern Hemisphere, with significant implications for ecosystems[1], human health[4], and global food security[5].

Although the strong UDZC phenomenon in recent decades is confirmed by the consistent strengthening of UDZC in multiple reanalysis data, different data selection may also lead to uncertainty in UDZC. Therefore, we calculated the inter-dataset uncertainty via the bootstrap method (shown as red shading in Fig. 7a). The shading is very small in the recent two decades due to the high consistency between modern datasets (e.g., the ERA-5, JRA-55, CFSR, and NCEP/DOE-R2). In fact, since the satellite era, and even since the late 1950s, the majority of uncertainty of dataset selection comes from two 20th

century products (e.g., the 20thCRv3 and CERA-20C). Therefore, the strong UDZC since the late 1990s remains robust between a number of modern datasets.

Although previous studies have revealed the roles of ENSO[59,60] and NAO[61,62] in the Silk Road teleconnection, the CGSR proposed here is relatively independent of ENSO and NAO (Supplementary Fig. 17; Supplementary Text 2; Supplementary Table 4). Role of decadal variations like the AMO and IPO in modulating the decadal UDZC are also explored (Supplementary Fig. 18). Although the AMO and IPO both exhibit a phase transition in the late-1990s, their correlation with decadal UDZC is not strong (Supplementary Fig. 18a). However, from the aspect of evolution tendency, the positive trend of UDZC tends to appear when the AMO shows a positive trend and the IPO shows a negative trend (Supplementary Fig. 18b). In other words, the UDZC tends to strengthen as the AMO evolves to its positive phase while the IPO evolves to its negative phase. This suggests that the different trends of the AMO and IPO may significantly modulate the UDZC of the Eurasian jet stream.

State-of-the-art models generally show limited capability in simulating the strong UDZC. As shown in Fig. 7a, the CMIP6 and LENS2 can generally capture the increasing trend of R (WJA, EJA) under global warming, albeit relatively weaker than that from Reanalysis data since about the year 2000. Ensemble means of both CMIP6 and LENS2 do not reach the consensus of UDZC future projection (Supplementary Fig. 15). Although part of LENS2 models successfully reproduce the surge of UDZC since the satellite era and suggest a future with strong UDZC, the UDZC and its trend are relatively weaker than that in the observations (Fig. 7a and Supplementary Fig. 15).

The CMIP6 and LENS2 models' failure to reproduce the UDZC evolution can be attributed to the following reasons: (1) the physical processes and projected warming amplitudes among the models are very different (Fig. 7c, d). (2) In addition to model bias in the jet stream climatology[55–57], current models' capabilities are also poor in capturing the variation in jet stream intensity[56,58], which leads to the poor performance of UDZC in Eurasian jet intensity. (3) A previous study has shown that climate models do not capture well the significant increase in jet intensity variability under global warming[28]. (4) CMIP6 models display insufficient capability in capturing the coupling between the Northeast Atlantic SST and the Eurasian jet.

We have performed EMCA similar to Fig.6 in each of the 42 CMIP6 models during 1979–2014 and found that only ten out of 42 models can generally simulate the coupling between Northeast Atlantic SST and the Eurasian jet, albeit the simulated relationship is relatively weaker than observation (Supplementary Fig. 19). Therefore, the future projection of UDZC may contain uncertainty induced from the poor simulation capability of CMIP6 and LENS2 models, which should be interpreted with caution. In addition, future CMIP developments need not only to enhance the simulation fidelity of key circulation systems like the UDZC, but also to strengthen the characterization of the key physical mechanisms that drive its variability, especially the North Atlantic air-sea coupling process[63,64].

However, the substantial inter-model variability of CMIP6 allows for an assessment of the relationship between warming amplitude and future UDZC (Fig. 7b). Therefore, we performed an inter-model correlation analysis among all 42 CMIP6 models. Notably, the future UDZC strongly correlates with the Northern Hemisphere warming amplitude. Key warming hotspots include mid-latitude Eurasia and North America (red box) and the North Pacific and subtropical western Atlantic (blue box). Land areas exhibit greater inter-model warming discrepancies (1.5–5.0 K) than ocean areas (1–2.5 K) due to the ocean heat capacity differences. Both land and ocean warming show significant linear inter-model relationships with future UDZC ($p < 0.001$, $R = 0.69$ for land and 0.65 for ocean; Fig. 7c, d). It is worth noting that results from the CESM2 model are close to the multi-model mean (green dots with red border), implying its appropriate projection of the future UDZC.

The outputs of LENS2 members do not exhibit a similar relationship between future UDZC and Northern Hemisphere warming. This is because that the outputs of LENS2 are all from the same model but with different initial conditions, leading to only small differences in warming amplitude (Supplementary Fig. 20). Interestingly, similar warming pattern is also seen in the observations (1999–2022 minus 1979–1998; Supplementary Fig. 21). These results suggest a potential role of anthropogenic global warming: the more rapid the Northern Hemisphere warming, the stronger the UDZC manifests in the Eurasian jet variation. Therefore, CMIP6 models' simulations suggest that the strong UDZC may intensify as mid-latitude warming becomes more pronounced.

## Methods

### WJA and EJA indices

The WJA and EJA indices are defined as the mean of 200-hPa zonal wind (U200) anomalies along the Eurasian westerly jet axis spanning longitude 40°E–80°E and 90°E–123°E, respectively. This axis-following approach is designed to minimize the potential bias caused by the latitudinal shift of the jet stream in different years, offering an advantage over methods that use a fixed geographical area for averaging. There are two selection criteria for the longitude range are: (1) WJA and EJA are selected at the western (i.e., 40°E–80°E) and eastern (i.e., 90°E–123°E) parts of the Eurasian westerly jet axis, respectively; (2) WJA and EJA are separated by 10° of longitude (i.e., 80°E–90°E) where is determined to be in the overlap of the maximum position of the climatological U200 and the minimum position of the U200 standard deviation of the jet stream (Fig. 1a). Note that the main results are not sensitive to the domain selection of WJA and EJA indices (Supplementary Table 1).

### Quantification of the UDZC

To straightforwardly characterize the changes in UDZC of the Eurasian westerly jet stream, here we quantify the UDZC phenomenon as the Pearson correlation (R) between WJA and EJA (e.g., Fig. 1c). Then, we can simply divide the main study period into weak and strong UDZC periods (i.e., P1 and P2), respectively. If the WJA-EJA covariance (i.e., $R^2$) of a period exceeds 30%, that period is identified as a strong UDZC period, and vice versa. Accordingly, P1 is the weak UDZC period (i.e., 1958–1998), whereas P2 is the strong UDZC period (i.e., 1999–2022).

### Projected ESWJ index

Considering that robust UDZC phenomenon between WJA and EJA emerges only in recent decades, we can only obtain the interannual variability of this phenomenon decades ago via projecting the U200 anomaly field onto Fig. 2c. In other words, a uniformly projected ESWJ index is therefore needed to describing interannual variability of UDZC phenomenon especially for pre-1999 periods, when EOF cannot well capture such UDZC-like mode as shown in Fig. 2c. Accordingly, we project the U200 anomaly field across the entire study period onto the EOF1 mode during P2 (regression pattern as contours shown in Fig. 2c) over the EOF region (i.e., blue box in Fig. 2c). Thus, the projected ESWJ index is actually completely consistent with PC1 during the P2 period: A strong positive (negative) value reflects the degree of increased (decreased) consistency between the upstream and downstream sections of the Eurasian jet stream.

### Observation-based data

We mainly use the monthly and/or daily European Centre for Medium-Range Weather Forecasts' (ECMWF's) fifth-generation reanalysis (ERA-5)[65] data at a high resolution of 0.25° × 0.25° in this study. Other multi-source monthly mean atmospheric datasets are also used to verify the robustness of our results, including the Japanese 55-year reanalysis (JRA-55)[66] on a 1.25° × 1.25° horizontal grid, ERA-Interim[67] Reanalysis with the spatial resolution of 1° × 1°, ERA-40[68] with spatial resolution of

1.125° × 1.125°, 80-member ensemble of the NOAA–CIRES Twentieth Century Reanalysis V3 (20thCRV3)[69] covering 1806–2015 with resolution of 1° × 1°, the ECMWF twentieth century reanalysis (ERA-20C)[70] and 10-member ensemble of coupled climate reanalysis of the 20th century (CERA-20C)[71] covering 1901–2010 with resolution of 1° × 1°, the National Centers for Environmental Prediction-National Center for Atmospheric Research (NCEP-NCAR)[72] Reanalysis 1 data on a 2.5° × 2.5° grid, the National Centers for Environmental Prediction/Department of Energy (NCEP/DOE)[73] Reanalysis 2 data on a 2.5° × 2.5° grid and the NCEP Climate Forecast System Reanalysis (CFSR[74,75]) with the spatial resolution of 0.5° × 0.5°.

Note that the "ERA-20C & ERA-40" in Fig. 1c represents a combination of ERA-20C (1901–1957) and ERA-40 (1958–2002): Namely, these two datasets were merged to form a continuous record spanning the period 1901–2002. The multi-source mean of R (WJA, EJA) shown as a red curve in Fig. 7a is calculated by averaging it from all available data for each 21-year sliding window. For example, the value of R (WJA, EJA) centered at 1920 is obtained by averaging results from 20thCRV3, CERA-20C, and ERA-20C, while that in 1980 is obtained by averaging results from all datasets (20thCRV3, CERA-20C, ERA-40, ERA-5, JRA-55, ERA-Interim, CFSR, NCEP/NCAR-R1, and NCEP/DOE-R2).

We employ the Standardized Precipitation Evapotranspiration Index (SPEI) to evaluate drought conditions across Eurasia. The monthly SPEI data are obtained from the Global SPEI Database[76] on a 1-month timescale with a spatial resolution of 0.25° × 0.25°, and the daily SPEI data[77] is used to identify extreme hot and dry events, with a spatial resolution of 0.5° × 0.5° and time coverage from 1982 to 2021. The SPEI is calculated using monthly precipitation and temperature records from the CRU TS4.04 data, incorporating the relationship between temperature fluctuations and drought severity[76].

Specifically, a simple climatic water balance is determined by calculating the monthly difference ($D_i$) between precipitation and potential evapotranspiration (using monthly mean temperature[78]). Next, the calculated $D_i$ values are aggregated at 1-month timescale, similar to the procedure for the Standardized Precipitation Index[79]. A probability distribution is then fitted to the aggregated $D_i$. Since the $D_i$ contains negative values, the log-logistic distribution is then fitted to the aggregated $D_i$, with parameters estimated using the L-moment procedure[80]. Finally, the SPEI is obtained as the standardized value of the cumulative distribution function of the aggregated $D_i$. The positive value represents a wet condition, and the negative value represents a drought condition.

For precipitation, two data are used: the National Oceanic and Atmospheric Administration's Precipitation Reconstruction over Land (NOAA Land)[81] at a 1° spatial resolution and the Global Precipitation Climatology Project (GPCP)[82] on a 2.5° × 2.5° grid. Note that in Figs. 3d and 3g, the rainfall data over land and ocean adopted the NOAA Land and GPCP precipitation, respectively. So, in Fig. 3d, the regressed rainfall anomalies from GPCP over ocean are calculated during 1979–1998. The surface air and seawater temperature data are directly extracted from ERA-5. Tropical cyclone data are extracted from the Joint Typhoon Warning Center (JTWC) best-track data.

### Model simulation data

The Coupled Model Intercomparison Project phase 6 (CMIP6)[83] is used to exam its ability in simulating the Eurasian jet zonal consistency in the present climate and future change. It simulates the historical climate from 1850 to 2014 and the future climate from 2015 under different scenarios. We select the historical period (1900–1970) and the future period (2000–2070) in SSP5-8.5 (Shared Socioeconomic Pathways 5-8.5) scenario, with 42 models (Supplementary Table 5). The significance test on changes of zonal consistency between historical and future periods is conducted using the bootstrap method. In addition, the CESM2 Large Ensemble Community Project (LENS2) is used to focus on the internal variability, using 50 members at 1° spatial

resolution. We also select the present period (1900–1970) and the future period (2000–2070) in SSP3-7.0 (Shared Socioeconomic Pathways 3–7.0) scenario.

In Fig. 7b, the inter-model R (future UDZC, future warming amplitude) reflects the inter-model relationship among 42 CMIP6 models. Specifically, we first calculate the future UDZC and future warming amplitude in each model: here, the future UDZC is the correlation between WJA and EJA during 2000–2070, whereas the future warming amplitude is calculated as the climatology changes in surface air temperature during the future period (2000–2070) relative to the historical period (1900–1970). And thereby we can obtain 42 pairs of future UDZC and future warming amplitude for each grid. Then, the R (future UDZC, future warming amplitude) is calculated as the Pearson correlation coefficient between the two among 42 models. The significance of the inter-model correlation is indicated with hatching, after controlling for the false discovery rate ($\alpha_{FDR} = 0.10$).

### EOF, EMCA, correlation, regression, and composite analysis

The empirical orthogonal function (EOF) analysis is performed to extract the leading modes of the Eurasian jet. It is used to obtain the leading modes of U200 anomalies over the Eurasia continent (20°–65°N, 0°–150°E). In this study, we use the EOFs and principal components (PCs) to represent the spatial patterns and temporal series, respectively. The statistical separability of the EOF modes is tested using North's rule of thumb[84].

The maximum covariance analysis (MCA)[85,86] is a commonly used tool to obtain coupling modes between two fields, also called the singular value decomposition. It operates on the covariance matrix between two fields and provides pairs of spatial patterns (modes) whose temporal covariance is high. The MCA modes are ordered according to the amount of squared covariance explained. In this study, we used the extended MCA (EMCA)[53,54] to obtain the dominant coupled modes between the evolution of Northeast Atlantic SST (35°–80°N, 20°W–25°E) from February to August and U200 (25°–80°N, 20°W–140°E) over the Eurasian continent during high summer. For the two fields here, the SST mode can be physically interpreted as the oceanic forcing, and the U200 mode as the atmospheric response.

In this study, the statistical significance of linear regression and Pearson correlation is evaluated by a two-tailed Student's t-test. The degrees of freedom for P1(1958–1998), P2(1999–2022) are 39 and 22 respectively. For the composite pattern of the Tibetan Plateau High (TPH) and the western North Pacific subtropical high (WNPSH), strong (weak) Eurasian jet years are selected as the year whose corresponding value of the projected ESWJ index is above +0.5 (below −0.5). This threshold was chosen to ensure a sufficient sample size for a statistically robust composite analysis, particularly for the shorter P2 (1999–2022) period. During P1, there are 14 strong Eurasian jet years and 14 weak Eurasian jet years. During the P2, there are six strong Eurasian jet years and six weak Eurasian jet years (See Supplementary Table 3 for details). A sensitivity analysis using a stricter threshold of ±1.0 standard deviation confirmed that the main conclusions of this study are robust (Supplementary Fig. 13).

In addition, for the correlation filed in Fig. 2 and Fig. 7, the Benjamini–Hochberg false discovery rate (FDR) correction for P-values is used to mitigate the increase in false positives from multiple testing[87]. We set the FDR control level at $\alpha_{FDR} = 0.10$ to maintain a global α level of 0.05[88]. The $P_{FDR}$ was estimated as follows:

$$P_{FDR} = \max_{j=1,\ldots,k}[P_j : P_j \leq (j/N)\alpha_{FDR}] \tag{1}$$

where $P_j$ represents the j-th P-value after sorting all N local test P-values in ascending order. Grid cells with local test P-values less than $P_{FDR}$ are considered significant.

## Identification of the ESWJ axis during high summer

This study focuses on the high summer to minimize potential latitude bias of the Eurasian jet between early and high summer[28]. Unlike the traditional method, which obtains the jet stream index with area-averaging over a fixed area, we calculate the jet stream axis index by averaging the 200-hPa zonal wind along the jet axis. Such choice of jet stream index is mainly due to the variation of jet axis latitude, which can be influenced by planetary-scale Rossby wave propagation and evolution[89], as well as thermal and dynamical interactions between polar and mid-to-low latitudes[56,90]. Therefore, the latitude of jet stream axis actually varies from year to year.

To define the ESWJ axis, we first identify, at each longitude within a latitudinal band (25°N–60°N), the latitude of the maximum 200-hPa zonal wind speed. The line connecting these points of maximum wind speed across the longitudes constitutes the ESWJ axis, which is determined in every specific year. Therefore, this method is capable of capturing the year-to-year variation in the core position of the jet stream. This axis-following approach is designed to minimize the potential bias caused by the latitudinal shift of the jet stream in different years, offering an advantage over methods that use a fixed geographical area for averaging.

## Net effective temperature and Heatwave frequency

Instead of daily temperature, here we use the net effective temperature (NET) to better quantify the thermal comfort conditions[35] for human body, which is established as follows:

$$NET = 37 - \frac{37 - T}{0.68 - 0.0014RH + \frac{1}{1.76 + 1.4v^{0.75}}} - 0.29T(1 - 0.01RH) \quad (2)$$

where T, RH, and v are the ERA-5 daily mean near-surface temperature (°C), relative humidity (%), and wind speed (m s$^{-1}$), respectively. Therefore, the NET-heatwave frequency at one grid is defined as the number of days whose NET is above the threshold (75th percentile of the records of that calendar day at 5 days window)[6].

## The local finite-amplitude wave activity

The Local finite-amplitude wave activity (LWA) can be used as a diagnostic of anomalous weather events[91], such as atmospheric blocking. The LWA is calculated based on the ERA-5 daily reanalysis data over the period 1958–2022 and is monthly averaged. The month-mean of LWA measures the overall activity of synoptic disturbance in each month. LWA in each day is defined as

$$\widetilde{A}^{*}(\lambda, \phi, z, t) = -\frac{a}{\cos \phi} \int_{0}^{\Delta \phi} q_{e}(\lambda, \phi, \phi', z, t) \cos(\phi + \phi') d\phi' \, z > 0 \quad (3)$$

where $a$ is the earth radius, $(\lambda, \phi, z)$ represents longitude, latitude, and pressure pseudo-height. $q_{e}(\lambda, \phi, \phi', z, t)$ is the "eddy" component of the quasi-geostrophic PV.

## Rossby wave activity flux

To describe the energy propagation of the quasi-stationary Rossby waves, we calculated the horizontal component of the wave activity flux (WAF)[92] using monthly mean data. The WAF in pressure coordinates is expressed as:

$$WAF = \frac{p}{2000|\vec{U}|} \begin{cases} U\left(\psi'^{2}_{x} - \psi'\psi'_{xx}\right) + V(\psi'_{x}\psi'_{y} - \psi'\psi'_{xy}) \\ U\left(\psi'_{x}\psi'_{y} - \psi'\psi'_{xy}\right) + V\left(\psi'^{2}_{y} - \psi'\psi'_{yy}\right) \end{cases} \quad (4)$$

where $\psi'$, U, and V are the perturbation stream function of quasi-geostrophic flow the climatology of zonal and meridional winds at $p$-level, respectively.

## Bootstrap

We employ the bootstrap[93] technique to evaluate the statistical significance of alterations in the zonal consistency of the Eurasian jet as depicted by CMIP6 models and LENS2 ensemble members. This involves randomly resampling the results from models of CMIP6 (members of LENS2) to generate an ensemble of 10,000 bootstrapped realizations, with the possibility of selecting any model more than once. For each designated period, we calculate the standard deviation of these realizations. A change between two sub-periods is considered statistically significant at the 95% confidence level if the difference in the multi-model mean exceeded the combined standard deviations of the respective periods.

## Data availability

The ERA-5 data: https://cds.climate.copernicus.eu/datasets. The ERA-Interim data: https://www.ecmwf.int/en/forecasts/dataset/ecmwf-reanalysis-interim. The JRA-55 data: https://rda.ucar.edu/datasets/ds628.0/. The NCEP-NCAR Reanalysis 1 data: https://psl.noaa.gov/data/gridded/data.ncep.reanalysis.html. The NCEP/DOE Reanalysis 2 data: https://psl.noaa.gov/data/gridded/data.ncep.reanalysis2.html. The 20thCRV3 data: https://portal.nersc.gov/project/20C_Reanalysis/. The CERA-20C data: https://www.ecmwf.int/en/forecasts/dataset/coupled-reanalysis-20th-century. The ERA-20C data: https://www.ecmwf.int/en/forecasts/dataset/ecmwf-reanalysis-20th-century. The ERA-40 data: https://apps.ecmwf.int/datasets/data/era40-daily/levtype=sfc. The CFSR data: https://rda.ucar.edu/datasets/ds093.2/. The CFSV2 data: https://rda.ucar.edu/datasets/ds094.2/. The GPCP data: https://psl.noaa.gov/data/gridded/data.gpcp.html. The NOAA's Precipitation Reconstruction data over Land: https://psl.noaa.gov/data/gridded/data.precl.html. The JTWC TC best-track data: https://www.metoc.navy.mil/jtwc/jtwc.html?best-tracks. The simulation outputs of CMIP6: https://esgf-node.llnl.gov/search/cmip6/. The simulation outputs of LENS2: https://www.cesm.ucar.edu/projects/community-projects/LENS2/. The Global Monthly SPEI data: https://digital.csic.es/handle/10261/332007. The Global Daily SPEI data: https://doi.org/10.5281/zenodo.8060268. The processed data used in this study are available in the Zenodo repository https://doi.org/10.5281/zenodo.18195657.

## Code availability

Codes used for this study are provided at https://github.com/Feliks151450/Eurasian_Jet_UDZC. All base maps in this study are draw directly from the NCAR Command Language (NCL; Version 6.6.2), which is available at https://github.com/NCAR/ncl/.

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

## Acknowledgements

We thank Professor Fengfei Song at the Ocean University of China for helpful suggestion. We thank Dr. Shuai Hu at the Institute of Atmospheric Physics, Chinese Academy of Sciences for providing the ERA-

20C and CERA-20C data. This work is supported by the Guangdong Major Project of Basic and Applied Basic Research (Grant 2020B0301030004), the Southern Marine Science and Engineering Guangdong Laboratory (Zhuhai) (SML2023SP209), the National Natural Science Foundation of China (Grant Nos. 42088101), the Zhejiang University "Ocean College Seed Fund": Excellent Young Teachers Training Project (2025YQ001), the Innovation Group Project of Southern Marine Science and Engineering Guangdong Laboratory (Zhuhai) (Grant No. 311024001) and Open Foundation of State Key Laboratory of Satellite Ocean Environment Dynamics, Second Institute of Oceanography, MNR (Grant No.QNHX2331). We are grateful to the research start-up funding support of the "Top 100 Talents Plan" Project of Zhejiang University and the high-performance computing condition of the Ocean College at Zhejiang University.

## Author contributions

C.H. conceived, designed, and supervised this study. L.L. performed the visualization and analyses. L.L. and C.H. conducted the analysis and drafted the manuscript. C.H., D.C., R.W., T.L., Z.W., Q.C. and S.Y. provided comments and revised the manuscript. C.H. edited the paper with inputs from all the other coauthors. All the authors contributed to the scientific discussion of the results.

## Competing interests

The authors declare no competing interests.
