## [Transparent Peer Review file · Nature Communications]

Record-breaking emergence of upstream-downstream zonal-consistent variation in Eurasian jet axis

Corresponding Author: Dr Chundi Hu

Version 0:

Reviewer comments:

Reviewer #1

(Remarks to the Author)

This manuscript presents an innovative study on the upstream-downstream zonal consistency (UDZC) of the Eurasian jet axis on an interannual timescale, employing a simple and straightforward method. The authors further associate this phenomenon with a newly identified circumglobal wavenumber-6 Rossby wavetrain, which establishes connections among Europe, Asia, and North America in terms of climate co-variations, highlighting its significant implications for hemispheric-scale weather and climate extremes. The CMIP6 inter-model analysis indicates that stronger mid-latitude warming in the future may enhance this phenomenon. The findings are clear and well-supported, contributing to an improved understanding of recent changes in the westerly jet stream and associated large-scale teleconnections. Overall, the manuscript is well-written and well-structured. I believe it has the potential to be accepted following major revisions.

General comments

1. Heatwave intensification in Western Europe. The authors discuss how the zonal-consistent jet influences heatwave variability across Eurasia and, through global teleconnections, affects heatwave distribution in North America. However, as shown in Fig. 3, the jet change also corresponds to significant heatwave intensification over Western Europe, which is not mentioned in the manuscript. Further analysis is needed to explore the impact of the jet change on heatwaves in Europe and the underlying mechanism. It would be more appropriate to include the heatwave anomalies in Europe and North America together with the CGSRT in Figure 5.
2. Relationship between SSTs and the Eurasian jet. The authors note a significant connection between the circumglobal wavenumber-6 teleconnection pattern and sea surface temperatures (SSTs) in the eastern North Atlantic, with notable precursor SST signals. I recommend using the extended maximum covariance analysis (EMCA, Polo et al. 2008; García-Serrano et al. 2008) method to analyze the evolution of North Atlantic SSTs from spring onward and their relationship with the Eurasian jet. This analysis should be presented as a main figure (or as a supporting figure, if necessary) and discussed in detail.
3. Climate model capability. The manuscript should discuss the ability of climate models to simulate the observed coupling between eastern North Atlantic SSTs and the Eurasian jet. Specifically, can the CMIP6 models capture the coupling between North Atlantic SSTs and the Eurasian jet as identified by the MCA or EMCA method?
4. Roles of ENSO, NAO, AMO, and IPO. Previous studies suggest that the Silk Road Pattern is influenced by ENSO (Yang et al. 2022) and NAO (Hong et al. 2022). Is the wavenumber-6 circumglobal wave train proposed here related to tropical forcing or the NAO? If the ENSO signal is removed, does this impact the results? Additionally, are changes in the UDZC modulated by the Atlantic Multidecadal Oscillation (AMO) or the Interdecadal Pacific Oscillation (IPO)?

Specific comments

1. Title Improvement. The current title, "Emergence of unprecedented zonal-consistent variation in Eurasian jet axis," could be improved to better reflect the broader scope of the study.
2. Abstract (L22). In the abstract, the sentence "Besides, we further report a brand-new circumglobal wavenumber-6 wavetrain originating from Northeastern Atlantic" is not well connected to the preceding sentence. The logical flow of ideas should be improved.
3. Seasonality of UDZC (L78). Is there also a strong UDZC signal in other seasons (e.g., the winter Eurasian jet)? Why is the

UDZC signal prominent during high summer?

4. Drought-heatwave coupling (L147). The concept of "drought-heatwave coupling" requires a more detailed explanation.
5. Wording adjustment (L159). Add "more" to the sentence: "This effect is much stronger and (more) zonal-symmetrical.....".
6. Model selection criteria (L244). The selection criteria for the 18 CMIP6 models and 11 LENS2 members should be explicitly stated.
7. Clarification on model usage (L273-274). It should be clarified that all 42 models were used in the analysis, not just the selected 18 models. Do the LENS2 members also exhibit a similar relationship between future UDZC and Northern Hemisphere warming?
8. In Figure 6a, the study focuses on the interannual relationship between the western jet axis (WJA) and the eastern jet axis (EJA) in the context of... (sentence incomplete, please clarify).

(Remarks on code availability)

Reviewer #2

(Remarks to the Author)

Weather and climate extremes are strongly related to jet stream variability, but their mechanisms still unclear. Lin et al. explored the linkages between Eurasian jet using upstream and downstream zonal consistency (UDZC) and local wave activity based on the reanalysis and model data. The authors projected that a strong UDZC could cause synchronized climate extremes. The idea is promising and the results is interesting. I provide some suggestions to help improve the manuscript.

The innovation of the manuscript should be addressed currently both in the introduction and discussion, especially regarding the knowledge gaps in the jet stream.

Some parts of the result section should be rephrased for readability (see also my detailed comments).

For the spatial correlation and regression analysis (figures 2-6), the potential "false discovery" that may be raised from multiple hypothesis testing while simultaneously testing thousands of grids should be considered. Controlling the false discovery rate for multiple tests is a possible way to test the results.

The uncertainty of the two-centuries data (20th CRV3 reanalysis) should be considered, especially for the early-period and Eurasian regions.

In the results, the authors used the jet indices from U200, but for other local wave activities from different heights, such as H150, H500, H300, and V200. Is there a way to reduce the number of the heights? Or there are other ways to illustrate their connections and mechanisms?

The discussion should be strengthened to highlight this study's innovation. The discussion just focused on the model results and future changes of UDZC. I think that it is not enough to address the innovation of the whole study, such as how the Eurasian jet variability and potential mechanisms work.

Details:

L57: Could you specify "the satellite era"?

L62: Is it right? The Eurasian jet is meandering in Figure 1.

L66-67, 89-92: It is unclear for me how did you decide the point year of "1958"?

L68-71: This should be moved down and combined with the next paragraph.

L73: Could you specify the number of the sliding time window?

L110-112: I think that this should be in the introduction.

L182-185: How about the results if you used H200?

L305: SEPI? Is it SPEI?

L308-309: The reference(s) for how you calculate SPEI is needed.

L328-338: A little confusion about the analysis. The Eurasian jet is based on U200, why you calculate correlations between PCs of U200 with the Eurasian jet again? How do you link the Eurasian jet and SST?

L342-343: How many years of the strong or weak Eurasian jet are selected in the analysis? This information is unclear here and figures.

L346-348: Could you specify the jet axis? Why the jet axis vary? How about the coherent variability between this jet index

and the traditional jet stream index because you stated that the new method has some advantages?

L353-355: Why did you project the U200 from 1958-2022 onto EOF1 during 1999-2022, rather than the whole period? Is it reasonable?

Figures:

Fig 1. Which data are used in WJA and EJA calculation in panel b? Could you change the color of confidence lines as black or gray to reduce the color in panel c?

Fig 2. Could you consider the false discovery rate in the correlation and regression maps (see also my main comments)? Is it better to change the ESWJ as a line rather than a bar?

Fig 4. It will be better to provide the strong (weak) jet years in the figure or supplementary.

Fig 5. Why not select the same height for geopotential height and V200? It looks a little confusing for different heights. Panel c, please double check the significant results are colored.

Fig 6. Could you show the UDCZ variability from the observation data? This would be better to validate the changes between observation and model results as well as the history and future changes. A little confusing from the legend of panel b, it is the differences between two periods (200-2070 vs 1900-1970), but it shows a correlation map. How about the significance?

(Remarks on code availability)

Reviewer #3

(Remarks to the Author)

Review for: Emergence of unprecedented zonal-consistent variation in Eurasian jet axis

This paper introduces a new index tracking the strength of the 200-hPa zonal wind (U200) jet across '40°E–80°E for the west part of Eurasian jet axis (WJA index), 90°E–123°E for the east part (EJA index)' is defined as the 'upstream-downstream zonal consistency' (UDZC) of the jet, and then tracks its variability in several datasets - with substantial differences in the index amongst them. The motivation is to better understand Eurasian weather/climate extremes, although the UDZC index defined is not actually shown to be related to them. Shifts in the index are found throughout its record, including near the start of the satellite record, and given the location of the study as potentially sparse with data (across western, northern Asia, for instance) means that data quality could be a source of the 'profound shift in the UDZC' that is reported.

A different index, the 'Eurasian subtropical westerly jet' (ESWJ) index, is subsequently defined using the second principal component of EOF analysis performed on U200 for the period 1999-2022. It is somehow – rather uncertainly – linked to UDZC, but how is not quite clear. The ESWJ is loosely related to shifts in heatwaves, the standardized precipitation index and precipitation, but with 90% confidence that suggests a less robust relationship than hoped for. It also is unclear why the UDZC is relevant or introduced when it is not the index that is related to extremes. It is unclear what is physically meaningful about the UDZC or ESWJ indices defined, which are inferred as the 'cause' of weather and climate extremes when they are, rather, symptomatic of the conditions conducive to extremes. The jet and the eddies producing extremes coevolve, and tracking one is to track the other. As such, the statistical analysis regarding 'abrupt' but probably random shifts in these vaguely-defined quantities is likely being overinterpreted. It is the same as concluding that there are random shifts in the waves and extremes related to them.

While it is possible that there is some dynamical relevance to the UDZC, the methodology used is subject to many limitations whose mention are missing in the text, such as likely presence of spurious EOFs that are poorly resolved, thus accounting for their shifting between P1 and P2. Much more robust analysis is needed to draw a conclusion that a random shift in UDZC represents a 'new paradigm of Eurasian circulation', particularly since UDZC's dynamical relevance is not shown.

It does not seem, to me, that the results from this analysis are always reliably interpreted, such as this shift in UDZC representing a 'with a brand-new paradigm of Eurasian upper-level circulation' as stated in the Discussion. The null hypothesis that this shift is random is not robustly interrogated to draw the conclusion that the Eurasian circulation has dramatically shifted due to shifts in the indices used here. So much further information is needed to support the claims put forth that the results and significance do not seem substantial or robust enough for Nature Communications.

Major Comments

What UDZC physically represents remains elusive throughout the text. It seems that two areas of high zonal wind variance are tracked and that it happens to have had a strong difference in the year 2022. Why this is important is not, however, demonstrated by any of the present analysis. What does a positive value represent in terms of the jet, in absolute terms? A negative value? These basic pieces of information are crucial to be convinced that UDZC warrants any consideration, and the subsequent analysis about its variability over a long period of record does not seem much of interest.

Figure 1: Given limited in situ data in this region prior to the satellite era, and the lack of agreement between datasets prior to that time – and even during it for some estimates – it seems that data quality could preclude drawing robust conclusions about the multidecadal variability of this quantity. And, importantly, there is no information shown about the pattern of the jet during high or low UDZC values, which is crucial to understand what this index represents. The subsequent relation to the EOFs of u200 and their PCs, below, is therefore nebulous regarding the relation to UDCZ – which is itself unclear.

Figure 2: it is possible that the difference in pattern between period 1 and period 2 simply suggests that the variability in the region is poorly resolved with EOFs. Are the EOFs statistically distinct from one another according to North's rule (1982)? How do the EOF patterns manifest in the full-field zonal wind? I don't see any relation between the UDZC and the EOFs, in part because so little information about what the UDZC represents is provided.

G.R. North, T.L. Bell, R.F. Cahalan, and F.J. Moeng. (1982). Sampling errors in the estimation of empirical orthogonal functions. *Mon. Wea. Rev.*, 110:699-706.

Minor comments

Line 87-88 Much more evidence of the dynamical importance and uniqueness of the UDZC is needed before asserting that it is the leading feature of all Eurasia circulation, where shifts in UDZC dictate circulation shifts. "The emergence of strong UDZC in Eurasian jet implies a notable change in the dominant mode of jet stream variations" is not a statement I agree with, given the analysis presented here.

Line 332 'singular value' not 'singularly valuable'.

Line 342 "Strong (weak) Eurasian jet year are selected as the year whose corresponding value of ESWJ index is above + 0.5 (below -0.5)" this is a rather low threshold for standardized values, isn't it?

Lines 347-350 "we calculate the jet stream axis index by averaging the 200-hPa zonal wind along the jet axis, which varies from year to year" – it's not actually stated how the jet axis's location is determined from year to year in the two longitude bands that are subsequently defined. This is a key part of the methods that needs explaining. In any case I don't quite agree with the following assertion that differencing zonal wind between two locations along the jet core is 'dynamical' – perhaps if the locations were selected according to the convergence of the ageostrophic wind along the jet entrance and exit regions or something, dynamical would be a defensible description.

(Remarks on code availability)

Version 1:

Reviewer comments:

Reviewer #1

(Remarks to the Author)

I appreciate the authors for adequately addressing my previous questions and comments. I have no further questions and recommend accepting the manuscript.

(Remarks on code availability)

Reviewer #2

(Remarks to the Author)

The manuscript has been improved after the revision; however, the concerns of Reviewer 3, especially for the stability and reliability of the shift of UDZC index are not fully resolved. Some of them should be further explored thoroughly.

1. Although the authors explained the dynamical mechanism of UDZC and WNPSH, and TP high pressure, a dynamic example analysis or composite analysis should be provided in the supplementary material to improve the explanation.
2. At present, the domain of EOFs covers the whole Eurasia continent (20-65N,0-150E), and the UDZC index is based on the domain of WJA and EJA. How did you select the domains of WJA and EJA? Please clarify it. Why the domain of EOF and UDZC is different?
3. The definition of the UDZC index is arbitrary and needs to be justified. The authors just project the U200 anomalies onto the EOF1 mode during the period 1999 to present, which is defined as the interannual variability of UDZC. This definition is

arbitrary and lacks of mechanism because the EOF modes are different between the two sub-periods 1958-1998 and 1999-2022 (Figure 2). The authors consider that the EOF1 for the P1 and P2 represent different implications: EOF 1 of P1 represents the Eurasian jet meridional shift, whereas the EOF1 of P2 represents the strengthening of the whole Eurasian jet. This may further indicate that the EOF mode may strongly depend on the period we used and further affect the UDZC index. This should be further explained and explored. This is also a concern by reviewer 3 about the robustness of the shift or stability of UDZC, as well as the definition of UDZC. Although the authors argue that "this research is not to quantify the exact UDZC value, but to demonstrate the recent surge in UDZC in recent two decades" , if the UDZC is not robust, the increase of UDZC is still unreliable. The authors should pay more attention to this.

4. The UDZC index is based on the correlation between EJA and WJA, whereas the correlations are based on the sample size (time periods or slide windows), and that from different datasets are not the same. These would result in a very large uncertainty of the UDZC index, for example, the domain selections of EJA and WJA, the dataset selection, and the correlation window. The correlation between EJA and WJA is not significant in most of the periods in both Figure 1, 7, Supplementary Figure 1, 2, and 14, from a statistical perspective, the relationship between EJA and WJA is random (not significant). This may be one of the concerns of the robustness of UDZC index. Furthermore, in Figure 7, the correlations of EJA and WJA are different between the model projection and reanalysis datasets. Does this also suggest the possible unreliability of the UDZC index ?

minor comments:

1. L353-354: Which results or references are you refer to, especially for the past two centuries?

2. L582-291: This part is not necessary because the NCL software is generally used in climatology , hydrology, and other related research.

3. Figure 3: Regression maps are shown for the influences of UDZC on climate in the Euroasia during the different periods. I suggest that the position jet stream (climatology of jet stream axis) should also be shown for different periods because the climatology of the jet stream axis has been shown in Figure 2.

4. Figure 7 : Why is the correlation between the WJA and EJA (UDZC) from reanalysis much stronger than that from the CMIP6 and LENS2? Why the variability of the UDZC from reanalysis data is larger than from the CMIP and LENS2? These analyses are very important to explain the stability of the UDZC and its physical mechanism.

Minor edit in Lines: 466-467: "The significance of the " ?

(Remarks on code availability)

Version 2:

Reviewer comments:

Reviewer #2

(Remarks to the Author)

I appreciate the authors for addressing my previous concerns and comments. I have no additional questions and recommend publishing this manuscript.

(Remarks on code availability)

I appreciate the authors for addressing my previous concerns and comments. I have no additional questions and recommend publishing this manuscript.

Response to Reviewers' Comments

We thank the reviewers for their insightful comments and detailed suggestions with emphases on different aspects, which are helpful in improving the quality of manuscript. Accordingly, we have carefully revised the manuscript, taking all comments and suggestions into account in the revision.

Our point-by-point responses to these comments are summarized as follows. Meanwhile, we give the marked version (i.e., tracked-changes) of the manuscript. Note that the **Line numbers** in this response are corresponded to the **marked version** of the revised manuscript.

Response to reviewer #1:

This manuscript presents an innovative study on the upstream-downstream zonal consistency (UDZC) of the Eurasian jet axis on an interannual timescale, employing a simple and straightforward method. The authors further associate this phenomenon with a newly identified circumglobal wavenumber-6 Rossby wavetrain, which establishes connections among Europe, Asia, and North America in terms of climate co-variations, highlighting its significant implications for hemispheric-scale weather and climate extremes. The CMIP6 inter-model analysis indicates that stronger mid-latitude warming in the future may enhance this phenomenon. The findings are clear and well-supported, contributing to an improved understanding of recent changes in the westerly jet stream and associated large-scale teleconnections. Overall, the manuscript is well-written and well-structured. I believe it has the potential to be accepted following major revisions.

Response: Thank you for your positive and supportive comments and the detailed instruction on how to improve the manuscript. The quality of the manuscript has been greatly improved based on your comments. Please find our response below.

General comments

1. Heatwave intensification in Western Europe. The authors discuss how the zonal-consistent jet influences heatwave variability across Eurasia and, through global teleconnections, affects heatwave distribution in North America. However, as shown in Fig. 3, the jet change also corresponds to significant heatwave intensification over Western Europe, which is not mentioned in the manuscript. Further analysis is needed to explore the impact of the jet change on heatwaves in Europe and the underlying mechanism. It would be more appropriate to include the heatwave anomalies in Europe and North America together with the CGSRT in Figure 5.

Response: Thank you for your suggestion. We acknowledge that we did not specifically mention heatwaves in Western Europe in the manuscript. Following your advice, we have replaced main Figures 5b and 5c with heatwave anomalies in western Europe and North America (**Fig. R1**). And explored the impact of the jet change on heatwaves in Europe and North America and the underlying mechanism. We have also made the following modifications in the manuscript:

L268–280: “As the Wave Activity Flux reveals, this wave train originates from the eastern North Atlantic, propagates along the westerly jet and terminates in the western North Atlantic (vector in **Fig. 5a**).

As expected, the CGSR teleconnection also takes WCEs over western Europe and North America (**Fig. b, c**). Over the western Europe, where the CGSR originates, the overlapped anomalies of upper-level high-pressure, heatwave frequency and SPEI strongly suggests the role of heat dome which favors more heatwaves and severe drought events *in-situ* (**Fig. 5b**) via strengthened incoming solar radiation, enhanced evaporation but less precipitation^{14,41}. Similar heat dome is also observed in 2022, as a result of northward shift of North Atlantic jet, which leads to an unprecedented hot July in western Europe³⁷. As the CGSR wavetrain can propagate downstream, it further exerts impacts on the North America (**Fig. 5c** and Supplementary Figure 10).”

References

14. Shi, T. *et al.* Comparative Analysis of the 2013 and 2022 Record-Breaking Heatwaves over the Yangtze River Basin. *Ocean-Land-Atmos Res* **3**, 0071 (2024).
37. Li, X., Zheng, J., Wang, C., Lin, X. & Yao, Z. Unraveling the roles of jet streams on the unprecedented hot July in Western Europe in 2022. *npj Clim Atmos Sci* **7**, 323 (2024).
41. Wu, S. *et al.* Local mechanisms for global daytime, nighttime, and compound heatwaves. *npj Clim. Atmos. Sci.* **6**, 36 (2023).

Fig. R1 (Fig. 5 in revision) | Circumglobal impacts of strong UDZC in Eurasian jet. **a** Regression map of H200 (shading; units: gpm) and V200 (contour; units: $m s^{-1}$) against the UDZC index during P2, with 95% confidence hatched. Vectors represent the regressed wave activity flux (WAF) at 200-hPa. **b** Regression map of NET-heatwave frequency (shading; units: days month⁻¹) over western Europe against the UDZC index during P2, with 95% confidence stippled. Contours represent the regressions of SPEI anomalies, with 95% confidence hatched. **c** Similar to **b**, but for the North America.

2. Relationship between SSTs and the Eurasian jet. The authors note a significant connection between the circumglobal wavenumber-6 teleconnection pattern and sea surface temperatures (SSTs) in the eastern North Atlantic, with notable precursor SST signals. I recommend using the extended maximum covariance

analysis (EMCA, Polo et al. 2008; García-Serrano et al. 2008) method to analyze the evolution of North Atlantic SSTs from spring onward and their relationship with the Eurasian jet. This analysis should be presented as a main figure (or as a supporting figure, if necessary) and discussed in detail.

Response: Thank you for your suggestion. We have performed an EMCA between the U200 over Eurasia and the North Atlantic SST from February to August (**Fig. R2**). The results show a significant coupled mode between them. We have included this as Figure 6 and discussed it in the text as follows:

L299–314: “Given the memory effect of ocean^{51,52}, if the CGSR is triggered by the North Atlantic SST, the preceding SST signals from several months in advanced shall be observed. Therefore, to capture this entire evolutionary process, we employ the Extended Maximum Covariance Analysis (EMCA) method^{53,54} to examine the coupled evolution of North Atlantic SST starting from February and the subsequent summer U200 over Eurasian during P2 (**Fig. 6**). For the two fields analyzed here, the SST mode with persistent signals from preceding February (**Fig. 6a**) can be physically interpreted as the oceanic forcing, and the U200 mode as the atmospheric response. The first pair of EMCA modes (EMCA1) clearly reveal the coupled mode between North Atlantic SST and strong UDZC. The SST warming around the eastern North Atlantic exhibit an apparent persistent evolution since the previous February (**Fig. 6a**). Correspondingly, significant westerly wind anomalies are observed along the Eurasian jet axis, with easterly wind anomalies at high- and low- latitudes (**Fig. 6b**). The correlation between the EMCA1-U200 and UDZC index is robust ($R = 0.94$, $p = 5 \times 10^{-13}$; **Fig. 6c**). The U200 pattern is similar to the EOF1 during P2 (pattern correlation reaches 0.97). Therefore, results from EMCA strongly suggests the potential causality between the North Atlantic SST anomalies and the strong UDZC of Eurasian jet.”

References:

51. Namias, J. & Born, R. M. Temporal coherence in North Pacific sea-surface temperature patterns. *J. Geophys. Res.* **75**, 5952–5955 (1970).

52. Deser, C., Alexander, M. A. & Timlin, M. S. Understanding the persistence of sea surface temperature anomalies in midlatitudes. *J. Clim.* **16**, 57–72 (2003).
53. Polo, I., Rodríguez-Fonseca, B., Losada, T. & García-Serrano, J. Tropical Atlantic Variability Modes (1979–2002). Part I: Time-Evolving SST Modes Related to West African Rainfall. *J. Clim.* **21**, 6457–6475 (2008).
54. García-Serrano, J., Losada, T., Rodríguez-Fonseca, B. & Polo, I. Tropical Atlantic Variability Modes (1979–2002). Part II: Time-Evolving Atmospheric Circulation Related to SST-Forced Tropical Convection. *J. Clim.* **21**, 6476–6497 (2008).

EMCA1 (SST, U200)

Fig. R2 (Fig. 6 in revision) | The EMCA1 modes of co-varying North Atlantic SST evolution and Eurasian U200. a–b, the homogeneous correlation maps against the temporal coefficients corresponding to EMCA1, with 95% confidence stippled. The red line in (b) indicates the Eurasian jet axis. Red (blue) boxes denote the SST (U200) regions for EMCA. c, Time series of EMCA1-SST (red) and EMCA1-U200 (blue), with the UDZC index (black bar).

3. Climate model capability. The manuscript should discuss the ability of climate models to simulate the observed coupling between eastern North Atlantic SSTs and the Eurasian jet. Specifically, can the CMIP6 models capture the coupling between North Atlantic SSTs and the Eurasian jet as identified by the MCA or

EMCA method?

Response: Thank you for your suggestion. We have performed EMCA similar to Fig.6 in each 42 CMIP6 models during 1979–2014. There are only ten out of 42 models which successfully simulates the coupling between SST warming and strengthened Eurasian jet in the first two pairs of EMCA (**Fig. R3**). The results suggests that less than a quarter CMIP6 model is capable of capturing the coupling between North Atlantic SSTs and the Eurasian jet. We have added these in the Discussion section:

L383–393: “In addition, the CMIP6 models’ capability of capturing the coupling between North Atlantic SSTs and the Eurasian jet is also investigate. We have performed EMCA similar to **Fig.6** in each 42 CMIP6 models during 1979–2014. There are only ten out of 42 models which successfully simulates the coupling between SST warming and strengthened Eurasian jet. However, as the multi-model mean of these ten model shows (Supplementary Figure 15), the simulated relationship with North Atlantic SST during JA is much weaker than observation; similarly, the easterly wind anomalies over lower latitudes of Eurasia is very feeble. Therefore, future CMIP development need to not only enhance the simulation fidelity of key circulation systems like the UDZC, but also strengthen the characterization of the key physical mechanisms that drive its variability, especially the North Atlantic air-sea coupling process^{63,64}”

References:

63. Wu, Z., Li, J., Jiang, Z., He, J. & Zhu, X. Possible effects of the North Atlantic Oscillation on the strengthening relationship between the East Asian Summer monsoon and ENSO. *Int. J. Climatol.* **32**, 794–800 (2012).
64. Wu, Z., Wang, B., Li, J. & Jin, F.-F. An empirical seasonal prediction model of the east Asian summer monsoon using ENSO and NAO. *J. Geophys. Res.: Atmos.* **114**, D18120 (2009).

Fig. R3 (Supplementary Figure 15 in revision) | CMIP6 model mean of EMCA1 mode of co-varying North Atlantic SST evolution and Eurasian circulation (U200) in high summer. a–b, multi model mean of the homogeneous correlation maps against the temporal coefficients corresponding to EMCA1, with 90% confidence colored. The blue line in (b) indicates the Eurasian jet axis. Purple boxes denote the regions for EMCA1. Those selected 10 models include ACCESS-ESM1-5, AWI-CM-1-1-MR, CAMS-CSM1-0, CESM2, CESM2-WACCM, INM-CM5-0, IPSL-CM6A-LR, MIROC-ES2H, NorESM2-MM and UKESM1-0-LL.

4. Roles of ENSO, NAO, AMO, and IPO. Previous studies suggest that the Silk Road Pattern is influenced by ENSO (Yang et al. 2022) and NAO (Hong et al. 2022). Is the wavenumber-6 circumglobal wave train proposed here related to tropical forcing or the NAO? If the ENSO signal is removed, does this impact the results? Additionally, are changes in the UDZC modulated by the Atlantic Multidecadal Oscillation (AMO) or the Interdecadal Pacific Oscillation (IPO)?

Response: Thank you for your suggestion. We have explored the role of ENSO, NAO, AMO and IPO, the CGSR proposed here is relatively independent of ENSO and NAO (**Fig. R4** and **Table R1**). However, we found that the different trends of AMO and IPO may significantly modulate the UDZC of the Eurasian jet stream (**Fig. R5** and **R6**): UDZC tends to strengthen as AMO evolves to its positive phase while IPO evolves to its negative phase.

We have added a brief discussion in manuscript and the detailed discussion is

shown in Supplementary Information:

L369–378 in manuscript: “Although previous studies have revealed the role of ENSO^{59,60} and NAO^{61,62} in the Silk Road teleconnection, the CGSR proposed here is relatively independent of ENSO and NAO (Supplementary Figure 13; Supplementary Text 2). Role of decadal variations like AMO and IPO in modulating the decadal UDZC are also explored (Supplementary Figure 13 and 14). Although AMO and IPO both exhibit a phase transition in the late-1990s, their correlation with decadal UDZC is not strong (Supplementary Figure 14a). However, from the aspect of evolution tendency, the positive trend of UDZC tends to appear when AMO shows positive trend and IPO shows negative trend (Supplementary Figure 14b). In other words, the UDZC tends to strengthen as AMO evolves to its positive phase while IPO evolves to its negative phase. This suggests that the different trends of AMO and IPO may significantly modulate the UDZC of the Eurasian jet stream.”

References:

59. Yang, X., Huang, P., Liu, Y. & Chen, D. An interdecadal enhancement of the impact of ENSO on the summer Northeast Asia circulation around 1999/2000 through the Silk Road Pattern. *J. Clim.* 1–40 (2022)
60. Tang, S. et al. Recent changes in ENSO’s impacts on the summertime circumglobal teleconnection and mid-latitude extremes. *Nat. Commun.* **16**, 646 (2025).
61. Hong, X., Lu, R., Chen, S. & Li, S. The relationship between the North Atlantic Oscillation and the Silk Road pattern in summer. *J. Clim.* (2022)
62. Hong, X., Lu, R. & Li, S. Differences in the Silk Road Pattern and Its Relationship to the North Atlantic Oscillation between Early and Late Summers. *J. Clim.* **31**, 9283–9292 (2018).

Supplementary Text S2: “Previous studies reveals that the Silk Road Pattern / Circumglobal pattern is modulated by ENSO^{1,2} and NAO^{3,4}. To investigate the role of ENSO and NAO, we firstly defined a CGSR index based on the V200 results in **Fig. 5a**. The wave train corresponding to this index is consistent with that shown in **Fig 5a**, with stronger amplitude and more pronounced propagation path (Supplementary Figure 13a, b). The correlation coefficient between the

CGSR index and the UDZC index reaches 0.75 ($p = 1.2 \times 10^{-5}$; Supplementary Figure 13e), indicating a strong relationship between the CGSR and strong UDZC. This signal does not show a significant linear relationship with either NAO or tropical precipitation forcing (Supplementary Figure 13c, d).

We further investigated the relationship between the UDZC index and CGSR with ENSO, and found that neither concurrent ENSO nor preceding winter ENSO shows a significant relationship with UDZC and CGSR (Supplementary Table S1).

From the aspect of decadal variability, although the reported strong UDZC events mainly occurred after the late 1990s, which coincides with the phase transition times of the IPO and AMO (Supplementary Figure 14a), the interdecadal evolution of UDZC state (21-year sliding correlation between WJA and EJA indices), shows a significant but relatively weak relationship with the AMO index ($R = 0.29$, $p = 3.6 \times 10^{-4}$), and non-significant relationship with IPO index ($R = -0.10$, $p = 0.22$). However, from the aspect of evolution tendency, it can be observed that the rapid increase in the UDZC since 1979 also coincides with the positive trend of the AMO and the negative trend of the IPO (Supplementary Figure 14b). This suggests that the different trends of AMO and IPO may significantly modulate the UDZC of the Eurasian jet stream.”

References:

1. Yang, X., Huang, P., Liu, Y. & Chen, D. An interdecadal enhancement of the impact of ENSO on the summer Northeast Asia circulation around 1999/2000 through the Silk Road Pattern. *J. Clim.* 1–40 (2022)
2. Tang, S. et al. Recent changes in ENSO’s impacts on the summertime circumglobal teleconnection and mid-latitude extremes. *Nat. Commun.* 16, 646 (2025).
3. Hong, X., Lu, R., Chen, S. & Li, S. The relationship between the North Atlantic Oscillation and the Silk Road pattern in summer. *J. Clim.* (2022)
4. Hong, X., Lu, R. & Li, S. Differences in the Silk Road Pattern and Its Relationship to the North Atlantic Oscillation between Early and Late Summers. *J. Clim.* 31, 9283–9292 (2018).

Fig. R4 (Supplementary Figure 13 in revision) | CGSR index and related atmospheric anomalies. **a–b**, regression map of V200 (units: m s^{-1}) against standardized UDZC index (**a**) and CGSR index (**b**), with vectors represent regressed WAF ($\text{m}^{-2} \text{s}^{-2}$). Green boxes in (**a**) outlines the definition regions to defining CGSR index. **c–d**, regression maps of SLP (**c**; units: Pa; contour interval: 40 Pa) and precipitation (**d**; units: mm day^{-1}) against standardized CGSR index, with shadings represents 95% confidence. **e**, year-to-year variation of standardized CGSR index (black bar) with UDZC index (red curve).

Fig. R5 (Supplementary Figure 14 in revision) | Modulation of AMO and IPO on UDZC. a, The 21-year sliding correlation of WJA and EJA indices (red curve), with AMO (green) and IPO (blue) index. **b,** similar to a, but their 21-yr sliding trend.

Table R1 (Supplementary Table 1 in revision) | Correlation coefficients between different ENSO index UDZC and CGSR index.

	Nino3 (JF)	Nino4 (JF)	Nino3.4 (JF)	Nino3 (JA)	Nino4 (JA)	Nino3.4 (JA)
UDZC index	0.04	0.22	0.13	-0.17	0.04	-0.09
CGSR index	-0.01	0.05	0.01	0.02	0.04	0.13

Specific comments

1. Title Improvement. The current title, "Emergence of unprecedented zonal-consistent variation in Eurasian jet axis," could be improved to better reflect the broader scope of the study.

Response: Thank you for your thoughtful suggestion regarding the title. We have carefully considered it. While we agree our findings have broader implications, we decided to retain the original title for two main reasons. Firstly, we are constrained by the journal's character limit, which makes it challenging to

incorporate these wider aspects. Secondly, we believe a title should be a concise summary of the core investigation, as it is difficult for any title to be all-encompassing. We feel our current title most accurately and clearly reflects the central focus of our analysis. Therefore, after careful discussion, we have decided to retain the original title.

2. Abstract (L22). In the abstract, the sentence "Besides, we further report a brand-new circumglobal wavenumber-6 wavetrain originating from Northeastern Atlantic" is not well connected to the preceding sentence. The logical flow of ideas should be improved.

Response: This sentence has been rephrased as below:

L20–23: “Such large-scale circulation adjustment is closely related to a new circumglobal Silk Road (CGSR) teleconnection, a Rossby wavetrain with a wavenumber of 6, which originates from Northeastern Atlantic and connects Asian climate with western Europe and North America heatwave and drought conditions.”

3. Seasonality of UDZC (L78). Is there also a strong UDZC signal in other seasons (e.g., the winter Eurasian jet)? Why is the UDZC signal prominent during high summer?

Response: In fact, we have checked the UDCZ signal in other seasons, and the strong UDZC state is only found during high summer (**Fig. R6**). The prominent UDZC during high summer may due to the climatology of Eurasian jet.

We have added these in the manuscript and Supplementary Information:

L99–101: “This feature is unique to the high summer since the late-1990s, with the absence of similar signal in the early summer or other season (i.e., June, Supplementary Figure 2 and 3; see Supplementary Text 1 for details).”

Supplementary Text S1: “We have checked the UDCZ signal in other seasons, and the strong UDZC state is only found during high summer of P2.

We present the sliding correlation between the EJA and WJA indices in other seasons (DJF, MAM, SON). As shown in Supplementary Figure 2, the sliding correlation in winter or autumn is non-significant during 1940–2022. There is barely significant UDZC at $\alpha = 0.05$ at spring around 1950 and 1980 but much weaker compared to high summer and non-significant during the recent decades.

The emergence of a strong UDZC may be related to the climatological state of the jet stream. In high summer, the subtropical jet stream is at its weakest and northernmost position, making it more susceptible to signals from mid- to high-latitudes and corresponding to a larger Rossby wave train wavelength^{1,2}.”

References:

1. Hong, X., Lu, R. & Li, S. Differences in the Silk Road Pattern and Its Relationship to the North Atlantic Oscillation between Early and Late Summers. *J. Clim.* **31**, 9283–9292 (2018).
2. Zhang, Y., Kuang, X., Guo, W. & Zhou, T. Seasonal evolution of the upper - tropospheric westerly jet core over East Asia. *Geophys. Res. Lett.* **33**, 2006GL026377 (2006).

Fig. R6 (Supplementary Figure 3 in revision) | Seasonality of UDZC interdecadal evolution. The 21-year sliding correlation between WJA and EJA indices during winter (DJF, blue), spring (MAM, green) and autumn (SON, red), using results from ERA-5.

4. Drought-heatwave coupling (L147). The concept of "drought-heatwave coupling" requires a more detailed explanation.

Response: We have added more explanation about the coupling between drought and heatwave as follows:

L216–223: “In addition, the similar distribution of heatwave and drought anomalies indicates a strong coupling between drought and heatwave (**Fig. 3d, e**), leading to compound hot-dry WCEs^{6,36}, which exert greater impact on ecosystem and human health⁶. For example, the drought condition can quickly trigger a heatwave in one day, while the heatwave takes 2–7 days to induce a drought event³⁶. Such drought-heatwave coupling often occurs when persistent atmospheric circulation patterns, such as the subtropical high/anticyclonic patterns, create a heat dome environment^{7,37}. In addition, the atmosphere-land feedbacks are also responsible for the compound hot-dry WCEs^{38,39}, which is reported to strengthen since the late-1990s⁶.”

References:

6. Zhang, P. *et al.* Abrupt shift to hotter and drier climate over inner East Asia beyond the tipping point. *Science* **370**, 1095–1099 (2020).
 7. Zhang, T., Deng, Y., Chen, J., Yang, S. & Dai, Y. An energetics tale of the 2022 mega-heatwave over central-eastern China. *npj Clim. Atmos. Sci.* **6**, 162 (2023).
 36. Mukherjee, S., Mishra, A. K., Zscheischler, J. & Entekhabi, D. Interaction between dry and hot extremes at a global scale using a cascade modeling framework. *Nat. Commun.* **14**, 277 (2023).
 37. Li, X., Zheng, J., Wang, C., Lin, X. & Yao, Z. Unraveling the roles of jet streams on the unprecedented hot July in Western Europe in 2022. *npj Clim. Atmos. Sci.* **7**, 323 (2024).
 38. Zhou, S. *et al.* Soil moisture–atmosphere feedbacks mitigate declining water availability in drylands. *Nat. Clim. Chang.* **11**, 38–44 (2021).
 39. Lansu, E. M., Van Heerwaarden, C. C., Stegehuis, A. I. & Teuling, A. J. Atmospheric Aridity and Apparent Soil Moisture Drought in European Forest During Heat Waves. *Geophys. Res. Lett.* **47**, e2020GL087091 (2020).
5. Wording adjustment (L159). Add "more" to the sentence: "This effect is much stronger and (more) zonal-symmetrical.....".

Response: Done as suggestion.

6. Model selection criteria (L244). The selection criteria for the 18 CMIP6 models and 11 LENS2 members should be explicitly stated.

Response: We have added more description about the model selection criteria:

L332–335: “To explore whether models can simulate the evolution UDZC in observation, models are selected based on the criteria that the correlation coefficient between WJA and EJA is significant ($R > 0.24$; $p < 0.05$) during 2000–2070 and is higher than 1900–1970 (bars with gray background in Supplementary Figure 11)

7. Clarification on model usage (L273-274). It should be clarified that all 42 models were used in the analysis, not just the selected 18 models. Do the LENS2 members also exhibit a similar relationship between future UDZC and Northern Hemisphere warming?

Response: For the first point, we have clarified that all 42 models were used in the analysis in the manuscript:

L395–396: “Therefore, we performed the inter-model correlation analysis among all 42 CMIP6 models.”

For the second question, our result suggests that LENS2 members do not exhibit a similar pattern of Fig. 6b. This is because that the outputs of LENS2 are all from the same model but with different initial status, leading to small difference in warming amplitude (**Fig. R7**). We have added more discussion on the results from LENS2:

L404–407: “The outputs of LENS2 members do not exhibit similar relationship between future UDZC and Northern Hemisphere warming. This is because that the outputs of LENS2 are all from the same model but with different initial status, leading to small difference in warming amplitude (Supplementary Figure 16).”

Fig. R7 (Supplementary Figure 16 in revision) | Inter-member correlation map of future surface warming amplitude (2000–2070 minus 1900–1970) with future UDZC of Eurasian jet. Similar to Fig. 7b, but using output from LENS2 Project.

8. In Figure 6a, the study focuses on the interannual relationship between the western jet axis (WJA) and the eastern jet axis (EJA) in the context of... (sentence incomplete, please clarify).

Response: We have updated the description of **Figure 7a** (the original **Figure 6a**):

L902–904: “This figure focuses on the evolution of interannual relationship between WJA and the EJA in the context of history and future warming scenario.”

Response to reviewer #2:

Weather and climate extremes are strongly related to jet stream variability, but their mechanisms still unclear. Lin et al. explored the linkages between Eurasian jet using upstream and downstream zonal consistency (UDZC) and local wave activity based on the reanalysis and model data. The authors projected that a strong UDZC could cause synchronized climate extremes. The idea is promising and the results is interesting. I provide some suggestions to help improve the manuscript.

Response: Thank you for your positive and supportive comments and detailed instruction on how to improve the manuscript. The quality of the manuscript has been greatly improved based on your comments. Please find our response below.

Major Comments

1. The innovation of the manuscript should be addressed currently both in the introduction and discussion, especially regarding the knowledge gaps in the jet stream.

Response: We have rephrased the last paragraph of introduction and the first paragraph of discussion to emphasize the innovation of the study.

L50–69: “While the impacts of jet stream strength alterations on WCEs’ distribution are acknowledged^{18,19}, existing research has predominantly focused on its long-term trends, meridional shifts and meandering^{25,26}. A significant knowledge gap exists in understanding the zonal coherence of jet stream intensity variations across vast longitudinal expanses, such as the Eurasian continent. The summertime Eurasian jet, characterized by its extensive longitudinal range (~120°) and notable regional variations²⁷, presents a complex system where the degree of coordinated intensity changes between its upstream and downstream sectors – what we conceptualize as “upstream-downstream zonal consistency” (UDZC) – has been largely overlooked. Specifically, how this UDZC has evolved, particularly in recent decades, and the mechanisms through which it might govern

the spatial synchronicity and large-scale patterns of WCEs during boreal summer, remain critical unanswered questions. Strong UDZC, implying a more uniform, continent-wide modulation of jet intensity, could potentially lead to more spatially extensive and synchronized WCEs^{5,18,19}, yet this linkage is poorly constrained.

This study addresses these crucial knowledge gaps by investigating the long-term evolution of UDZC and climate impact of strong UDZC. We demonstrate an unprecedented surge in Eurasian jet UDZC since the satellite era, a phenomenon unparalleled in the past two centuries. This surge is accompanied by the emergence of a new dominant paradigm in Eurasian upper-level circulation, characterized by jet intensity changes rather than mere latitudinal shifts. We further reveal its profound implications for widespread heatwave and drought occurrences across Eurasia and its teleconnections extending to North America.”

L351–368: “This study reveals a critical yet previously overlooked aspect of Eurasian jet stream variability: that is the variation in its UDZC. Our primary innovation lies on identifying and quantifying a rapid and unprecedented intensification of UDZC since the 1970s (**Fig. 1c**), a feature largely absent in historical records even spanning the past two centuries. This finding directly addresses a significant knowledge gap concerning the recent evolution of jet stream coherence over continental scales, moving beyond traditional analyses of meridional shifts or overall weakening/strengthening trends^{25,26}. We demonstrate that this surge in UDZC is related to a different Eurasian upper-level circulation, where the dominant mode of variability has transitioned towards synchronized jet intensity changes across the continent.

Furthermore, our research innovatively links this enhanced UDZC to a more zonally uniform and synchronized pattern of heatwave and drought occurrences across Eurasia. A key finding is the identification of the CGSR teleconnection, a wavenumber-6 Rossby wave train originating from the North Atlantic, as the mechanism through which the influence of Eurasian jet UDZC extends to remote

regions like North America. This provides a new perspective on hemispheric-scale climate linkages mediated by the jet stream. Collectively, these results underscore the profound and previously underestimated impact of a zonally coherent jet stream in triggering synchronized WCEs across the Northern Hemisphere, with significant implications for ecosystems¹, human health⁴, and global food security⁵.”

References:

1. Coumou, D. & Rahmstorf, S. A decade of weather extremes. *Nat. Clim. Chang.* 2, 491–496 (2012).
 4. Chen, H. et al. Spatiotemporal variation of mortality burden attributable to heatwaves in China, 1979–2020. *Sci. Bull.* 5 (2022).
 5. Kornhuber, K. et al. Risks of synchronized low yields are underestimated in climate and crop model projections. *Nat. Commun.* 14, 3528 (2023).
 18. Nakamura, N. & Huang, C. S. Y. Atmospheric blocking as a traffic jam in the jet stream. *Science* 361, 42–47 (2018).
 19. Chen, G., Nie, Y. & Zhang, Y. Jet Stream Meandering in the Northern Hemisphere Winter: An Advection–Diffusion Perspective. *J. Clim.* 35, 2055–2073 (2022).
 24. Li, S., Sato, T., Nakamura, T. & Guo, W. East asian summer rainfall stimulated by subseasonal Indian monsoonal heating. *Nat. Commun.* 14, 5932 (2023).
 25. Dong, B., Sutton, R. T., Shaffrey, L. & Harvey, B. Recent decadal weakening of the summer Eurasian westerly jet attributable to anthropogenic aerosol emissions. *Nat. Commun.* 13, 1148 (2022).
 26. Jiang, J. et al. Precipitation regime changes in High Mountain Asia driven by cleaner air. *Nature* (2023) doi:10.1038/s41586-023-06619-y.
 27. Li, D., Chen, H., Liu, P. & Zhou, C. Zonally asymmetric mode of anomalous activity in summer Asian subtropical westerly jet and its possible sources. *Theor. Appl. Climatol.* 139, 17–32 (2020).
2. Some parts of the result section should be rephrased for readability (see also my detailed comments).

Response: Checked. Please see our response to detailed comments.

3. For the spatial correlation and regression analysis (figures 2-6), the potential "false discovery" that may be raised from multiple hypothesis testing while

simultaneously testing thousands of grids should be considered. Controlling the false discovery rate for multiple tests is a possible way to test the results.

Response: Thank you for your valuable suggestion regarding the potential for "false discovery" arising from multiple hypothesis testing across thousands of grids in our spatial correlation and regression analyses (Figures 2–7). We agree that this is an important consideration.

We have explored the use of the Benjamini-Hochberg False Discovery Rate (FDR^{1,2}) procedure to control for multiple testing. For **Figures 2** and **7** (the original **Figure 6**), as also mentioned in the Methods section, we have used the FDR correction for P-values at $\alpha_{\text{FDR}} = 0.10$, which had a relatively minor impact on the primary conclusions drawn from these figures (**Fig R1** and **R2**). The significant patterns generally remained robust.

However, when consider the impact of UDZC on surface climate (for **Figures 3, 4, and 5**), we found a substantial reduction in the number of grid points identified as statistically significant. Such reduction in significant grid also appears when we test the well-known wavetrain impact of the Silk Road Pattern on surface temperature^{3,4,5} (**Fig. R3a and b**), as well as the preceding ENSO's impact on summer western Pacific precipitation^{6,7,8} (**Fig. R3c and d**), as examples. We believe this marked decrease in significant regions for **Figures 3, 4, and 5** upon FDR application may be attributable to a combination of factors:

1. Relatively smaller effective sample sizes for some of the analyses involved in these figures (degree of freedom at 22).

2. The inherent complexity and multifactorial nature of near-surface climate anomalies (e.g., heatwave frequency, SPEI, LWA, TC track density as shown in these Figures). These variables are often influenced by a wider array of processes compared to upper-level circulation features, potentially leading to weaker individual correlations that are more susceptible to being filtered out by stringent FDR control.

3. The conservative nature of the FDR method itself, particularly in contexts with complex dependency structures or when the proportion of true null hypotheses is very high. While effective at controlling false positives, it can sometimes increase the risk of Type II errors (false negatives), potentially obscuring some genuinely correlated regions by classifying them as non-significant².

Given these considerations, and the concern that a strict application of FDR to these Figures might overly penalize true signals in these more complex, near-surface analyses, we decided not to use the FDR-corrected p-values for interpreting the primary results in these specific figures.

Instead, to enhance the reliability of our findings and reduce the likelihood of false positives in these Figures, we have adopted a more stringent significance threshold for the Student's *t*-tests, setting it at $\alpha=0.05$ (95% confidence level) for all analyses presented in these figures.

We believe this approach provides a reasonable balance between controlling for spurious findings and retaining the ability to identify potentially meaningful, albeit complex, relationships in the near-surface climate system.

Reference:

1. Benjamini, Y. & Hochberg, Y. Controlling the False Discovery Rate: A Practical and Powerful Approach to Multiple Testing. *J. R. Stat. Soc. B* **57**, 289–300 (1995).
2. Wilks, D. S. “The Stippling Shows Statistically Significant Grid Points”: How Research Results are Routinely Overstated and Overinterpreted, and What to Do about It. *Bull. Am. Meteorol. Soc.* **97**, 2263–2273 (2016).
3. Li, X., Lu, R. & Ahn, J.-B. Combined Effects of the British–Baikal Corridor Pattern and the Silk Road Pattern on Eurasian Surface Air Temperatures in Summer. *J. Clim.* **34**, 3707–3720 (2021).
4. Wang, L., Xu, P., Chen, W. & Liu, Y. Interdecadal variations of the silk road pattern. *J. Clim.* **30**, 9915–9932 (2017).
5. Liu, Y. Relationship between the Silk Road and Circumglobal Teleconnection Patterns on the Interannual and Interdecadal Timescales. *Atmosphere* **14**, 1626 (2023).

6. Xie, S.-P. et al. Indo-Western Pacific Ocean Capacitor and Coherent Climate Anomalies in Post-ENSO Summer: A review. *Adv. Atmos. Sci.* **33**, 411–432 (2016).
7. Wang, B., Wu, R. & Fu, X. Pacific–East Asian Teleconnection: How Does ENSO Affect East Asian Climate? *J. Clim.* **13**, 1517–1536 (2000).
8. Chen, W. et al. Recent Progress in Studies of the Variabilities and Mechanisms of the East Asian Monsoon in a Changing Climate. *Adv. Atmos. Sci.* **36**, 887–901 (2019).

We have added the detailed information of FDR in Methods section:

L495–501: “In addition, for the correlation filed in **Fig. 2** and **Fig. 7**, the Benjamini–Hochberg false discovery rate (FDR) correction for P-values is used to mitigate the increase in false positives from multiple testing⁸⁴. We set the FDR control level at $\alpha_{\text{FDR}} = 0.1$ to maintain a global α level of 0.05⁸⁵. The P_{FDR} was estimated as follows:

$$P_{\text{FDR}} = \max_{j=1,\dots,k} [P_j: P_j \leq (j/N)\alpha_{\text{FDR}}] \quad (1)$$

where P_j represents the j -th P-value after sorting all N local test P-values in ascending order. Grid cells with local test P-values less than P_{FDR} are considered significant.”

Reference:

84. Benjamini, Y. & Hochberg, Y. Controlling the False Discovery Rate: A Practical and Powerful Approach to Multiple Testing. *J. R. Stat. Soc. B* **57**, 289–300 (1995).
85. Wilks, D. S. “The Stippling Shows Statistically Significant Grid Points”: How Research Results are Routinely Overstated and Overinterpreted, and What to Do about It. *Bull. Am. Meteorol. Soc.* **97**, 2263–2273 (2016).

Fig. R1 (Figure 2 in revision) | Shifting dominant mode of Eurasian westerly. **a** and **c** The EOF1 mode of U200 (shading: correlation map) during P1 (1958–1998) and P2 (1999–2022), respectively. Black lines outline regions exceeding the 95% confidence level, with that after controlling for the false discovery rate ($\alpha_{\text{FDR}} = 0.10$) stippled. The red line indicates the climatology of jet stream axis. The blue box denotes the regions for performing EOF. **b** and **d** Same as **a** and **c**, but for EOF2. Shown in **e** are the normalized UDZC index (black bar) and the PCs corresponding to EOFs in **a–d**, as well as the WJA (blue curve) index and the EJA (red curve) index during P2. **f** Correlations between U200 PCs and UDZC, WJA and EJA indices for P1 and P2.

Fig. R2 (Figure 7 in revision) | Future change in UDZC of Eurasian jet. **a** On the left is the evolution of Corr (WJA, EJA) in a 21-year sliding window, including results from the multi-source reanalysis data-based observation mean (red) and from the selected CMIP6 models (green) and LENS2 members (purple) according to Supplementary Figure 11. This figure focuses on the evolution of interannual relationship between WJA and EJA in the context of history and future warming scenario. The solid green/purple lines and shading indicate multi-model mean and inter-model spreading, respectively. On the right is the multi-model mean of Corr (WJA, EJA) during historical period (1900–1970) and future period (2000–2070). **b** Map of inter-model correlation between future surface warming amplitude and future UDZC of Eurasian jet (see “Model simulation data” in Methods for details), with statistically significant values after controlling for the false discovery rate ($\alpha_{FDR} = 0.10$) hatched. Red and blue boxes outline the two hot-spots of future warming regions over the represented land (20°N–60°N, 0–70°E) and ocean (15°N–55°N, 170°E–130°W), respectively. **c–d** Inter-model relationship of future UDZC and future surface warming over the hotspot of land and ocean. The thick red dash line represents the ordinary least-squares fit, with shading as the 95% confidence interval.

Fig. R3 | Climate impact of SRP and ENSO **a** The regression map of SAT against the normalized SRP index (units: K) during 1979-2018, with significance at $\alpha=0.05$ stippled. **b** Same as **a**, but with statistically significant values after controlling for the false discovery rate ($\alpha_{FDR} = 0.10$) stippled. **c** and **d** Similar to **a** and **b**, but for preceding ENSO's impact (represented by Nino3b index) on precipitation (units: mm day^{-1}).

4. The uncertainty of the two-centuries data (20thCRV3 reanalysis) should be considered, especially for the early-period and Eurasian regions.

Response: Thank you for this suggestion. Instead of using multi-member mean of 20thCRV3 data, here we used its 80-member ensemble to investigate its uncertainty. In addition, we also update the data, which can extend the study periods to 1806. Therefore, the 20thCRV3 reanalysis covers the period 1806-2015.

We firstly investigate the uncertainty of EJA and WJA indices (**Fig. R4**), which is quantified as the inter-model spread. Both EJA and WJA indices exhibit a strengthening trend in the past two centuries, but with decreasing uncertainty. The inter-model spread reaches 2.5 m s^{-1} during the early period (before 1880s) and decreasing to about 1 m s^{-1} after 1958.

With the data of those members, the result of R between EJA and WJA in 20thCRV3 is now calculated in each member and shown as their multi-member mean with inter-member spread (Light green curve and shading in **Fig. R5c**). Similar to **Fig. R4**, the correlation between them also exhibits relatively high uncertainty before 1880s (**Fig. R5c**). However, although the uncertainty is much smaller after the 1990s, multi-member mean of 20thCRV3 still underestimate the strong UDZC. We have added the description of uncertainty in the manuscript:

L101–106: “Using 80 members of 20thCRV3 reanalysis, which extends the record back to 1806 (**Fig. 1c**), we further confirmed it as an unprecedented phenomenon over the past nearly two centuries. The 20thCRV3 reanalysis exhibits relatively high uncertainty before 1880s, as green shading indicates. Although the uncertainty is much smaller after the 1990s, their multi-member means still underestimate the strong UDZC (**Fig. 1c**, light green).”

Fig. R4 | EJA and WJA indices in 20thCRV3 members. a Multi-member mean of EJA indices (black curve) with red shading represents the inter-model spread (inter-model standard deviation). The inter-model spread is also shown as the blue curve. **b** Same as **a**, but for the WJA indices.

Fig. R5 (Figure 1 in revision) | Unprecedented surge in UDZC of the Eurasian jet axis. **a** Climatology of the Eurasian jet (green shading) and the standard deviation of U200 (contours; units: m s^{-1}). **b** Year-to-year variability of WJA and EJA indices, with R1 and R2 represent the correlation coefficient between the two indices during P1 (1958–1998) and P2 (1999–2022). **c** The 21-year sliding correlation of WJA and EJA indices in multi-source data. The gray dash lines indicate the 95% (99.9%) confidence level.

5. In the results, the authors used the jet indices from U200, but for other local wave activities from different heights, such as H150, H500, H300, and V200. Is there a way to reduce the number of the heights? Or there are other ways to illustrate their connections and mechanisms?

Response: According to your suggestion, we have unified the height-level of upper-troposphere in this study to 200 hPa (Fig. R6 and R7). In addition, we have adopted a more stringent significance threshold for the students's *t*-tests, setting it

at $\alpha = 0.05$ (95% confidence level).

Fig. R6 (Figure 4 in revision) | Regime shifts linked to enhancing UDZC of the Eurasian jet axis. **a** Regression map of 200-hPa Local Wave Activity (LWA) against the UDZC index during P1, with 95% confidence hatched. Also shown are the composite pattern of Tibet Plateau High (TPH; blue; measured by H200 in 12530 gpm) and western North Pacific subtropical high (WNPSH; red; measured by H500 in 5870 gpm). Shadings represent their climatology with thick solid (dash) lines represent their mean states in strong (weak) jet years (selected base on the threshold of 0.5 SD; see Methods section for detail), respectively. **b** Correlation map of tropical cyclone (TC) track density (shading; 95% confidence enclosed by red line) against the UDZC index during P1. Vectors represent the regressed vertical-integrated water vapor flux (WVT), with 95% confidence in blue color. **c–d** Same as **a–b**, but for P2. The purple line indicates the climatology of jet axis.

Fig. R7 (Figure 5 in revision) | Circumglobal impacts of strong UDZC in Eurasian jet. **a** Regression map of H200 (shading; units: gpm) and V200 (contour; units: m s^{-1}) against the UDZC index during P2, with 95% confidence hatched. Vectors represent the regressed wave activity flux (WAF; units: $\text{m}^{-2} \text{s}^{-2}$) at 200 hPa. **b** Regression map of NET-heatwave frequency (shading; units: days month^{-1}) over western Europe against the UDZC index during P2, with 95% confidence stippled. Contours represent the regressions of SPEI anomalies, with 95% confidence hatched. **c** Similar to **b**, but for the North America.

- The discussion should be strengthened to highlight this study's innovation. The discussion just focused on the model results and future changes of UDZC. I think that it is not enough to address the innovation of the whole study, such as how the Eurasian jet variability and potential mechanisms work.

Response: We have rephrased the discussion section to emphasize the innovation of our study, please see our responses to your major comment 1.

Details:

- L57: Could you specify "the satellite era"?

Response: the satellite era refers to the period after 1979, we have added more detail in the manuscript:

L64–66: “We demonstrate an unprecedented surge in Eurasian jet UDZC since the satellite era (i.e., the post-1979 period), a phenomenon unparalleled in

the past two centuries.”

- L62: Is it right? The Eurasian jet is meandering in Figure 1.

Response: The map projection may lead to misunderstanding of the jet stream meandering. Therefore, we redraw the climatology distribution of Eurasian jet stream with lat-lon projection, as **Fig. R8** shows. Apparently, in climatology, both the U200 distribution or the jet axis reveals that the majority of Eurasian jet is quite straight with less meandering.

So we have rephrased our description as follow:

L72–73: “The Eurasian jet, peaking at 32 m s^{-1} in climatology, is quite straight in the west-east direction with its axis situated around 42°N during high summer (July–August; **Fig. 1a**).”

Fig. R8 | Climatology of Eurasian jet at U200. Shadings represent the climatology of high summer Eurasian jet (units: m s^{-1}) during 1958–2022. Red line indicates the jet stream axis.

- L66-67, 89-92: It is unclear for me how did you decide the point year of "1958"?

Response: The year 1958 marks the earliest period when the global observational system, particularly for vital upper-air data, became sufficiently developed and standardized to support more reliable and comprehensive atmospheric reanalysis, primarily due to the advancements achieved by the International Geophysical Year³² (1957–1958).

The International Geophysical Year³² led to a substantial expansion and standardization of the global radiosonde (weather balloon) network. Regular,

globally coordinated upper-air observations, crucial for understanding the three-dimensional state of the atmosphere, became more systematic from 1958 onwards. These improvements in data quantity, quality, and global coverage from 1958 provided a more robust and scientifically defensible baseline for reanalysis projects aiming to create consistent historical atmospheric datasets³².

In other words, the reanalysis data after 1958 is more reliable, especially for the upper-air data. And to further support the robustness of the main findings in this study, we also reproduce the **Fig.1b** and **Fig. 2** using data from the JRA-55 (**Fig. R9** and **R10**), which has also been provided since 1958. We have added a brief explanation on the choice of 1958 as the start year:

L128–132: “Since the reanalysis data after 1958 is more reliable due to the improved observation network after the International Geophysical Year³², and to support the robustness of the main findings in this study, we further use the JRA-55 data (this official data has been provided since 1958) to reproduce the results obtained from ERA5 data, thus we mainly focus on the post-1958 period unless otherwise stated.”

L191–193: “Above results can be mutual mirrored with each other between results from the ERA5 (**Fig. 2**) and those from the JRA55 (Supplementary Figure 7), suggesting the reliability of our findings.”

Reference:

- 32. Grant, A. N., Brönnimann, S., Ewen, T. & Nagurny, A. A New Look at Radiosonde Data prior to 1958. *Journal of Climate* **22**, 3232–3247 (2009).

Fig. R9 | Year-to-year variability of WJA and EJA indices. Same as Fig. 1b, but produced using JRA-55

Fig. R10 (Supplementary Figure 7 in revision) | Shifting Eurasian westerly modes. Same as Fig. 2, but produced using JRA-55

- L68-71: This should be moved down and combined with the next paragraph.

Response: Done as suggestion.

L95-99: “Specifically, these two indices are highly correlated since the 1999 ($R = 0.79, p = 4.7 \times 10^{-6}$) while almost independent of each other before 1998 ($R = 0.03, p = 0.85$). The strong consistence of result after the late-1990s between datasets supports the robustness of the sudden emergence of strong UDZC state in Eurasian jet (Fig. 1b).”

- L73: Could you specify the number of the sliding time window?

Response: Done as suggestion. We also tried the results in different sliding window. As Fig. R11 shows, the choice on sliding window does not affect our results.

L85–87: “Therefore, the interdecadal evolution of UDZC state can be simplified as the R between WJA and EJA indices in a sliding window (**Fig. 1c**). The 21-year window is used here and the result is not impact by the choice on sliding window (Supplementary Figure 1).”

Fig. R11 (Supplementary Figure 1 in revision) | Evolution of High Summer UDZC in different sliding window. Sliding correlation of WJA and EJA indices in 17-year (blue), 19-year (green) and 21-year (red) sliding window.

6. L110-112: I think that this should be in the introduction.

Response: Done as suggestion.

L46–49: “Different structures of upper-level jet stream also have great impacts on surface weather and climate anomalies, mainly via changes in the moisture transport²⁰, jet-related secondary circulation^{21,22}, activities of synoptic disturbance²³, blocking events¹⁸ and Rossby wave behavior²⁴.”

References:

20. Gimeno-Sotelo, L., Sorí, R., Nieto, R., Vicente-Serrano, S. M. & Gimeno, L. Unravelling the origin of the atmospheric moisture deficit that leads to droughts. *Nat. Water* **2**, 242–253 (2024).
21. Song, S.-Y., Yeh, S.-W. & Park, J.-H. Change in Relationship between the East Asian Winter Monsoon and the East Asian Jet Stream during the 1998–99 Regime Shift. *J. Clim.* **32**, 6163–6175 (2019).
22. Zhan, R., Wang, Y. & Ding, Y. Impact of the Western Pacific Tropical Easterly Jet on Tropical Cyclone Genesis Frequency over the Western North Pacific. *Adv. Atmos. Sci.* **39**, 235–248 (2022).

23. Lee, S. & Kim, H. The Dynamical Relationship between Subtropical and Eddy-Driven Jets. *J. Atmos. Sci.* 60, 1490–1503 (2003).
24. Li, S., Sato, T., Nakamura, T. & Guo, W. East asian summer rainfall stimulated by subseasonal Indian monsoonal heating. *Nat. Commun.* 14, 5932 (2023).

7. L182-185: How about the results if you used H200?

Response: The regression map of H200 is similar to that of H300. **According to your comment 5**, we have unified the height-level of upper-troposphere in this study to 200 hPa (Please see our response to your major comment 5).

8. L305: SEPI? Is it SPEI?

Response: Corrected, thanks.

9. L308-309: The reference(s) for how you calculate SPEI is needed.

Response: We have given more detail about the calculation of SPEI with references:

L435–441: “Specifically, first, a simple climatic water balance is determined by calculating the monthly difference (D_i) between precipitation and potential evapotranspiration (using monthly mean temperature⁷⁵). Next, the calculated D_i values are aggregated at 1-month timescale, similar to the procedure for the Standardized Precipitation Index⁷⁶. A probability distribution is then fitted to the aggregated D_i . Since the D_i contains negative values, the log-logistic distribution is then fitted to the aggregated D_i , with parameters estimated using the L-moment procedure⁷⁷. Finally, the SPEI is obtained as the standardized value of the cumulative distribution function of the aggregated D_i , with parameters estimated using the L-moment procedure⁷⁷”.

References:

75. Thornthwaite, C. W. An Approach toward a Rational Classification of Climate. *Geographical Review* **38**, 55 (1948).
76. McKee, T. B., Doesken, N. J. & Kleist, J. The relationship of drought frequency and duration to time scales. *Eighth Conf. on Applied Climatology* 179–184 (1993).
77. Ahmad, M. I., Sinclair, C. D. & Werritty, A. Log-logistic flood frequency analysis. *Journal of Hydrology* **98**, 205–224 (1988).

10. L328-338: A little confusion about the analysis. The Eurasian jet is based on U200, why you calculate correlations between PCs of U200 with the Eurasian jet again? How do you link the Eurasian jet and SST?

Response: Thank you for this insightful comment and for raising this important point. We have revised the manuscript to clarify our rationale.

For the first point, our primary goal with this correlation analysis was not simply to show that indices derived from the same U200 data are correlated, but to demonstrate a fundamental regime shift in the Eurasian jet's dominant behavior. During P1, the jet's main variability (EOF1; **Fig. R1a**) was its meridional displacement, which logically showed little correlation with our jet axis indices (i.e., WJA & EJA; **Fig. R1f**), as well as the second mode (**Fig. R1b** and **f**). In contrast, during P2, the jet's dominant variability (EOF1; **Fig. R1c**) became the intensity change itself. Therefore, the new, high correlation that emerges in P2 is the key evidence of this shift. For the second mode (EOF2; **Fig. R1d**), which also reflects its meridional displacement, the corresponding PC2 is nearly independent with three Eurasian jet indices (UDCZ, WJA and EJA indices; **Fig. R1f**). This stark difference between the two periods validates our conclusion that a new paradigm, dominated by zonal-consistent intensity changes, has emerged.

We have modified the analysis about the relationship between PCs of U200 and Eurasian jet indices.

L186–191: “To further explore how PCs of U200 is related to Eurasian jet indices here (UDZC, WJA and EJA index), we calculated their Rs during two sub-periods (**Fig. 2f**). The UDZC index is nearly independent to other PCs (**Fig. 2f**), except the PC1 during P2, and highly correlated with the WJA and EJA index during P2 (R reaches 0.87 and 0.95 respectively; **Fig. 2e**). Therefore, climate impacts of Eurasian jet with strong UDZC at interannual scale can be illustrated by a comparison of UDZC index related pattern between the two sub-periods.”

For the second point, we firstly observed the CGSR originates from the eastern North Atlantic which is the key region triggering Rossby wavetrains (WAF in **Fig. 5a**). Therefore, apart from internal process of atmosphere, the SST *in-situ* can also be a possible driver of the CGSR wavetrain. In addition, following suggestion of Reviewer 1, to better illustrate the relationship between SST and Eurasian jet, we used the Extended MCA to capture the potential evolution of North Atlantic SST and Eurasian jet (**Fig. R12**). We have included this as Figure 6 and discussed it in the text as follows.

L292–314: “The high pressure around the eastern North Atlantic and western Europe, which also resides in the exit region of the North Atlantic jet, is the key center from which the CGSR wave train emanates³⁰ (shading in **Fig. 5a**). The excitation of the Rossby wavetrain may involve the barotropic instability⁴⁹. In high summer, the air flow over this region is likely to be barotropically unstable because of the strong zonal shear from the basic zonal wind⁵⁰. Therefore, if one disturbance is excited by certain forcing like the sea surface temperature (SST) anomalies, it could develop readily through gaining barotropic energy from the basic flow and propagate eastward⁵⁰.

Given the memory effect of ocean^{51,52}, if the CGSR is triggered by the North Atlantic SST, the preceding SST signals from several months in advanced shall be observed. Therefore, to capture this entire evolutionary process, we employ the Extended Maximum Covariance Analysis (EMCA) method^{53,54} to examine the coupled evolution of North Atlantic SST starting from February and the subsequent summer U200 over Eurasian during P2 (**Fig. 6**). For the two fields analyzed here, the SST mode with persistent signals from preceding February (**Fig. 6a**) can be physically interpreted as the oceanic forcing, and the U200 mode as the atmospheric response. The first pair of EMCA modes (EMCA1) clearly reveal the coupled mode between North Atlantic SST and strong UDZC. The SST warming around the eastern North Atlantic exhibit an apparent persistent

evolution since the previous February (**Fig. 6a**). Correspondingly, significant westerly wind anomalies are observed along the Eurasian jet axis, with easterly wind anomalies at high- and low- latitudes (**Fig. 6b**). The correlation between the EMCA1-U200 and UDZC index is robust ($R = 0.94$, $p = 5 \times 10^{-13}$; **Fig. 6c**). And the EMCA-U200 pattern is highly similar to the EOF1 mode during P2 (**Fig. 2c**; pattern correlation reaches 0.97). Therefore, results from EMCA strongly suggests the potential causality between the North Atlantic SST anomalies and the strong UDZC of Eurasian jet.”

References:

30. Xu, P., Wang, L. & Chen, W. The British–Baikal Corridor: A Teleconnection Pattern along the Summertime Polar Front Jet over Eurasia. *J. Clim.* **32**, 877–896 (2019).
49. Xu, P. *et al.* The british–okhotsk corridor pattern and its linkage to the silk road pattern. *J. Clim.* **35**, 5787–5804 (2022).
50. Simmons, A. J., Wallace, J. M. & Branstator, G. W. Barotropic Wave Propagation and Instability, and Atmospheric Teleconnection Patterns. *J. Atmos. Sci.* **40**, 1363–1392 (1983).
51. Namias, J. & Born, R. M. Temporal coherence in North Pacific sea-surface temperature patterns. *J. Geophys. Res.* **75**, 5952–5955 (1970).
52. Deser, C., Alexander, M. A. & Timlin, M. S. Understanding the persistence of sea surface temperature anomalies in midlatitudes. *J. Clim.* **16**, 57–72 (2003).
53. Polo, I., Rodríguez-Fonseca, B., Losada, T. & García-Serrano, J. Tropical Atlantic Variability Modes (1979–2002). Part I: Time-Evolving SST Modes Related to West African Rainfall. *J. Clim.* **21**, 6457–6475 (2008).
54. García-Serrano, J., Losada, T., Rodríguez-Fonseca, B. & Polo, I. Tropical Atlantic Variability Modes (1979–2002). Part II: Time-Evolving Atmospheric Circulation Related to SST-Forced Tropical Convection. *J. Clim.* **21**, 6476–6497 (2008).

Fig. R12 (Fig. 6 in revision) | The EMCA1 modes of co-varying North Atlantic SST evolution and Eurasian U200. a–b, the homogeneous correlation maps against the temporal coefficients corresponding to EMCA1, with 95% confidence stippled. The red line in (b) indicates the Eurasian jet axis. Red (blue) boxes denote the SST (U200) regions for EMCA. c, Time series of EMCA1-SST (red) and EMCA1-U200 (blue), with the UDZC index (black bar).

11. L342-343: How many years of the strong or weak Eurasian jet are selected in the analysis? This information is unclear here and figures.

Response: Thank you for raising this important point. We have provided a Table showing these years selected (**Table R1**) and added more detailed about the selected years in Methods section:

L490–492: “During P1, there are 14 strong Eurasian jet years and 14 weak Eurasian jet years. During the P2, there are six strong Eurasian jet years and six weak Eurasian jet years (See Supplementary Table 2 for detail).”

Table R1 (Supplementary Table 2 in revision) | Selected strong/weak Eurasian jet years in two sub-periods. Strong (weak) Eurasian jet year are selected as the year whose corresponding value of normalized UDZC index is above + 0.5 (below -0.5).

	Strong Eurasian jet years	Weak Eurasian jet years
P1	1959, 1961, 1963, 1964, 1967, 1969, 1972, 1976, 1981, 1987, 1990, 1991, 1993, 1995	1958, 1962, 1965, 1966, 1968, 1970, 1974, 1975, 1978, 1984, 1986, 1989, 1996, 1997
P2	2003, 2006, 2009, 2013, 2020, 2022	1999, 2000, 2002, 2008, 2012, 2015

12. L346-348: Could you specify the jet axis? Why the jet axis vary? How about the coherent variability between this jet index and the traditional jet stream index because you stated that the new method has some advantages?

Response: Thank you for this important clarification. For the first question, to be more specific about why the variation of jet axis and how this annually varying jet axis is determined, we have added the following description to the 'Jet stream indices' subsection within the 'Methods' section in the revised manuscript:

L504–514: “Unlike traditional method which obtain the jet stream index with area-averaging over fixed area, we calculate the jet stream axis index by averaging the 200-hPa zonal wind along the jet axis. Such choice of jet stream index is mainly due to the variation of jet axis latitude, which can be influenced by planetary-scale Rossby wave propagation and evolution⁸⁶, as well as thermal and dynamical interactions between polar and mid-to-low latitudes^{56,87}. Therefore, the latitude of jet stream axis actually varies from year to year.

To define the jet axis, we first identify, at each longitude within a latitudinal band (25°N–60°N), the latitude of the maximum 200-hPa zonal wind speed. The line connecting these points of maximum wind speed across the longitudes constitutes the jet axis, which is determined in every specific year. Therefore, this method is capable of capturing the year-to-year variation in the core position of the jet stream.”

References:

56. Chowdary, J. S. et al. The Eurasian Jet Streams as Conduits for East Asian Monsoon Variability. *Curr. Clim. Change Rep.* **5**, 233–244 (2019).

86. Stendel, M., Francis, J., White, R., Williams, P. D. & Woollings, T. The jet stream and climate change. in *Climate Change* 327–357 (Elsevier, 2021).
87. Yan, Y., Li, C. & Lu, R. Meridional Displacement of the East Asian Upper-tropospheric Westerly Jet and Its Relationship with the East Asian Summer Rainfall in CMIP5 Simulations. *Adv. Atmos. Sci.* **36**, 1203–1216 (2019).

For the second question, to better compare the difference between the jet axis index and traditional area-fixed index, we defined the jet area index for western and eastern part of Eurasian jet, as the blue and red boxes outlines (**Fig. R13**). For both parts, although the jet axis and jet area index are highly correlated ($R > 0.94$), the jet axis index always shows stronger jet intensity than jet area index. For years like 1996 in **Fig. R13b** and 2022 in **Fig. R13c**, the traditional jet area-fixed index highly underestimates the jet intensity, mainly due to strong deviated jet axis.

Fig. R13 | Comparison of Jet Axis index and Jet Area index. **a** Shadings represent the climatology of high summer Eurasian jet (units: m s^{-1}) during 1958–2022. Blue and red boxes represent the area for defining Jet Area indices (red curve in **b** and **c**). **a–b** Jet Axis indices (black curve in **b** and **c**) are the same to WJA and EJA indices in the manuscript.

13. L353-355: Why did you project the U200 from 1958-2022 onto EOF1 during 1999-2022, rather than the whole period? Is it reasonable?

Response: Thank you for this key question. Our goal was to track the historical evolution of this "zonal consistent intensity change" mode over our main study period. However, since the "zonal consistent intensity change" mode emerged

only in the post-1999 period (P2), the corresponding spatial pattern can only be obtained by performing EOF during P2. Analyzing the EOF for the entire period would have blended different modes, obscuring this critical shift we aimed to highlight. Then, the “projection” method allows us to investigate the amplitude of such pattern in each year even beyond P2.

Specifically, we choose the period 1958–2022 due to higher reliability of datasets since 1958, due to the advancements achieved by the International Geophysical Year¹. Thereafter, we define its spatial pattern from the period where it is dominant (P2 EOF1; **Fig. 2c**). Then, projecting the full dataset during 1958–2022 onto this pattern allowed us to create the UDZC index, which quantifies the emergence and strengthening of this new circulation paradigm.

Reference:

1. Grant, A. N., Brönnimann, S., Ewen, T. & Nagurny, A. A New Look at Radiosonde Data prior to 1958. *J. Clim.* **22**, 3232–3247 (2009).

We have added more explanation in the Methods section:

L536–547: “Second, to investigate and contrast the interannual relationship between the Eurasian jet and climate within periods of different UDZC states, a uniformly defined UDZC index is therefore needed to describing year-to-year variability of UDZC especially for pre-1999 during which the EOF cannot capture the UDZC mode. To achieve this, we first used EOF analysis on the post-1999 period to capture the strong UDZC mode (EOF1 in P2; **Fig. 2c**), also referred to as the “intensity change” mode. To derive the interannual variability of this mode across the entire study period (1958–2022), we project the U200 anomalies onto the EOF1 mode during P2 (regression pattern as contours shown in **Fig. 2c**) over the EOF region (i.e., blue box in **Fig. 2c**). This projection yields the interannual UDZC index for 1958–2022. In other words, this uniformly defined interannual UDZC index can directly reflect the strength of UDZC in a specific year. For example, a strong positive (negative) value reflects to the degree of consistent enhancement of the upstream and downstream sections of the Eurasian jet.”

Figures:

1. Fig 1. Which data are used in WJA and EJA calculation in panel b? Could you change the color of confidence lines as black or gray to reduce the color in panel c?

Response: As **Fig. R14** shows, we used the ERA-5 reanalysis to calculate WJA and EJA index, we have added “ERA-5” in **Fig. 1b** and change the confidence lines in **Fig. 1c** to gray lines.

Fig. R14 (Figure 1 in revision) | Unprecedented surge in UDZC of the Eurasian jet axis. a Climatology of the Eurasian jet (green shading) and the standard deviation of U200 (contours; units: $m s^{-1}$). **b** Year-to-year variability of WJA and EJA indices, with R1 and R2 represent the correlation coefficient between the two indices during P1 (1958–1998) and P2 (1999–2022). **c** The 21-year sliding correlation of WJA and EJA indices in multi-source data. The gray dash lines indicate the 95% and 99.9% confidence level.

- Fig 2. Could you consider the false discovery rate in the correlation and regression maps (see also my main comments)? Is it better to change the ESWJ as a line rather than a bar?

Response: For the first point, we have applied the false discovery rate in **Fig. 2**, please see our response to your major comment 3.

For the second point, we have tried change the UDZC index (the original ESWJ index) as a line but found it does not show strong clarity (**Fig. R15a**).

Therefore, we decided to remain it as a bar for the clarity of **Fig. 2e**.

Fig. R15 | Comparison between two plot type of UDZC index. a Same as Fig. 2e, but change UDZC index as a line. **b** The original Fig. 2e, as a comparison.

- Fig 4. It will be better to provide the strong (weak) jet years in the figure or supplementary.

Response: We have provided a supplementary Table to show the selected years (**Table R1**). Please see our response to your detail comment 11.

- Fig 5. Why not select the same height for geopotential height and V200? It looks a little confusing for different heights. Panel c, please double check the significant results are colored.

Response: Done as suggestion. According to your suggestion, we have unified the height-level of upper-troposphere in this study to 200 hPa (**Fig. 4 and 5**). Please see our response to your major comment 5.

5. Fig 6. Could you show the UDCZ variability from the observation data? This would be better to validate the changes between observation and model results as well as the history and future changes. A little confusing from the legend of panel b, it is the differences between two periods (200-2070 vs 1900-1970), but it shows a correlation map. How about the significance?

Response: Done as suggestion. For **fig.7a** (the original **fig. 6a**), we have added the UDZC calculated from the multi-source mean of reanalysis data (red curve in **Fig. R16a**).

Calculation of the multi-source mean is explained in Methods section:

L530–535: “The multi-source mean of Corr (WJA, EJA) shown as red curve in **Fig. 7a** is calculated by averaging it from all available data for each 21-year sliding window. For example, the value of Corr (WJA, EJA) centered at 1920 is obtained by averaging results from 20thCRV3, CERA-20C and ERA-20C, while that in 1980 is obtained by averaging results from all datasets (20thCRV3, CERA-20C, ERA-40, ERA5, JRA-55, ERA-Interim, CFSR, NCEP/NCAR-R1 and NCEP/DOE-R2).”

For **fig. 7b** (the original **fig. 6b**), we used the inter-model correlation map to better reflects the relationship between UDZC and future warming in CMIP6 outputs.

We have added more details of calculation the inter-model correlation map in “Methods” section and updated the legend of fig. 7b:

L459–468: “In **Fig. 7b**, the inter-model Corr (future UDZC, future warming amplitude) reflects the inter-model relationship among 42 CMIP6 models. Specifically, we first calculate the future UDZC and future warming amplitude in

each model: here the future UDZC is the correlation between WJA and EJA during 2000–2070, whereas the future warming amplitude is calculated as the climatology changes in surface air temperature during future period (2000–2070) relative to historical period (1900–1970). And thereby we can obtain 42 pairs of future UDZC and future warming amplitude for each grid. Then, the Corr (future UDZC, future warming amplitude) is calculated as the Pearson correlation coefficient between the two among 42 models. The significance the inter-model correlation is indicated with hatching, after controlling for the false discovery rate ($\alpha_{FDR} = 0.10$).

L906–909: “**b** Map of inter-model correlation between future surface warming amplitude and future UDZC of Eurasian jet (see “Model simulation data” in Methods for details), with statistically significant values after controlling for the false discovery rate ($\alpha_{FDR} = 0.10$) hatched.”

Fig. R16 (Figure 7 in revision) | Future change in UDZC of Eurasian jet. **a** On the left is the evolution of Corr (WJA, EJA) in a 21-year sliding window, including results from the multi-source reanalysis data-based observation mean (red) and from the selected CMIP6 models (green) and LENS2 members (purple) according to Supplementary Figure 11. This figure focuses on the evolution of interannual relationship between WJA and EJA in the context of history and

future warming scenario. The solid green/purple lines and shading indicate multi-model mean and inter-model spreading, respectively. On the right is the multi-model mean of Corr (WJA, EJA) during historical period (1900–1970) and future period (2000–2070). **b** Map of inter-model correlation between future surface warming amplitude and future UDZC of Eurasian jet (see “Model simulation data” in Methods for details), with statistically significant values after controlling for the false discovery rate ($\alpha_{\text{FDR}} = 0.10$) hatched. Red and blue boxes outline the two hot-spots of future warming regions over the represented land (20°N–60°N, 0–70°E) and ocean (15°N–55°N, 170°E –130°W), respectively. **c–d** Inter-model relationship of future UDZC and future surface warming over the hotspot of land and ocean. The thick red dash line represents the ordinary least-squares fit, with shading as the 95% confidence interval.

Response to reviewer #3:

Review for: Emergence of unprecedented zonal-consistent variation in Eurasian jet axis

This paper introduces a new index tracking the strength of the 200-hPa zonal wind (U200) jet across '40°E–80°E for the west part of Eurasian jet axis (WJA index), 90°E–123°E for the east part (EJA index)' is defined as the 'upstream-downstream zonal consistency' (UDZC) of the jet, and then tracks its variability in several datasets - with substantial differences in the index amongst them. The motivation is to better understand Eurasian weather/climate extremes, although the UDZC index defined is not actually shown to be related to them. Shifts in the index are found throughout its record, including near the start of the satellite record, and given the location of the study as potentially sparse with data (across western, northern Asia, for instance) means that data quality could be a source of the 'profound shift in the UDZC' that is reported.

A different index, the 'Eurasian subtropical westerly jet' (ESWJ) index, is subsequently defined using the second principal component of EOF analysis performed on U200 for the period 1999-2022. It is somehow – rather uncertainly – linked to UCZC, but how is not quite clear. The ESWF is loosely related to shifts in heatwaves, the standardized precipitation index and precipitation, but with 90% confidence that suggests a less robust relationship than hoped for. It also is unclear why the UDZC is relevant or introduced when it is not the index that is related to extremes. It is unclear what is physically meaningful about the UDZC or ESWJ indices defined, which are inferred as the 'cause' of weather and climate extremes when they are, rather, symptomatic of the conditions conducive to extremes. The jet and the eddies producing extremes coevolve, and tracking one is to track the other. As such, the statistical analysis regarding 'abrupt' but probably random shifts in these vaguely-defined quantifies is likely being overinterpreted. It is the same as concluding that there are random shifts in the waves and extremes related to them.

While it is possible that there is some dynamical relevance to the UDZC, the methodology used is subject to many limitations whose mention are missing in the text, such as likely presence of spurious EOFs that are poorly resolved, thus accounting for their shifting between P1 and P2. Much more robust analysis is needed to draw a the conclusion that a random shift in UDCZ represents a ‘new paradigm of Eurasian circulation’, particularly since UDCZ’s dynamical relevance is not shown.

It does not seem, to me, that the results from this analysis are always reliably interpreted, such as this shift in UDZC representing a ‘with a brand-new paradigm of Eurasian upper-level circulation’ as stated in the Discussion. The null hypothesis that this shift is random is not robustly interrogated to draw the conclusion that the Eurasian circulation has dramatically shifted due to shifts in the indices used here. So much further information is needed to support the claims put forth that that the results and significance do not seem substantial or robust enough for Nature Communications.

Response: Thank you for your valuable time and for providing such insightful and constructive feedback on our manuscript. We appreciate the reviewer's careful reading and agree that the comments have helped us identify key areas for improvement.

We have undertaken a thorough revision of the manuscript to address all the points raised. The major changes include: (1) clarifying the methodological justification and physical meaning of our indices, (2) strengthening our statistical analysis by adopting a stricter significance level, (3) explaining the relationship between jet and eddies, (4) discussing on robustness of such “abrupt shift”, (5) performing additional robustness checks for our EOF analysis.

- (1) We have added more explanation about the physical mean of UDZC and its quantification, including the correlation between WJA and EJA indices, and the interannual UDZC index (the original ESWJ index; defined by the projection of U200 during 1958–2022 on the EOF1 of P2), please see our response to your major comment 1.

- (2) For **Figure 3–5**, we have strengthened our statistical analysis by adopting a stricter significance level ($\alpha = 0.05$), which shows similar results with the previous significance level ($\alpha = 0.10$), as **Fig. R1–3** shows.
- (3) For the co-evolution of the jet and synoptic eddies, we agree that it is a fundamental principle of atmospheric dynamics. However, we respectfully argue that this classic relationship, which is most characteristic of the eddy-driven polar front jet¹⁻³, is less applicable to our specific subject of study: the Eurasian subtropical jet during the boreal summer⁴. Since previous studies suggest that the eastern part of Eurasian jet in summer is mainly maintained by the strong high-pressure: Tibet Plateau High, a system maintained by the intense sensible heat from the Tibet Plateau³⁻⁵; while the transient eddies are the weakest during summer⁶. This entire mechanism is indicative of a process dominated by thermal-dynamic coupling rather than transient eddy forcing. Therefore, the influence of Eurasian jet on synoptical eddies is likely to be stronger than feedback from eddies.

In addition, the changing strength of Eurasian jet is capable of influence the region of eddy growth rate maximum, which further influence the strength of polar jet stream⁷.

References:

1. Xu, P., Wang, L. & Chen, W. The British–Baikal Corridor: A Teleconnection Pattern along the Summertime Polar Front Jet over Eurasia. *J. Clim.* **32**, 877–896 (2019).
2. Stendel, M., Francis, J., White, R., Williams, P. D. & Woollings, T. The jet stream and climate change. in *Climate Change 327–357* (Elsevier, 2021). doi:10.1016/B978-0-12-821575-3.00015-3.
3. Ossó, A. et al. Advancing Our Understanding of Eddy-driven Jet Stream Responses to Climate Change – A Roadmap. *Curr Clim Change Rep* **11**, 2 (2024).
4. Zhang, Y., Kuang, X., Guo, W. & Zhou, T. Seasonal evolution of the upper-tropospheric westerly jet core over East Asia. *Geophys. Res. Lett.* **33**, 2006GL026377 (2006).
5. Kuang, X. & Zhang, Y. Seasonal variation of the East Asian Subtropical Westerly Jet and its association with the heating field over East Asia. *Adv. Atmos. Sci.* **22**, 831–840 (2005).

6. Ren, X., Yang, X. & Chu, C. Seasonal Variations of the Synoptic-Scale Transient Eddy Activity and Polar Front Jet over East Asia. *J. Clim.* **23**, 3222–3233 (2010).
7. Lee, S. & Kim, H. The Dynamical Relationship between Subtropical and Eddy-Driven Jets. *J. Atmos. Sci.* **60**, 1490–1503 (2003).

And we have added more explanation about the relationship between jet and synoptic eddies:

L114–122: “Apart from the thermal-driven Eurasian jet^{23,29}, there is another jet stream over the Eurasian high-latitudes which is called the polar jet or the eddy-driven jet^{23,30}. It grows at the high-latitudes due to the active baroclinic eddies *in-situ* and is much weaker than the subtropical Eurasian jet³¹. Previous study has revealed the impact of subtropical jet to polar jet, using an idealized multilevel primitive equation model²³. It is found that the position where the baroclinic wave grows is modulated by the strength of subtropical jet stream. Therefore, a weak Eurasian jet favors development of baroclinic wave at its 20° to 30° poleward side, leading to a strengthened eddy-driven polar jet. The out-of-phase relationship is related to the atmospheric blocking with the phenomenon called double jets³, favoring the occurrence of WCEs.”

References :

3. Rousi, E., Kornhuber, K., Beobide-Arsuaga, G., Luo, F. & Coumou, D. Accelerated western European heatwave trends linked to more-persistent double jets over Eurasia. *Nat. Commun.* **13**, 3851 (2022)
23. Lee, S. & Kim, H. The Dynamical Relationship between Subtropical and Eddy-Driven Jets. *J. Atmos. Sci.* **60**, 1490–1503 (2003).
29. Ren, X., Yang, X. & Chu, C. Seasonal Variations of the Synoptic-Scale Transient Eddy Activity and Polar Front Jet over East Asia. *J. Clim.* **23**, 3222–3233 (2010).
30. Xu, P., Wang, L. & Chen, W. The British–Baikal Corridor: A Teleconnection Pattern along the Summertime Polar Front Jet over Eurasia. *J. Clim.* **32**, 877–896 (2019).
31. Panetta, R. L. & Held, I. M. Baroclinic Eddy Fluxes in a One-Dimensional Model of Quasi-geostrophic Turbulence. *J. Atmos. Sci.* **45**, 3354–3365 (1988).

- (4) We present multiple lines of evidence to demonstrate that the recent surge in the UDZC of the Eurasian jet strength is robust and unprecedented in the historical record. Please see response to major comment 2 for detail.
- (5) We have further investigated the first five leading modes (EOF1–5) of Eurasian jet using multi-variable EOF (MVEOF), with robustness checks according the North’s rule (**Fig. R8** and **R9**). Our results further confirm the uniqueness of the “zonal consistent intensity change” mode, since such mode of change in the whole Eurasian jet is not found in EOF1–5 during P1, which totally account for 68.2% of Eurasian jet variability. In addition, from vertical section of zonal wind, we have shown that the “intensity change” mode is related to the change of zonal wind throughout the troposphere; and MVEOF analyses on meridional circulation throughout the troposphere is performed to further demonstrate that such “zonal consistent intensity change” mode during P2 is the leading feature of Eurasian circulation (**Fig. R10**). Please see our response to your major comment 3 and minor comment 1 for details.

Fig. R1 (Figure 3 in revision) | Impacts of enhancing UDZC on Eurasian WCEs. **a** Regression map of NET-heatwave frequency against the UDZC index during P1, with 95% confidence hatched. **b** and **c** Similar to **a**, but for the SPEI and rainfall anomalies, respectively. **d–f** Similar to **a–c**, but for the P2. The purple line indicates the climatology of jet stream axis. Note that here the rainfall data over land and ocean adopted the NOAA Land and GPCP precipitation, respectively. So, in **c**, the GPCP over ocean is during 1979–1998.

Fig. R2 (Figure 4 in revision) | Regime shifts linked to enhancing UDZC of the Eurasian jet axis. **a** Regression map of 200-hPa Local Wave Activity (LWA) against the UDZC index during P1, with 95% confidence hatched. Also shown are the composite pattern of Tibet Plateau High (TPH; blue; measured by H200 in 12530 gpm) and western North Pacific subtropical high (WNPSH; red; measured by H500 in 5870 gpm). Shadings represent their climatology with thick solid (dash) lines represent their mean states in strong (weak) jet years (see Methods section for detail), respectively. **b** Correlation map of tropical cyclone (TC) track density (shading; 95% confidence enclosed by red line) against the UDZC index during P1. Vectors represent the regressed vertical-integrated water vapor flux (WVT), with 95% confidence in blue color. **c–d** Same as **a–b**, but for P2. The purple line indicates the climatology of jet axis.

Fig. R3 (Figure 5 in revision) |Circumglobal impacts of strong UDZC in Eurasian jet. a Regression map of H200 (shading; units: gpm) and V200 (contour; units: m s⁻¹) against the UDZC index during P2, with 95% confidence hatched. Vectors represent the regressed wave activity flux (WAF; units: m⁻² s⁻²) at 200 hPa. **b** Regression map of NET-heatwave frequency (shading; units: days month⁻¹) over western Europe against the UDZC index during P2, with 95% confidence stippled. Contours represent the regressions of SPEI anomalies, with 95% confidence hatched. **c** Similar to **b**, but for the North America.

Major Comments

1. What UDZC physically represents remains elusive throughout the text. It seems that two areas of high zonal wind variance are tracked and that it happens to have had a strong difference in the year 2022. Why this is important is not, however, demonstrated by any of the present analysis. What does a positive value represent in terms of the jet, in absolute terms? A negative value? These basic pieces of information are crucial to be convinced that UDZC warrants any consideration, and the subsequent analysis about its variability over a long period of record does not seem much of interest.

Response: Thank you for raising this important point. In this study, we propose the concept of UDZC to quantify the degree of zonal consistency in the strength variation of Eurasian jet.

To clarify, our analysis of UDZC involves two distinct but related components to capture its characteristics on decadal and interannual timescales. The strengthening co-variability of WJA and EJA since late-1990s has led to a strong UDZC state at decadal scale (**Fig. 1c**) and enhanced variability of UDZC index at interannual scale (**Fig. 2e**). Not only the year 2022, but also years of 1999, 2000, 2003, 2008, 2009, 2013, 2015 exhibit strong anomalies than any year during P1.

First, to represent UDZC state for a specific period, we use the correlation coefficient between the WJA and EJA indices. We further employ a sliding window analysis for to investigate the temporal evolution of this correlation (**Fig. R4c**). Therefore, results presented in **Fig. R4c** describe the interdecadal evolution of UDZC state. A significant positive value in the sliding correlation for a given period indicates the presence of a strong UDZC regime during that time. This is how we identified the period after 1999 as an era of strong UDZC. The datasets with longer period provide a more comprehensive information about the evolution of UDZC, reveals that a strong UDZC state is intuitively not a common phenomenon in history, and helps confirm that the strong UDZC after 1999 is a state unprecedented in the last two centuries.

Second, since the different part of Eurasian jet can influence surface climate *in-situ*¹⁻⁵, the more zonally extended pattern strongly implies a more synchronized WCEs across Eurasia. Therefore, to investigate the interannual relationship between the Eurasian jet and climate within periods of different UDZC states, we needed a year-to-year index. To achieve this, we first used EOF analysis on the post-1999 period to capture the strong UDZC mode (**Fig. R5c**; EOF1 in P2), also referred to as the "intensity change" mode. To derive the interannual variability of this mode across the entire study period (1958–2022), we projected the full U200 dataset onto this spatial pattern. This projection yields the interannual UDZC index (**Fig. R5e**; originally named the ESWJ index).

For this interannual UDZC index, a strong positive (negative) value corresponds to a consistently strengthening (weakening) of both the upstream and downstream

sections of the Eurasian jet in that year, while a value near zero indicates either a weak overall change in the jet's intensity or an out-of-phase variation between the upstream and downstream parts. Therefore, the significantly larger variance of this index after 1999 reflects the fact that consistently strengthening or weakening of the jet's upstream and downstream sections became the dominant mode of year-to-year variability during this strong UDZC era.

References:

1. Fu, Z.-H., Zhou, W., Xie, S.-P., Zhang, R. & Wang, X. Dynamic pathway linking Pakistan flooding to East Asian heatwaves. *Sci. Adv.* **10**, eadk9250 (2024).
2. Li, X., Zheng, J., Wang, C., Lin, X. & Yao, Z. Unraveling the roles of jet streams on the unprecedented hot July in Western Europe in 2022. *npj Clim Atmos Sci* **7**, 323 (2024).
3. An, X. et al. Record-breaking summer rainfall in the asia–pacific region attributed to the strongest asian westerly jet related to aerosol reduction during COVID-19. *Environ. Res. Lett.* **18**, 074036 (2023).
4. Chowdary, J. S. et al. Meridional displacement of the Asian jet and its impact on Indian summer monsoon rainfall in observations and CFSv2 hindcast. *Clim. Dyn.* 1–19 (2021).
5. Wang, X., Lu, R. & Hong, X. Reduction of mid-summer rainfall in northern India after the late-1990s induced by the decadal change of the Silk Road pattern. *Environ. Res. Lett.* **16**, 104051 (2021).

We have added more explanation to clarify the concept of UDZC in manuscript:

Lines 73–76: “There are two prominent activity centers over its western and eastern part at 200-hPa zonal wind (U200; contours in **Fig. 1a**). The UDZC of Eurasian jet during a specific period can be simply quantified by the co-variation between these two activity centers.”

Lines 82–86: “Here, we use the correlation coefficient (R) between these two indices to quantify the state of UDZC in a specific period (see “Connotation and Definition of UDZC” in Methods). A positive and robust R indicates strong UDZC state during this period, while feeble or even negative R represents weak UDZC state. Therefore, the interdecadal evolution of UDZC state can be

simplified as the changing R between WJA and EJA indices in a sliding window (**Fig. 1c**).”

Lines 173–185: “Since the R between WJA and EJA indices can only quantify the UDZC state in a specific period instead of in a year, a new interannual UDZC index is needed. To obtain the year-to-year variability of UDZC, we performed the projection of U200 during 1958–2022 onto the “intensity change” mode (**Fig. 2c**), with the resultant time series named as the UDZC index (**Fig. 2e**; see “Jet stream indices” in Methods). The strong positive (negative) value of this index is related to the uniformly strengthening (weakening) of the whole Eurasian jet strength. Therefore, this index exhibits very weak variability during period of weak UDZC state (P1), especially after 1979 due to the negative relation between WJA and EJA, but shows a sudden increase in variability after 1999 (**Fig. 2e**), consistent with the strong UDZC state during this period (**Fig. 1c**). Therefore, the strengthening co-variability of WJA and EJA since late-1990s has led to a strong UDZC state at decadal scale (**Fig. 1c**) and enhanced variability of UDZC index at interannual scale (**Fig. 2e**). Not only the year 2022, but also years of 1999, 2000, 2003, 2008, 2009, 2013, 2015 exhibit strong anomalies than any year during P1.”

Lines 519–547 (“Connotation and Definition of UDZC” in Methods): “First, to represent UDZC state for a specific period, we use the correlation coefficient between the WJA and EJA indices, i.e., $\text{Corr}(\text{WJA}, \text{EJA})$. We further employ a sliding window analysis for to investigate the temporal evolution of $\text{Corr}(\text{WJA}, \text{EJA})$. Therefore, results presented in **Figure 1c** describe the interdecadal evolution of UDZC state. A significant positive value in the sliding correlation for a given period indicates the presence of a strong UDZC regime during that time. To check the robustness and long-term evolution of UDZC, the 21-year sliding correlation is calculated using multiple datasets (**Fig. 1c**), including 20thCRV3, CERA-20C, ERA-20C & ERA-40, ERA5, JRA-55, ERA-Interim, CFSR,

NCEP/NCAR-R1 and NCEP/DOE-R2. Note that here the “ERA-20C & ERA-40” represents a combination of ERA-20C (1901–1957) and ERA-40 (1958–2002): Namely, these two datasets were merged to form a continuous record spanning the period 1901–2002. The multi-source mean of Corr (WJA, EJA) shown as red curve in **Fig. 7a** is calculated by averaging it from all available data for each 21-year sliding window. For example, the value of Corr (WJA, EJA) centered at 1920 is obtained by averaging results from 20thCRV3, CERA-20C and ERA-20C, while that in 1980 is obtained by averaging results from all datasets (20thCRV3, CERA-20C, ERA-40, ERA5, JRA-55, ERA-Interim, CFSR, NCEP/NCAR-R1 and NCEP/DOE-R2).

Second, to investigate and contrast the interannual relationship between the Eurasian jet and climate within periods of different UDZC states, a uniformly defined UDZC index is therefore needed to describing year-to-year variability of UDZC especially for pre-1999 during which the EOF cannot capture the UDZC mode. To achieve this, we first used EOF analysis on the post-1999 period to capture the strong UDZC mode (EOF1 in P2; **Fig. 2c**), also referred to as the “intensity change” mode. To derive the interannual variability of this mode across the entire study period (1958–2022), we project the U200 anomalies onto the EOF1 mode during P2 (regression pattern as contours shown in **Fig. 2c**) over the EOF region (i.e., blue box in **Fig. 2c**). This projection yields the interannual UDZC index for 1958–2022. In other words, this uniformly defined interannual UDZC index can directly reflect the strength of UDZC in a specific year. For example, a strong positive (negative) value reflects to the degree of consistent enhancement of the upstream and downstream sections of the Eurasian jet.”

Fig. R4 (Figure 1 in revision) | Unprecedented surge in UDZC of the Eurasian jet axis. **a** Climatology of the Eurasian jet (green shading) and the standard deviation of U200 (contours; units: m s^{-1}). **b** Year-to-year variability of WJA and EJA indices, with R1 and R2 represent the correlation coefficient between the two indices during P1 (1958–1998) and P2 (1999–2022). **c** The 21-year sliding correlation of WJA and EJA indices in multi-source data. The gray dash line indicates the 95% (99.9%) confidence level.

Fig. R5 (Figure 2 in revision) | Shifting dominant mode of Eurasian westerly. **a** and **c** The EOF1 mode of U200 (shading: correlation map) during P1 (1958–1998) and P2 (1999–2022), respectively. Black lines outline regions exceeding the 95% confidence level, with that after controlling for the false discovery rate ($\alpha_{\text{FDR}} = 0.10$) stippled. The red line indicates the climatology of jet stream axis. The blue box denotes the regions for performing EOF. **b** and **d** Same as **a** and **c**, but for EOF2. Shown in **e** are the normalized UDZC index (black bar) and the PCs corresponding to EOFs in **a–d**, as well as the WJA (blue curve) index and the EJA (red curve) index during P2. **f** Correlations between U200 PCs and UDZC, WJA and EJA indices for P1 and P2.

- Figure 1: Given limited in situ data in this region prior to the satellite era, and the lack of agreement between datasets prior to that time – and even during it for some estimates – it seems that data quality could preclude drawing robust conclusions about the multidecadal variability of this quantity. And, importantly, there is no information shown about the pattern of the jet during high or low UDZC values, which is crucial to understand what this index represents. The

subsequent relation to the EOFs of u200 and their PCs, below, is therefore nebulous regarding the relation to UDCZ – which is itself unclear.

Response: Thank you very much for your valuable and constructive feedback. We would like to offer a detailed response to your concerns regarding data quality, the physical meaning of the UDZC index, and its connection to the subsequent EOF analysis.

(1) For your concern on the reliability of data prior to the satellite era, we fully agree with your point that the quality and consistency of reanalysis data are subject to greater uncertainty in the pre-satellite era due to the scarcity of observational data, especially in situ measurements. It was precisely to address this uncertainty that we used nine different mainstream reanalysis datasets for cross-validation in our study (**Fig. R4c**; also **Figure 1c** in manuscript).

Although there are some discrepancies in the exact correlation values (UDZC) before the 1970s, a core finding is highly consistent across all datasets: for most of the 20th century, the correlation between WJA and EJA was very low and often insignificant.

To specifically address the possibility of random, long-term fluctuations, we further used the 20thCRV3 reanalysis, which 80 members extend back to 1806. This longer-term perspective confirms that the strong UDZC observed in recent decades is "an unprecedented phenomenon over the past nearly two centuries".

Our main argument is not to quantify the exact UDZC value, but to demonstrate that the recent surge in UDZC is unique over the past two centuries. This significant strengthening, which starts in the late 1990s, is exceptionally clear and consistent across all datasets.

Furthermore, this historically weak UDZC is supported by state-of-the-art climate models (**Fig. R6**). Both the CMIP6 multi-model ensemble and the LENS2 large ensemble show a generally weak UDZC in their historical

simulations (1900–1970), with correlation coefficients typically below 0.3. This agreement between observational reanalysis and model ensembles suggests that a weak zonal consistency is the normal state for the Eurasian jet. This, in turn, further highlights the truly unique and unprecedented nature of the strong UDZC observed in recent decades, which is the central focus of our paper.

We have added more explanation about the data quality in the manuscript:

L88–93: “Since our result before the satellite era can be greatly influence due to the scarcity of observational data, especially *in-situ* measurements, we choose to use multiple reanalysis data for cross-validation (see “Observation data” in Methods). Although there are some discrepancies between different datasets before satellite era, a core finding is highly consistent across all datasets: strong UDZC state in the Eurasian jet has been rare throughout the past century in all reanalysis data (**Fig. 1c**).”

L97–99: “The strong consistence of result after the late-1990s between datasets supports the robustness of the sudden emergence of strong UDZC state in Eurasian jet (**Fig. 1b**).”

L323–325: “The weak UDZC in models also supports our finding that strong UDZC state is not a common phenomenon, emphasizing the uniqueness of the emergence of strong UDZC since the late-1990s.”

- (2) We have drawn the spatial pattern of U200 related to the interannual UDZC index during P1 and P2 (**Fig. R7**). During P1, where the R between WJA and EJA is low, the UDZC-related U200 pattern is similar to that of WJA index, indicating a strengthening and poleward shift of WJA with relatively weak anomalies around the EJA. During P2, where the R between WJA and EJA is high, the UDZC-related U200 pattern is identical to the EOF1 during P2 (**Fig. 2c**); the strong and zonal consistent westerly wind anomalies along the jet axis

represents strong UDZC. The relation between the UDZC and the EOFs is also explained in our response to your major comment 3.

We have added more discussion about the spatial pattern of Eurasian jet during two sub-periods in the manuscript:

L194–199: “We also drew the spatial pattern of U200 related to the interannual UDZC index during P1 and P2 (Supplementary Figure 8). During P1, the UDZC-related U200 pattern is similar to that of WJA index, indicating a strengthening and poleward shift of WJA with relatively weak anomalies around the EJA (Supplementary Figure 8a). During P2, the UDZC-related U200 pattern is identical to the EOF1 during P2 (**Fig. 2c**); the strong and zonal consistent westerly wind anomalies along the jet axis represents strong UDZC (Supplementary Figure 8b).”

Fig. R6 (Supplementary Figure 11 in revision) | Future change of UDZC in Eurasian jet in high summer. **a** Comparison of the Correlation coefficient between WJA and EJA indices in the history (1900–1970; blue bars) and future (2000–2070; orange bars) climate in 50 CMIP6 models. Models (members) simulate a significant increase in UDZC of Eurasian jet are shows in grey background color. The multi-model means of selected models, unselected models and all models are also shown in grey (history) and orange (future) bar. **b** Same as a, but obtained from 50 members of CESM2 Large Ensemble (LENS2). Error bars in the multi-model mean (ensemble mean) are SD of 10000 realizations (see “Bootstrap test” in Methods).

Fig. R7 (Supplementary Figure 8 in revision) | Different spatial pattern of interannual UDZC index during P1 and P2. a–b UDZC related U200 anomalies (shading: correlation map; contour: regression map, units: m s^{-1}) at P1(a) and P2 (b), respectively, with statistically significant values after controlling for the false discovery rate ($\alpha_{\text{FDR}} = 0.1$) hatched.

3. Figure 2: it is possible that the difference in pattern between period 1 and period 2 simply suggests that the variability in the region is poorly resolved with EOFs. Are the EOFs statistically distinct from one another according to North's rule (1982)? How do the EOF patterns manifest in the full-field zonal wind? I don't see any relation between the UDZC and the EOFs, in part because so little information about what the UDZC represents is provided.

G.R. North, T.L. Bell, R.F. Cahalan, and F.J. Moeng. (1982). Sampling errors in the estimation of empirical orthogonal functions. *Mon. Wea. Rev.*, 110:699-706.

Response: Thank you for your valuable and constructive comments. We fully agree that examining the statistical separability of the EOF modes is crucial. Following your suggestion, we have performed North's rule of thumb test for the leading five EOF modes in both periods (**Fig. R8** and **R9**). Note that here we performed a multi-variable EOF (MVEOF) combining U200 and zonally averaged vertical section of zonal wind from 500 hPa to 100 hPa, to better reflect the dominant modes of Eurasian jet.

The results show that for period P1 (1958–1998; **Fig. R8**), the first leading EOF modes are statistically distinct from other modes (explaining 20.41% of the variance), emphasizing the dominant role of Eurasian jet meridional shift (**Fig.**

R8a and **f**). In vertical structure, this mode reflects a clear dipole throughout the troposphere, with strong westerly anomalies in the south flank of Eurasian jet and relatively weaker easterly in its north flank (**Fig. R8a**, right). However, for period P2 (1999–2022), the first leading mode clearly represents the strengthening of the whole Eurasian jet (**Fig. R9a**, left), similar to **Fig. 2c**. It exhibits a clear tripolar structure throughout the troposphere, even reaching stratosphere (**Fig. R9a**, right). It can also be clearly separated from other modes, with explained variance reaching 23.28% (**Fig. R9a** and **f**; high than that of EOF1 during P1).

For the relation between UDZC and ‘intensity change’ mode, our manuscript defines UDZC as the "upstream-downstream zonal consistency" in the jet stream's strength alteration. A high UDZC implies that the WJA and EJA indices vary in unison (high positive correlation). Therefore, a strong UDZC mode should manifest as a zonal symmetrical U200 signal along the Eurasian jet. The EOF1 mode in P2 spatially manifests as a zonally uniform "meridional triple pattern" spanning the entire Eurasian continent (**Fig. R9a**). The physical meaning of this pattern is a simultaneous, continent-wide strengthening of the subtropical jet stream accompanied by a weakening of the polar jet and tropical easterly jet. This continent-wide, synchronized intensity change is the direct physical manifestation of a high UDZC in the spatial field. The high R between WJA and EJA is a result of the emergence of the ‘intensity change’ mode, indicating a high UDZC period. Furthermore, the interannual UDZC index is obtained by projecting the full U200 dataset onto this intensity change’ mode, as explained in our response to your major comment 1.

We have added more analysis about the relation between UDZC and the ‘intensity change’ mode:

L143–150: “The EOF1 mode in P2 spatially manifests as a zonally uniform "meridional triple pattern" spanning the entire Eurasian continent (**Fig. 2c**). The physical meaning of this pattern is a simultaneous, continent-wide strengthening

of the subtropical jet stream accompanied by a weakening of the polar jet and tropical easterly jet. This continent-wide, synchronized intensity change is the direct physical manifestation of a high UDZC in the spatial field. Therefore, this “zonal consistent intensity change” mode strongly reflects a high UDZC state in Eurasian jet intensity variation and aligns with the dynamical out-of-phase relationship between subtropical and polar jet stream^{23,24}”

References:

23. Lee, S. & Kim, H. The Dynamical Relationship between Subtropical and Eddy-Driven Jets. *J. Atmos. Sci.* **60**, 1490–1503 (2003).
24. Li, S., Sato, T., Nakamura, T. & Guo, W. East asian summer rainfall stimulated by subseasonal Indian monsoonal heating. *Nat. Commun.* **14**, 5932 (2023).

Fig. R8 | The first five leading EOF modes of Eurasian jet during P1. The EOF is performed on the combination of U200 (20°–65°N; 0°–150°E) and zonally averaged zonal wind (20°–65°N; 0°–150°E; 500 hPa–100 hPa). **a** The EOF1 mode of U200 anomalies (left; shading: correlation map; contour: regression map, units: m s^{-1}) and zonally averaged vertical section (right) during P1. Hatching indicates statistically significant values after controlling for the false discovery rate ($\alpha_{\text{FDR}} = 0.10$). The yellow line in left indicates the climatology of jet axis and yellow contours in right is the climatology of Eurasian jet at 20 and 25 m s^{-1} . **b–e** Same as a, but for the EOF2–5 mode. Box with purple dash line denotes the regions for performing EOF. **f** Percentage of explained variance for each mode. Error bar denotes the lower and upper test bounds according to North's rule. The error bar with star above denotes this mode is significantly separated to other mode.

Fig. R9 (Supplementary Figure 5 in revision) | The first five leading EOF modes of Eurasian jet during P2. Same as Fig. R8, but for P2.

Minor comments

- Line 87-88 Much more evidence of the dynamical importance and uniqueness of the UDZC is needed before asserting that it is the leading feature of all Eurasia circulation, where shifts in UDZC dictate circulation shifts. “The emergence of strong UDZC in Eurasian jet implies a notable change in the dominant mode of jet stream variations” is not a statement I agree with, given the analysis presented here.

Response: Thank you very much for your insightful comments.

For the first point, we agree that analysis in previous manuscript cannot demonstrate that the EOF results of upper-level circulation represent the all-Eurasian circulation. Therefore, we performed the MVEOF of zonal wind, as explained in response to your major comment 3. As **Fig. R9** shows, the EOF1 for P2 exhibit a clear tripolar structure identical to **Fig. 1c (Fig. R9a, left)**. From the vertical section, it clearly represents the strengthening of the whole Eurasian jet

throughout the troposphere (**Fig. R9a**, right), implying shift in the Eurasian circulation and strong impacts on surface climate. In addition, it is significantly separated from other modes (**Fig. R9f**), according to the North test, proving the statistical robustness of the “intensity change” mode.

In addition, we also performed the multi-variable EOF of meridional wind and vertical velocity throughout the troposphere to capture the dominant modes of meridional circulation over Eurasian continent (**Fig. R10**). The EOF1 of meridional circulation is similar to **Fig. R9a**, with their PC1s highly correlated ($R = 0.8, p = 9.5 \times 10^{-7}$). Therefore, those additional results can well explain that the strong UDZC mode is the leading feature of Eurasian circulation.

We have given more analysis to clarify and supplement our arguments in the revised version as follows.

Lines 156–172: “Nevertheless, it remains unclear that whether such change in U200 EOF mode actually represents change of the whole Eurasian circulation and influence the consequently near-surface climate. To further investigate the role of such “intensity change” mode in Eurasian circulation. We further performed the a multi-variable EOF (MVEOF) combining U200 and zonally averaged vertical section of zonal wind from 500 hPa to 100 hPa. The EOF1 exhibit a clear tripolar structure throughout the troposphere, even reaching stratosphere (Supplementary Figure 5a), which accounts for 23.28% of the total variance, significantly higher than that of EOF2 (16.55%; Supplementary Figure 5b). This mode is high identical to that obtained using only U200, suggesting that the “zonal consistent intensity change” mode is a phenomenon throughout the troposphere rather than a confined upper-level mode.

In addition, we also performed a MVEOF of meridional circulation (represented by the meridional wind and vertical velocity), its EOF1 exhibit a two-cell meridional circulation over south and north flank of Eurasian jet, with low-level southerly wind and northerly wind respectively.

The associated zonal wind anomalies also show a tripolar structure of zonal wind similar to EOF1 (Supplementary Figure 6a). Therefore, our results strongly suggest that the “zonal consistent intensity change” mode is the leading feature of Eurasian circulation throughout the troposphere.”

For the second point, we agree that the original wording might have been too direct and requires more detailed evidence to be fully convincing. Therefore, we have added more explanation on the possible influence of changing UDZC on Eurasian jet mode.

Lines 114–127: “Apart from the thermal-driven Eurasian jet^{23,29}, there is another jet stream over the Eurasian high-latitudes which is called the polar jet or the eddy-driven jet^{23,30}. It grows at the high-latitudes due to the active baroclinic eddies *in-situ* and is much weaker than the subtropical Eurasian jet³¹. Previous study has revealed the impact of subtropical jet to polar jet, using an idealized multilevel primitive equation model²³. It is found that the position where the baroclinic wave grows is modulated by the strength of subtropical jet stream. Therefore, a weak Eurasian jet favors development of baroclinic wave at its 20° to 30° poleward side, leading to a strengthened eddy-driven polar jet. The out-of-phase relationship is related to the atmospheric blocking with the phenomenon called double jets³, favoring the occurrence of WCEs.

As shown in Supplementary Figure 4, the strong UDZC state during P2 implies a strengthened mode with a zonally uniform “meridional triple pattern” spanning the entire Eurasian continent. Therefore, the emergence of strong UDZC state in Eurasian jet strength since the late-1990s may coincide or reflect a notable change in the dominant mode of jet stream variations, with consequent change in the circulation pattern across multiple latitudes.”

References:

3. Rousi, E., Kornhuber, K., Beobide-Arsuaga, G., Luo, F. & Coumou, D. Accelerated western European heatwave trends linked to more-persistent double jets over Eurasia. *Nat. Commun.* **13**, 3851 (2022).

23. Lee, S. & Kim, H. The Dynamical Relationship between Subtropical and Eddy-Driven Jets. *J. Atmos. Sci.* **60**, 1490–1503 (2003).
29. Ren, X., Yang, X. & Chu, C. Seasonal Variations of the Synoptic-Scale Transient Eddy Activity and Polar Front Jet over East Asia. *Journal of Climate* **23**, 3222–3233 (2010).
30. Xu, P., Wang, L. & Chen, W. The British–Baikal Corridor: A Teleconnection Pattern along the Summertime Polar Front Jet over Eurasia. *J. Clim.* **32**, 877–896 (2019).
31. Panetta, R. L. & Held, I. M. Baroclinic Eddy Fluxes in a One-Dimensional Model of Quasi-geostrophic Turbulence. *J. Atmos. Sci.* **45**, 3354–3365 (1988).

Fig. R10 (Supplementary Figure 6 in revision) | The first five leading EOF modes of meridional circulation (1000–100 hPa) during P2. **a–e** Shadings represents the correlation field of zonal wind with PCs with vectors representing the regressed vertical circulation against normalized PCs, with 95% confidence colored. Yellow contours represent the climatology of Eurasian jet at 20 and 25 m s⁻¹.

2. Line 332 ‘singular value’ not ‘singularly valuable’.

Response: Corrected, thanks.

3. Line 342 “Strong (weak) Eurasian jet year are selected as the year whose corresponding value of ESWJ index is above + 0.5 (below -0.5)” this is a rather low threshold for standardized values, isn’t it?

Response: Thank you very much for your valuable and constructive comment.

Our choice of the ± 0.5 standard deviation (SD) threshold was primarily guided by

ensuring a sufficient sample size. Our study divides the data into two periods: P1 (1958–1998) and P2 (1999–2022). For the P2 period, which spans only 24 years, there are only 6 strong Eurasian jet year and 6 weak Eurasian jet year on the choice of the current ± 0.5 SD threshold (**Table R1**). Applying a stricter threshold such as ± 1.0 SD would result in too few years for the composite analysis (4 strong jet year and 3 weak jet year). To address your concern and test the robustness of our conclusions, we have also performed a sensitivity analysis using a stricter threshold of ± 1.0 SD (**Fig. R11**). The results show that the synergistic variation of the Tibet Plateau High and western North Pacific subtropical high remain highly consistent with those obtained using the ± 0.5 SD threshold, with more apparent difference between strong and weak jet years. This confirms that our core findings are not sensitive to the specific threshold chosen.

Lines 488–494: “This threshold was chosen to ensure a sufficient sample size for a statistically robust composite analysis, particularly for the shorter P2 (1999–2022) period. During P1, there are 14 strong Eurasian jet years and 14 weak Eurasian jet years. During the P2, there are six strong Eurasian jet years and six weak Eurasian jet years (See Supplementary Table 2 for detail). A sensitivity analysis using a stricter threshold of ± 1.0 standard deviation confirmed that the main conclusions of this study are robust (Supplementary Figure 9).”

Table R1 (Supplementary Table 2 in revision) | Selected strong/weak Eurasian jet years in two sub-periods. Strong (weak) Eurasian jet year are selected as the year whose corresponding value of UDZC index is above + 0.5 (below -0.5).

	Strong Eurasian jet years	Weak Eurasian jet years
P1	1959, 1961, 1963, 1964, 1967, 1969, 1972, 1976, 1981, 1987, 1990, 1991, 1993, 1995	1958, 1962, 1965, 1966, 1968, 1970, 1974, 1975, 1978, 1984, 1986, 1989, 1996, 1997
P2	2003, 2006, 2009, 2013, 2020, 2022	1999, 2000, 2002, 2008, 2012, 2015

Fig. R11 (Supplementary Figure 9 in revision) | Regime shifts linked to enhancing UDZC of the Eurasian jet axis. a–b Same as Fig. 4a and c, but the strong (weak) jet years for calculating composite TPH and WNPSH is selected based on the threshold of 1SD.

4. Lines 347-350 “we calculate the jet stream axis index by averaging the 200-hPa zonal wind along the jet axis, which varies from year to year” – it’s not actually stated how the jet axis’s location is determined from year to year in the two longitude bands that are subsequently defined. This is a key part of the methods that needs explaining. In any case I don’t quite agree with the following assertion that differencing zonal wind between two locations along the jet core is ‘dynamical’ – perhaps if the locations were selected according to the convergence of the ageostrophic wind along the jet entrance and exit regions or something, dynamical would be a defensible description.

Response: Thank you for your insightful comments. We have added a detailed explanation in the "Jet stream indices" subsection of the Methods. We agree with your point that the term 'dynamical' is not precise in this context. Our original intention was to emphasize that our method adapts to the interannual latitudinal shifts of the jet, in contrast to methods using a fixed geographical area. We have made corresponding modification in the Methods as follows:

Lines 510–514: “To define the jet axis, we first identify, at each longitude within a latitudinal band (25°N–60°N), the latitude of the maximum 200-hPa

zonal wind speed. The line connecting these points of maximum wind speed across the longitudes constitutes the jet axis, which is determined in every specific year. Therefore, this method is capable of capturing the year-to-year variation in the core position of the jet stream.”

Lines 516–518: “This axis-following approach is designed to minimize the potential bias caused by the latitudinal shift of the jet stream in different years, offering an advantage over methods that use a fixed geographical area for averaging.”

Response to Reviewers' Comments

We thank the reviewers for their insightful comments and detailed suggestions with emphases on different aspects, which are helpful in improving the quality of manuscript. Accordingly, we have carefully revised the manuscript, taking all comments and suggestions into account in the revision.

Our point-by-point responses to these comments are summarized as follows. Meanwhile, we give the marked version (i.e., tracked-changes) of the manuscript. Note that the **Line numbers** in this response are corresponded to the **marked version** of the revised manuscript.

Response to reviewer #1:

I appreciate the authors for adequately addressing my previous questions and comments. I have no further questions and recommend accepting the manuscript.

Response to reviewer #2:

The manuscript has been improved after the revision; however, the concerns of Reviewer 3, especially for the stability and reliability of the shift of UDZC index are not fully resolved. Some of them should be further explored thoroughly.

Response: Thank you for your instructive comments on improving the detailed mechanism linking UDZC, TPH and WNPSH, as well as issues on the robustness of UDZC.

We noticed that the terminology of different UDZC indices (including the running correlation between WJA and EJA and the projected UDZC index) may cause confusion to readers. Therefore, we firstly revised the definition and quantification of the UDZC phenomenon and the Projected ESWJ index in Methods.

In the revised manuscript, the quantification of UDZC is the correlation between WJA and EJA, and the UDZC index refers to the running correlation (Fig. 1c). For the index by projecting U200 during 1958-2022 into the EOF1 during P2, we now simply named it as the “projected ESWJ index”, which better describe its nature as quantifying the change of the whole Eurasian jet strength.

To further explore and discuss the uncertainty and robustness of the UDZC index and projected UDZC jet index. We have conducted a series of sensitivity experiments, including the sensitivity to selection of WJA and EJA domain, choice of sliding window, determination of year of dividing sub-periods. Those sensitivity experiments further reinforce the robustness of the strong UDZC since the late-1990s. We also discuss the uncertainty from data selection and possible reason of poor performance of UDZC in CMIP6 and LENS2 outputs.

The quality of the manuscript has been greatly improved based on your comments. Please find our response below.

Major comments

1. Although the authors explained the dynamical mechanism of UDZC and WNPSH, and TP high pressure, a dynamic example analysis or composite analysis should be provided in the supplementary material to improve the explanation.

Response: Thank you for this constructive suggestion to improve the explanation of the dynamical mechanism. Previous studies have introduced a preliminary concept of the second circulation (**Fig. R1**). To further elucidate the dynamical link between the Eurasian jet UDZC, TPH, and the WNPSH, we have performed a composite analysis of the vertical cross-sections of circulation anomalies during strong UDZC years in P2 (**Fig. R2**).

The following composite analysis is also based on those selected years during P2 (Supplementary Table 2). Based on these composites, we propose a detailed dynamical mechanism involving the interplay between jet dynamics and topographic thermal forcing with a simple Schematic diagram of the secondary circulation (**Fig. R3**), which has been incorporated into the revised manuscript and supplementary text:

L245–251: “To show the changes in TPH and WNPSH related to the projected ESWJ index, we performed a composite analysis on the selected strong and weak Eurasian jet years for two sub-periods, respectively (Supplementary Table 3).

Detailed composite analyses confirm that this upper-level divergence, coupled with negative vorticity advection and ageostrophic flow at the south flank of jet exit, drives a closed meridional circulation cell that extends the high-pressure anomaly downward (see Supplementary Text S4 and Supplementary Figure 11 and 12 for more details).”

Supplementary Text S4 | Dynamical Mechanism Linking UDZC, TPH, and WNPSH: “To elucidate the physical processes linking the Eurasian jet UDZC with the coupling of the TPH and WNPSH, we performed a composite analysis of vertical circulation anomalies during the P2 period (Supplementary Figure 11). During high summer, East Asia is situated at the right exit region of the eastern part of ESWJ (also called East Asian jet). The region of enhanced westerly winds corresponds to significant upward motion, whereas the lower-latitude region dominated by easterly anomalies exhibits distinct subsidence (Supplementary Figure 11a and b).

Two primary dynamical mechanisms govern the associated secondary circulation. First, ageostrophic winds at the jet exit region typically induce upper-level convergence and subsequent subsidence on the equatorward side¹⁻² (the right exit region; Supplementary Figure 11c). Second, the region south of the jet axis is characterized by negative vorticity advection (Supplementary Figure 11d), favoring geopotential height rises and subsidence (Supplementary Figure 11a).

However, these dynamical factors interact with the distinct topographic effects of the Tibetan Plateau. An intensified jet enhances this warm advection from the Tibetan Plateau³, promoting downstream convection, precipitation, and low-level southerly convergence.

Consequently, a unique, closed secondary circulation is established south of the East Asian jet axis: (1) The strengthened jet intensifies the rainband, leading to strong rising motion and latent heat release. (2) This diabatic heating generates strong upper-level divergence (Supplementary Figure 11c). (3) Combined with the

negative vorticity advection at the jet exit, this divergence sustains a robust high-pressure anomaly in the upper troposphere (the eastward extended TPH). (4) Driven by the ageostrophic flow, air converges and sinks at lower latitudes, extending the high-pressure system downward to form a significant low-level anticyclonic anomaly (the westward extended WNPSH). (5) This low-level anticyclone suppresses local convection while enhancing the northward transport of moisture, which further fuels the rainband precipitation, thereby closing the positive feedback loop (Supplementary Figure 12).”

References:

1. Li, X., Zheng, J., Wang, C., Lin, X. & Yao, Z. Unraveling the roles of jet streams on the unprecedented hot July in Western Europe in 2022. *npj Clim. Atmos. Sci.* 7, 323 (2024).
2. Shou, S., Li, S., Shou, Y. & Yao, X. Front and jet stream. *An Introduction to Mesoscale Meteorology* 73–115 (Springer, Singapore, 2023).
3. Sampe, T. & Xie, S. Large-Scale Dynamics of the Meiyu-Baiu Rainband: Environmental Forcing by the Westerly Jet. *J. Clim.* 23, 113–134 (2010).

[REDACTED]

Fig. R1 | Distribution of the divergence of ageostrophic winds in entrance and exit regions of the jet stream, from Shou. et al. (2023)

Fig. R2 (i.e., Supplementary Fig. 11) | Meridional section of secondary circulation linking strengthened Eurasian jet to enhanced subtropical highs. a Composite difference of geopotential height anomalies (zonal mean from 90°N–135°N; shading; units: gpm) against the projected ESWJ jet index during P2, with 95% confidence stippled. Red vectors representing the composite difference vertical circulation against normalized PCs (meridional wind and vertical velocity). Deep pink (blue) contours represent the composite mean of Eurasian jet during strong (weak) jet year at 20 and 25 m s^{-1} .

Fig. R3 (i.e., Supplementary Fig. 12) | Schematic diagram of secondary circulation at East Asian landmass related to strengthened Eurasian jet at the south flank of Eurasian jet.

2. At present, the domain of EOFs covers the whole Eurasia continent (20-65N, 0-150E), and the UDZC index is based on the domain of WJA and EJA. How did you select the domains of WJA and EJA? Please clarify it. Why the domain of EOF and UDZC is different?

Response: Thank you for this insightful question, which allows us to clarify the physical basis for our domain selection.

The domains for the WJA (40°E–80°E) and the EJA (90°E–123°E) were chosen to specifically capture the two prominent centers of high interannual variability of the Eurasian summer jet stream. WJA and EJA are separated by 10 degrees of longitude (i.e., 80°E–90°E), and it is selected at the overlap of the maximum position of the climatological U200 and the minimum position of the U200 standard deviation of the jet stream axis (**Fig. 1a**).

The difference in the domains for the EOF analysis (0°–150°E) and the WJA/EJA indices is because they serve two distinct but complementary purposes. The EOF analysis is intended to objectively identify the dominant, continent-spanning modes of variability in the entire Eurasian jet system. By using a broad domain (0°–150°E), we allow the analysis to capture the full spatial structure of these modes without imposing any preconceived geographical limitations. This unbiased, large-scale approach is crucial for robustly identifying the new "zonal consistent intensity change" paradigm that is central to our findings.

In addition, we agree that the different domain selection of WJA and EJA can lead to uncertainty of the UDZC (correlation between WJA and EJA), which is also part of the concerns in the major comment 4. Therefore, a number of sensitivity test is performed to check whether our result is robust (**Table R1**). The results show that the strong UDZC is robust and not sensitive to the domain selection. For instance, the correlation remains near 0.8 even we extend the east domain of EAJ from 123°E to 140°E.

We have added a statement on the insensitivity of our result to the domain selection and a “WJA and EJA indices” section in Methods with more clear description:

L98–99: “Note that the results are not sensitive to the sliding windows (Supplementary Figure 1) and the domain selection of WJA and EJA indices (Supplementary Table 1).”

L449–459: “The WJA and EJA indices are defined as the mean of 200-hPa zonal wind (U200) anomalies along the Eurasian westerly jet axis spanning longitude 40°E–80°E and 90°E–123°E, respectively. This axis-following approach is designed to minimize the potential bias caused by the latitudinal shift of the jet stream in different years, offering an advantage over methods that use a fixed geographical area for averaging. There are two selection criteria for the longitude range are: (1) WJA and EJA are selected at the western (i.e., 40°E–80°E) and

eastern (i.e., 90°E–123°E) parts of the Eurasian westerly jet axis, respectively; (2) WJA and EJA are separated by 10 degrees of longitude (i.e., 80°E–90°E) where is determined to be in the overlap of the maximum position of the climatological U200 and the minimum position of the U200 standard deviation of the jet stream (Fig. 1a). Note that the main results are not sensitive to the domain selection of WJA and EJA indices (Supplementary Table 1).”

Table R1 (Supplementary Table 1 in revision) | A list of sensitivity test on the domain selection of WJA and EJA. The corresponding correlation coefficient during P1 (R1) and P2 (R2) are shown. The R with $p < 0.05$ is marked with bold and red color.

	WJA domain	EJA domain	R1	R2
Test 1	40°–80°E	90°–130°E	0.09	0.79
Test 2	40°–85°E	90°–130°E	0.11	0.82
Test 3	40°–80°E	90°–135°E	0.13	0.78
Test 4	40°–85°E	90°–135°E	0.15	0.82
Test 5	40°–85°E	90°–140°E	0.17	0.82
Test 6	35°–85°E	90°–140°E	0.18	0.81

3. The definition of the UDZC index is arbitrary and needs to be justified. The authors just project the U200 anomalies onto the EOF1 mode during the period 1999 to present, which is defined as the interannual variability of UDZC. This definition is arbitrary and lacks of mechanism because the EOF modes are different between the two sub-periods 1958-1998 and 1999-2022 (Figure 2). The authors consider that the EOF1 for the P1 and P2 represent different implications: EOF 1 of P1 represents the Eurasian jet meridional shift, whereas the EOF1 of P2 represents the strengthening of the whole Eurasian jet. This may further indicate that the EOF mode may strongly depend on the period we used and further affect the UDZC index. This should be further explained and explored. This is also a concern by reviewer 3 about the robustness of the shift or stability of UDZC, as well as the

definition of UDZC. Although the authors argue that “this research is not to quantify the exact UDZC value, but to demonstrate the recent surge in UDZC in recent two decades”, if the UDZC is not robust, the increase of UDZC is still unreliable. The authors should pay more attention to this.

Response: Thank you for this insightful and constructive comment. We agree that this critical point requires a more thorough justification and additional supporting evidence. In response, we firstly clarify the definition of UDZC to avoid potential misleading and seemly arbitrary to readers. Second, we have performed a number of sensitivity tests to explore the changing EOF modes due to changing period to address these concerns. Please see our response below:

For the first point, we agree that naming the projected index as the “interannual variability of UDZC” is confusing and would misleads readers for the meaning of UDZC. Therefore, we have revised our manuscript to avoid this misleading definition. In the revised manuscript, the quantification of UDZC is the correlation between WJA and EJA, and the UDZC index only refers to the running correlation (Fig. 1c). For the index by projecting U200 during 1958-2022 into the EOF1 during P2, we now simply named it as the “projected ESWJ index”, which better describe its nature as quantifying the change of the whole Eurasian jet strength.

We have revised the manuscript and added two sections in Method to clarify:

L174–177: “To characterize the year-to-year variability of the UDZC phenomenon, we performed the projection of U200 anomaly filed during 1958–2022 onto the UDZC-like mode shown in **Fig. 2c**. Then we name the resultant time series as the projected ESWJ index (**Fig. 3a**; see Methods for details).”

L460–466 (“Quantification of the UDZC” in Methods): “To straightforwardly characterize the changes in UDZC of the Eurasian westerly jet stream, here we quantify the UDZC phenomenon as the Pearson correlation (R) between WJA and EJA (e.g., **Fig. 1c**). Then we can simply divide the main study

period into weak and strong UDZC periods (i.e., P1 and P2), respectively. If the WJA-EJA covariance (i.e., R^2) of a period exceeds 30%, that period is identified as a strong UDZC period, and vice versa. Accordingly, P1 is the weak UDZC period (i.e., 1958–1998), whereas P2 is the strong UDZC period (i.e., 1999–2022).”

L467–478 (“Projected ESWJ index” in Methods): “Considering that robust UDZC phenomenon between WJA and EJA emerges only in recent decades, we can only obtain the interannual variability of this phenomenon decades ago via projecting the U200 anomaly field onto Fig. 2c. In other words, a uniformly projected ESWJ index is therefore needed to describing interannual variability of UDZC phenomenon especially for pre-1999 periods, when EOF cannot well capture such UDZC-like mode as shown in Fig. 2c. Accordingly, we project the U200 anomaly field across the entire study period onto the EOF1 mode during P2 (regression pattern as contours shown in Fig. 2c) over the EOF region (i.e., blue box in Fig. 2c). Thus, the projected ESWJ index is actually completely consistent with PC1 during the P2 period: A strong positive (negative) value reflects the degree of increased (decreased) consistency between the upstream and downstream sections of the Eurasian jet stream.”

For the second point, we agree that the EOF mode may strongly depend on the period used; therefore, choice on the period for P2 may affects the projected ESWJ index, leading to the uncertainty in this index. Therefore, we also conducted a number of sensitivity tests to explore the changing EOF modes due to changing definition period of P2 (**Fig. R4** and **R5**).

We have revised the manuscript and add a supplementary text to analyze the robustness of the UDZC-like mode with corresponding projected index.

L185–187: “Note that we have tested and confirmed that the UDZC-like mode is robust, not sensitive to the projection reference-period (see Supplementary Text S3 and Supplementary Figures 7 and 8).”

Supplementary Text S3 (in Supplementary Information)| Robustness of Eurasian jet “intensity change” mode with its projected ESWJ index: “The EOF mode may strongly depend on the period used. To explore such uncertainty, we conducted a number of sensitivity tests to investigate the changing EOF modes due to changing definition period of P2 (Supplementary Figure 7 and 8).

First, we choose two sub-periods of the original P2 (1999-2022): 1999-2016 and 2003-2016. For the period 1999-2016, the EOF1 still exhibit a structure highly identical to the EOF1 during 1999-2022, although with slightly weaker correlation at the low-latitudes (Supplementary Figure 7c). Similar results are also found for the periods 2003-2016, but the structure is not very well organized, possible due to the insufficient samples (only 14 years) to extract ideal modes (Supplementary Figure 7e, f).

We also extend the period to explore the dominant modes in a longer period (Supplementary Figure 8). For three tests (1993-2022, 1987-2022 and 1981-2022), the EOF1 modes actually represents the meridional shift of Eurasian jet. However, we found that this “intensity change” mode already exists as the EOF2 (Supplementary Figure 8a, c, e), due to the relatively smaller explained variance. For the period 1999-2022, the corresponding explained variance has increased to 26.40% during 1999-2022 and become the first leading mode (Supplementary Figure 7a). This confirms that the “intensity change” mode dominant the Eurasian jet variability only after the late-1990s.

In all sensitivity tests, the spatial patterns of such “intensity change” mode is quite stable and highly consistent; and all the corresponding projected ESWJ index is all highly correlated ($R > 0.96$; Supplementary Table 2). Therefore, the “intensity change” mode is stable and robust, as well as the projected ESWJ index.”

Fig. R4 (Supplementary Fig. 7 in revision) | Sensitivity tests of EOF mode that can reflect the UDZC phenomenon and corresponding projected ESWJ index to the period-choice of P2: Shorten the selected time period of P2.

Fig. R5 (Supplementary Fig. 8 in revision) | Same as in Supplementary Fig. X, except for lengthen the selected time period of P2.

Table R2 (i.e., Supplementary Table. 2) | Sensitivity test of projected ESWJ indices. Note that here all the correlations (R) are calculated for the projected ESWJ indices during 1958-2022, but obtained from different projection reference-period. In other words, the periods listed in this Table indicate the projection reference-period for the EOF mode that can reflect the UDZC phenomenon (see **Supplementary Figs. 7 and 8**). It is clear to see that all the correlations exceed 0.96 ($p < 0.001$), suggesting that the projected ESWJ index is not sensitive to the projection reference-period.

EOF mode, and the reference-period for the projected ESWJ	EOF2 1981-2022	EOF2 1987-2022	EOF2 1993-2022	EOF1 1999-2022	EOF1 1999-2016	EOF2 2003-2016
EOF2 1981-2022	1	0.989	0.997	0.985	0.983	0.985
EOF2 1987-2022	0.989	1	0.988	0.964	0.971	0.969
EOF2 1993-2022	0.997	0.988	1	0.992	0.980	0.979
EOF1 1999-2022	0.985	0.964	0.992	1	0.967	0.964
EOF1 1999-2016	0.983	0.971	0.980	0.967	1	0.995
EOF2 2003-2016	0.985	0.969	0.979	0.964	0.995	1

- The UDZC index is based on the correlation between EJA and WJA, whereas the correlations are based on the sample size (time periods or slide windows), and that from different datasets are not the same. These would result in a very large uncertainty of the UDZC index, for example, the domain selections of EJA and WJA, the dataset selection, and the correlation window. The correlation between EJA and WJA is not significant in most of the periods in both Figure 1, 7, Supplementary Figure 1, 2, and 14, from a statistical perspective, the relationship between EJA and WJA is random (not significant). This may be one of the concerns of the robustness of UDZC index. Furthermore, in Figure 7, the correlations of EJA and WJA are different between the model projection and

reanalysis datasets. Does this also suggest the possible unreliability of the UDZC index?

Response: We agree that a thorough assessment of uncertainty is crucial and not fully explored and discussed. We have conducted a number of analyses and discussions focusing on these sources of uncertainty:

- (1) For the uncertainty from the domain selections of EJA and WJA, we have conducted a number of sensitivity test to check whether our result is robust (Table R3). The results show that the strong UDZC is robust and not sensitive to the domain selection. Please see our response to your major comment 2.
- (2) For the uncertainty from the dataset selection, we have used the bootstrap method to calculate the inter-dataset uncertainty (shown as red shading in **Fig. R6a**). The shading is very small in recent two decades due to the highly consistency between modern datasets (e.g. the ERA-5, JRA-55, CFSR and NCEP/DOE-R2). In fact, since the satellite era, even since the late-1950s, the majority of uncertainty of dataset selection comes from two 20th century dataset (e.g. the 20thCRv3 and CERA-20C). Therefore, the strong UDZC since the late-1990s is still robust between a number of modern datasets.

We have added a discussion in the manuscript as following:

L383–391: “Although the strong UDZC phenomenon in recent decades is confirmed by the consistent strengthening UDZC in multiple reanalysis data, different data selection may also leads to the uncertainty in UDZC. Therefore, we calculated the inter-dataset uncertainty via the bootstrap method (shown as red shading in **Fig. 7a**). The shading is very small in recent two decades due to the highly consistency between modern datasets (e.g. the ERA-5, JRA-55, CFSR and NCEP/DOE-R2). In fact, since the satellite era, even since the late-1950s, the majority of uncertainty of dataset selection comes from two 20th century dataset (e.g. the 20thCRv3 and CERA-20C). Therefore, the strong UDZC since the late-1990s is still robust between a number of modern datasets.”

(3) For the uncertainty from correlation window, we also test the results with different sliding window (e.g. 17-year, 19-year, 23-year and 25-year; **Fig. R7**).

All results show a consistent turning point around the late-1990s and reaching ~0.8 at the recent two decades. We have revised the manuscript as following:

L98–99: “Note that the results are not sensitive to the sliding windows (Supplementary Figure 1) and the domain selection of WJA and EJA indices (Supplementary Table 1).”

(4) For the uncertainty from CMIP6 and LENS2 models, we agree that their non-uniform capability in simulating UDZC may cause uncertainty in the future projection.

We have revised the manuscript as follows, to acknowledge the potential uncertainty from CMIP6 and LENS2 modes:

L426–428: “Therefore, the future projection of UDZC may contains uncertainty induced from the poor simulation capability of CMIP6 and LENS2 modes, which should be interpreted with causation.”

Besides, The potential reasons for such non-uniform capability are further discussed in the following Response to your **Minor comments 4 (Page 20)**.

Fig. R6 (i.e., the revised Fig. 7) | Future change in UDZC of Eurasian jet. **a** On the left is the evolution of R(WJA, EJA) in a 21-year sliding window, including results from the multi-source reanalysis data-based observation mean (red) and from the selected CMIP6 models (green) and LENS2 members (purple) according to Supplementary Figure 15. This figure focuses on the evolution of interannual relationship between WJA and EJA in the context of history and future warming scenario. The solid green/purple lines and shading indicate multi-model mean and inter-model spreading, respectively. On the right is the multi-model mean of R (WJA, EJA) during historical period (1900–1970) and future period (2000–2070). **b** Map of inter-model correlation between future surface warming amplitude and future UDZC of Eurasian jet (see “Model simulation data” in Methods for details), with statistically significant values after controlling for the false discovery rate ($\alpha_{FDR} = 0.10$) hatched. Red and blue boxes outline the two hot-spots of future warming regions over the represented land (20°N–60°N, 0–70°E) and ocean (15°N–55°N, 170°E–130°W), respectively. **c–d** Inter-model relationship of future UDZC and future surface warming over the hotspot of land and ocean. The thick red dash line represents the ordinary least-squares fit, with shading as the 95% confidence interval.

Fig. R7 (i.e., Supplementary Fig. 1) | Evolution of High Summer UDZC in different sliding window. Sliding correlation of WJA and EJA indices in 17-year (blue), 19-year (green), 21-year (red), 23-year (black) and 25-year (purple) sliding window using ERA-5.

Minor comments

1. L353-354: Which results or references are you refer to, especially for the past two centuries?

Response: Thank you for pointing out this issue. We agree that the original statement lacked specific attribution. We have rewritten this sentence to be more specific on the innovation of our results and explicitly reference the 20thCRV3 dataset for the past two centuries:

L367–373: “Our primary innovation is list as follows: (1) conceptualizing the UDZC of the Eurasian jet and quantifying its state through the correlation between WJA and EJA, (2) revealing its rapid intensification since the 1970s (bottom panel of Fig. 1c), and (3) demonstrating this enhancement to be unprecedented over the past two centuries using the 20thCRV3 (green curve with shading in top panel of Fig. 1c), which shows consistently weak and statistically insignificant upstream-downstream correlations before the satellite era, although accounting for higher uncertainty in the pre-1950 periods.”

2. L582-291: This part is not necessary because the NCL software is generally used in climatology, hydrology, and other related research.

Response: Done as suggestion.

3. Figure 3: Regression maps are shown for the influences of UDZC on climate in the Euroasia during the different periods. I suggest that the position jet stream (climatology of jet stream axis) should also be shown for different periods because the climatology of the jet stream axis has been shown in Figure 2.

Response: Thanks for this suggestion. We have modified this figure according to your suggestion (**Fig. R8**). In addition, we also marked the WJA and EJA part with red and blue colors.

Fig. R8 (Fig. 3 in revision) | Impacts of enhancing UDZC on Eurasian WCEs. **a** the normalized projected ESWJ jet index (black bar) with the WJA (blue curve) index and the EJA (red curve) index. **b** Regression map of NET-heatwave frequency against the normalized projected ESWJ jet index during P1, with 95%

confidence hatched. **c** and **d** Similar to **b**, but for the SPEI and rainfall anomalies, respectively. The blue (red) line indicates the climatology of WJA (EJA), similar to that in Fig. 1a but obtained during P1. **e–g** Similar to **b–d**, but for the P2. Also, the blue (red) line is obtained during P2. Note that here the rainfall data over land and ocean adopted the NOAA Land and GPCP precipitation, respectively. So, in **d**, the regressed rainfall anomalies from GPCP over ocean are calculated during 1979–1998.

4. Figure 7: Why is the correlation between the WJA and EJA (UDZC) from reanalysis is much stronger than that from the CMIP6 and LENS2? Why the variability of the UDZC from reanalysis data is larger than from the CMIP and LENS2? These analyses are very important to explain the stability of the UDZC and its physical mechanism.

Response: We have added the follow discussions in the revised manuscript:

L403–406: “As shown in Fig. 7a, the CMIP6 and LENS2 can generally capture the increasing features of R (WJA, EJA) under global warming, albeit relative weaker than that from Reanalysis data since about the year 2000.”

L411–421: “The CMIP6 and LENS2 model’s failure to reproduce the UDZC evolution can be attributed to the following reasons: (1) the physical processes and projected warming amplitudes among the models is very different (**Fig. 7c, d**). (2) In addition to model bias in the jet stream climatology^{55–57}, current models’ capabilities in also poor in capturing the variation in jet stream intensity^{56,58}, which leads to the poor performance of UDZC in Eurasian jet intensity. (3) Previous study has shown that climate models do not capture well the significant increase in jet intensity variability under global warming²⁸. (4) CMIP6 models’ insufficient capability of capturing the coupling between Northeast Atlantic SST and Eurasian jet. We have performed EMCA similar to Fig.6 in each 42 CMIP6 models during 1979–2014 and found that only ten out of 42 models can generally simulate the coupling between Northeast Atlantic SST and Eurasian jet, albeit the simulated relationship is relatively weaker than observation (Supplementary Figure 19).”

References:

28. Lin, L. *et al.* Atlantic origin of the increasing Asian westerly jet interannual variability. *Nat. Commun.* **15**, 2155 (2024).
 55. Zhou, B. *et al.* Quantitative evaluations of subtropical westerly jet simulations over East Asia based on multiple CMIP5 and CMIP6 GCMs. *Atmos. Res.* **276**, 106257 (2022).
 56. Yan, Y., Li, C. & Lu, R. Meridional Displacement of the East Asian Upper-tropospheric Westerly Jet and Its Relationship with the East Asian Summer Rainfall in CMIP5 Simulations. *Adv. Atmos. Sci.* **36**, 1203–1216 (2019).
 57. Lin, Z. *et al.* Intermodel Diversity in the Zonal Location of the Climatological East Asian Westerly Jet Core in Summer and Association with Rainfall over East Asia in CMIP5 Models. *Adv. Atmos. Sci.* **36**, 614–622 (2019).
 58. Li, C. & Lin, Z. Predictability of the summer East Asian upper-tropospheric westerly jet in ENSEMBLES multi-model forecasts. *Adv. Atmos. Sci.* **32**, 1669–1682 (2015).
-
5. Minor edit in Lines: 466-467: "The significance of the "?

Response: Corrected, thanks.